# Yeast TLDc domain proteins regulate assembly state and subcellular localization of the V-ATPase

Samira Klössel [1], Ying Zhu [2], Lucia Amado[1], Daniel D Bisinski[1], Julia Ruta [2], Fan Liu[2,3] & Ayelén González Montoro [1,4]✉

## Abstract

**Yeast vacuoles perform crucial cellular functions as acidic degradative organelles, storage compartments, and signaling hubs. These functions are mediated by important protein complexes, including the vacuolar-type H⁺-ATPase (V-ATPase), responsible for organelle acidification. To gain a more detailed understanding of vacuole function, we performed cross-linking mass spectrometry on isolated vacuoles, detecting many known as well as novel protein-protein interactions. Among these, we identified the uncharacterized TLDc-domain-containing protein Rtc5 as a novel interactor of the V-ATPase. We further analyzed the influence of Rtc5 and of Oxr1, the only other yeast TLDc-domain-containing protein, on V-ATPase function. We find that both Rtc5 and Oxr1 promote the disassembly of the vacuolar V-ATPase in vivo, counteracting the role of the RAVE complex, a V-ATPase assembly chaperone. Furthermore, Oxr1 is necessary for the retention of a Golgi-specific subunit of the V-ATPase in this compartment. Collectively, our results shed light on the in vivo roles of yeast TLDc-domain proteins as regulators of the V-ATPase, highlighting the multifaceted regulation of this crucial protein complex.**

**Keywords** Vacuole; Cross-linking Mass Spectrometry; V-ATPase; TLDc
**Subject Category** Organelles

## Introduction

Lysosomes and their yeast counterparts, vacuoles, serve as crucial catabolic organelles within cells, facilitating the recycling of macromolecules into reusable building blocks. The degradative capacity of these organelles relies on luminal hydrolases and on the acidic lumen of the organelle. Moreover, vacuoles play essential roles in amino acid and ion storage, sequestration of toxic molecules, and serve as vital signaling hubs (Li and Kane, 2009; Binda et al, 2009). These metabolic, homeostatic, and signaling functions establish the vacuole/lysosome as a central hub in cellular physiology. Notably, dysfunctions in lysosomal processes contribute significantly to various diseases, including lysosomal storage disorders, as well as neurodegenerative conditions characterized by protein deposition, such as Parkinson's and Alzheimer's diseases (Colacurcio and Nixon, 2016).

The protein complexes that reside in this organelle mediate these important functions. For example, acidification and thus hydrolytic capacity rely on the action of the vacuolar-type H⁺-ATPase (V-ATPase), a conserved multi-subunit protein complex that pumps protons into the lumen of the organelle, through a rotary mechanism energized by ATP hydrolysis (reviewed in (Vasanthakumar and Rubinstein, 2020)). The generated proton gradient is used as a driving force for the accumulation of amino acids or ions for storage, through the action of specific transporters (Jefferies et al, 2008; Banerjee and Kane, 2020). The role of the vacuole in signaling relies on the presence of the TORC1 complex, a major regulator of cell growth, and its upstream regulators, the EGO complex, and the SEA complex (Péli-Gulli et al, 2015; Panchaud et al, 2013a; Dokudovskaya et al, 2011; Panchaud et al, 2013b). Finally, phosphorylated versions of phosphatidylinositol are important identity determinants for organelles of the endolysosomal system. In endosomes, phosphatidylinositol can be phosphorylated by the Vps34 complex II, to generate PI(3)P (Schu et al, 1993). PI(3)P can be further phosphorylated to PI(3,5)P₂ in the vacuolar membrane, by the action of the Fab1 kinase complex, homologous to the mammalian PIK-FYVE complex (Gary et al, 1998). These lipids are important identity determinants, recognized by proteins involved in vesicular transport, lipid metabolism, and transport, as well as signaling.

To characterize in detail the interactions among the proteins of the vacuole, we have performed cross-linking mass spectrometry (XL-MS) on vacuoles isolated from *Saccharomyces cerevisiae* cells. We were able to recapitulate many known interactions and structural information about these major protein complexes, and we detected many cross-links that could indicate novel protein–protein interactions. Among these, we focus on the protein Rtc5, a protein of unknown function, which we found to be cross-linked to different subunits of the V-ATPase.

V-ATPases are present throughout eukaryotic organisms and can be located either in the membranes of intracellular

¹Osnabrück University, Department of Biology/Chemistry, Cellular Communication Laboratory, Barbarastrasse 13, 49076 Osnabrück, Germany. ²Department of Structural Biology, Leibniz – Forschungsinstitut für Molekulare Pharmakologie (FMP), Robert-Roessle-Str. 10, Berlin 13125, Germany. ³Charité - Universitätsmedizin Berlin, Charitépl. 1, 10117 Berlin, Germany. ⁴Osnabrück University, Center of Cellular Nanoanalytic Osnabrück (CellNanOs), Barbarastrasse 11, 49076 Osnabrück, Germany.
✉E-mail: ayelen.gonzalez.montoro@uos.de

compartments like the Golgi complex, endosomes, lysosomes, and secretory vesicles or in the plasma membrane. In intracellular compartments, they are the main source of acidification of the lumen of these organelles, and the generated proton gradient energizes the transport of other metabolites and plays crucial roles in protein trafficking, including secretion and endocytosis. On the other hand, V-ATPases present in the plasma membrane of specialized animal tissues pump protons into the extracellular space and are essential for bone remodeling, sperm maturation, and blood pH maintenance (Merkulova et al, 2015; Collins and Forgac, 2020). Furthermore, the activity of this complex is of important clinical relevance because the low pH of endocytic compartments acts as a trigger for infection by viruses like Ebola or Influenza, and because its dysregulation is associated with ageing and neurodegenerative disorders (Collins and Forgac, 2020; Colacurcio and Nixon, 2016).

V-ATPases consist of two domains: a membrane-embedded domain ($V_O$) and a peripheral domain ($V_1$). The $V_1$ domain mediates ATP hydrolysis, and the released energy is translated into a rotational motion that drives proton pumping through the $V_O$ domain (reviewed in (Marshansky et al, 2014)). Given the crucial function of V-ATPases for organelle and cell homeostasis, its activity is tightly regulated, and interconnected with other processes and signaling pathways. One of the main regulatory mechanisms, which is conserved from yeast to mammals, involves the reversible dissociation of the $V_1$ domain from the $V_O$ domain, resulting in the inactivation of the pump. In yeast, the main trigger for disassembly is glucose deprivation (Bond and Forgac, 2008), and is reversed when cells re-encounter high glucose levels. However, other parameters like pH and osmotic stress can also influence the levels of assembly of the complex (Dechant et al, 2010; Li et al, 2014). The re-assembly of the $V_1$ onto the $V_O$ domain when glucose becomes available again, is aided by a dedicated chaperone complex known as the RAVE complex, which also likely has a general role in V-ATPase assembly (Seol et al, 2001; Smardon et al, 2002, 2014).

Recent findings indicate that all mammalian proteins containing a domain called TLDc (Tre2/Bub2/Cdc16, LysM, domain catalytic) interact with the V-ATPase (Eaton et al, 2021; Tan et al, 2022; Merkulova et al, 2015). Also recently, the yeast TLDc domain-containing protein Oxr1 was shown to interact with the $V_1$ peripheral domain of the V-ATPase and to mediate its disassembly in vitro (Khan et al, 2022). However, in vivo information on this protein was lacking, and previous studies suggested a localization of Oxr1 at mitochondria (Elliott and Volkert, 2004). In our cross-linking mass spectrometry interactome map of isolated vacuoles we found that the only other TLDc-domain-containing protein of yeast, Rtc5, is a novel interactor of the V-ATPase. Rtc5 is a protein of unknown function, originally described in a genetic screen for genes related to telomere capping (Addinall et al, 2008). In this work, we show that Rtc5 is a vacuolar protein, and that it depends on its interaction with the V-ATPase and N-terminal myristoylation to achieve this localization. We further characterize that both yeast TLDc domain-containing proteins, Oxr1 and Rtc5, promote the disassembly of the V-ATPase in vivo. Furthermore, the lack of Oxr1 results in the re-localization of the Golgi-localized isoform of V-ATPase subunit a (Stv1) to the vacuole. Thus, TLDc domain-containing proteins are novel regulators of V-ATPase function in yeast, adding to the complex network of regulators of these enzymes.

# Results

## A cross-linking mass spectrometry map of vacuolar protein interactions

We isolated intact vacuoles from yeast cells using the established ficoll gradient protocol (Haas, 1995) and cross-linked them with Azide-A-DSBSO (Kaake et al, 2014), a biomembrane-permeable cross-linker that connects lysine residues within and between proteins that are in reach of its spacer arm (Fig. 1A). Mass spectrometric analysis revealed 16694 unique lysine-lysine connections among 2051 proteins at a 2% false discovery rate (FDR), including 11658 intra-protein and 5036 inter-protein cross-links (Dataset EV1). Our cross-linking mass spectrometry (XL-MS) data covered several well-characterized vacuolar protein complexes, many of which form a tightly connected subnetwork (Fig. 1B).

To confirm that our XL-MS approach preserved the structural integrity of vacuolar proteins, we mapped the identified cross-links onto available high-resolution structures of vacuolar protein complexes. Considering the lengths of the cross-linker spacer and lysine side chains as well as a certain degree of in-solution flexibility not captured in the analyzed crystallographic or cryo-EM structural snapshots, distances of up to 35 Å between Cα atoms of cross-linked residues are conceivable. For structural mapping of the cross-links, we used the structures of the HOPS tethering complex (Shvarev et al, 2022), the AP-3 vesicle-forming complex (Schoppe et al, 2021), the PI3K complex II (Rostislavleva et al, 2015), and the EGO, SEA, and TORC1 complexes, involved in signaling (Zhang et al, 2019; Tafur et al, 2022; Prouteau et al, 2023) (Fig. 1C and Appendix Fig. S1). We excluded the V-ATPase in this analysis due to the co-existence of many conformational and rotational states. Because of this mobility, we only expect a partial agreement of cross-links to a steady-state structure, as observed for structurally similar complexes such as the ATP synthase in other studies (Schweppe et al, 2017; Liu et al, 2018). For the selected complexes, we found a 95.2% agreement between the measured cross-link distances and existing structures, suggesting a high fidelity of the XL-MS dataset (Fig. 1C,D; Appendix Fig. S1; and Dataset EV2).

In addition to reproducing known vacuolar protein complexes, our data also showed numerous cross-links representative of known protein–protein interactions (Dataset EV1). For example, we found cross-links between the vacuolar Rab GTPase Ypt7 and its effector proteins Vam6, Vps41, and Ivy1 (Numrich et al, 2015; Lürick et al, 2017). The protein Pib2 has recently arisen as an important regulator of the TORC1 complex, and our dataset included cross-links of this protein to three different subunits of the complex (Kog1, Lst8, and Tor1) (Hatakeyama, 2021). In addition, we identified numerous cross-links representative of the formation of SNARE complexes during fusion events involving the vacuole (Vam3-Nyv1, Vam7-Vti1, Vam7-Vam3, Vti1-Nyv1, and Ykt6-Vam3) as well as interactions of SNAREs with the SNARE binding module of the HOPS tethering complex (Nyv1-Vps16, Nyv1-Vps33, and Vam3-Vps16) or with the homolog of alpha-SNAP Sec17 (Nyv1-Sec17, Ykt6-Sec17, and Vam3-Sec17). We also observed a cross-link between the ubiquitin ligase Rsp5 and its adapter protein Ssh4 (Léon et al, 2008) and a cross-link between the palmitoyltransferase Akr1 and the palmitoylated protein Lcb4, likely representing an enzyme–substrate interaction (Roth et al, 2006).

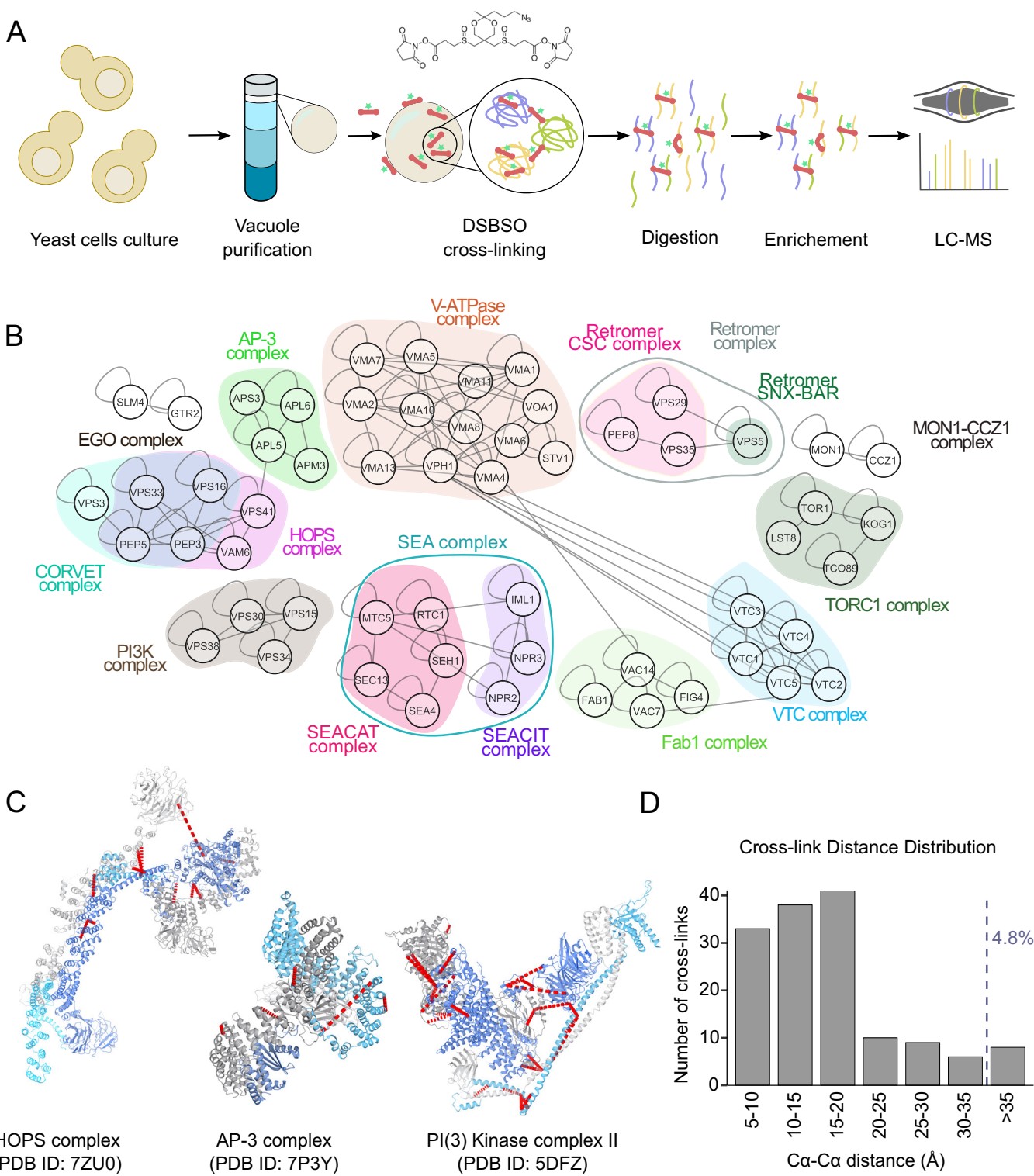

Figure 1. A XL-MS-based vacuole interactome.

(A) Schematic representation of vacuolar XL-MS workflow. (B) XL-MS-based vacuolar interactome; selected PPIs corresponding to known vacuolar protein complexes are shown. All PPIs are listed in Dataset EV1. (C, D) Cross-link mapping onto available high-resolution structures of selected vacuolar protein complexes, including HOPS, AP-3, PI3K complex II (shown in C), EGO, SEA, and TORC1 (shown in Appendix Fig. S1). Cross-links are shown in red dashed lines. The cross-link distances were measured between Cα-Cα of the two linked lysines, using the measuring function of Pymol v.2.5.2. The graph in (D) shows the distance distribution of the mapped cross-links. The allowed maximum distance restraint for the DSBSO cross-linker is considered 35 Å.

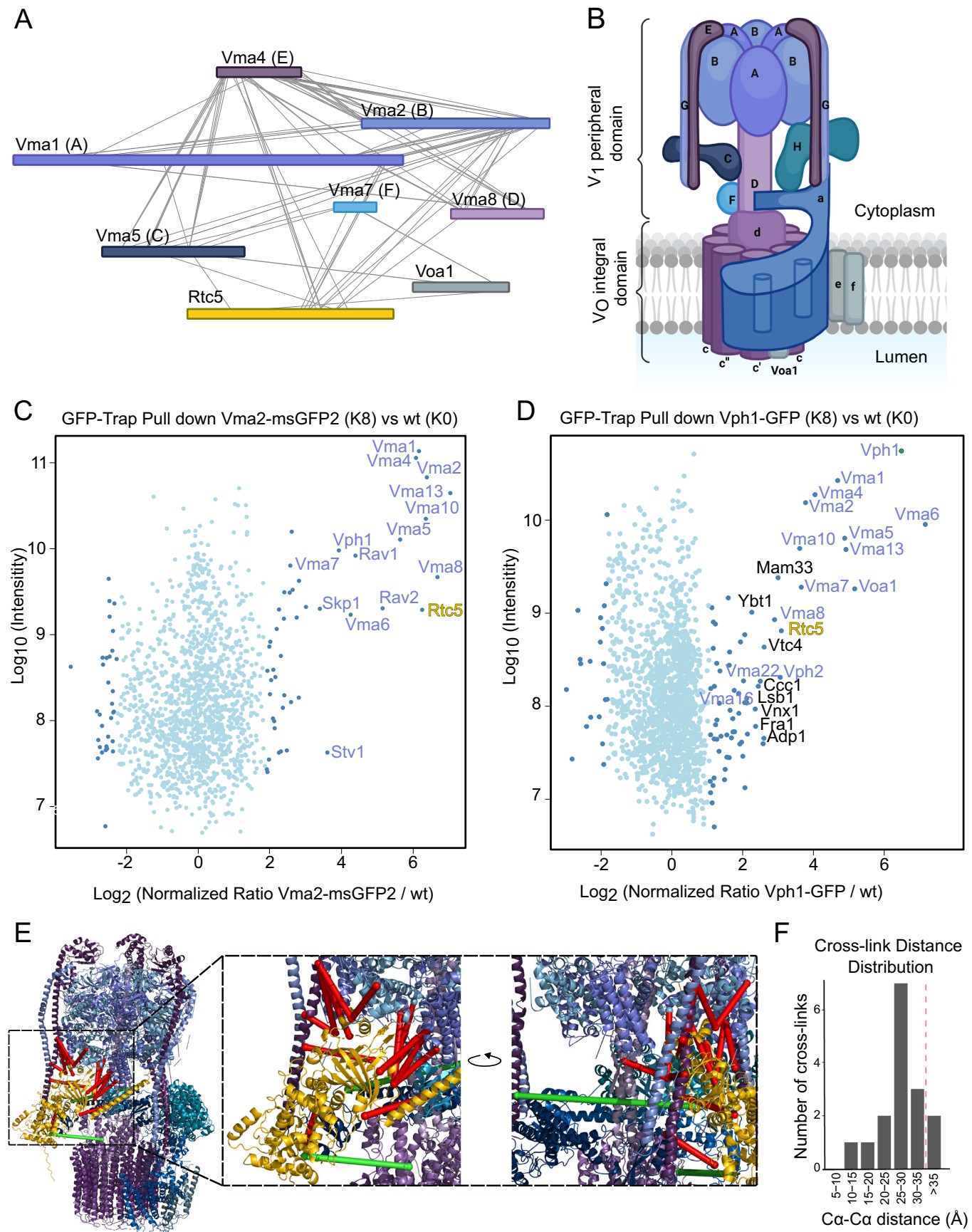

**Figure 2. Rtc5 is a novel interactor of the V-ATPase.**

(A) XL-based interactions of V-ATPase subunits with Rtc5. (B) Diagram of the structure of the V-ATPase, with the different subunits labeled. The subunits found cross-linked to Rtc5 are shown in the same color in panels (A) and (B). Adapted from BioRender.com. Retrieved from https://app.biorender.com/biorender-templates. (C, D) SILAC-based GFP-Trap pull down of Vma2–msGFP2 (C) or Vph1-GFP (D) and mass spectrometry analysis. Light isotope-labeled control cells and heavy isotope-labeled cells expressing Vma2–msGFP2 or Vph1-GFP were used. The Log10 of protein intensity is plotted against the Log2 of the normalized heavy/light SILAC ratio. The dots represent proteins with significant enrichment based on a two-group, two-tailed Student's t-test. ($P < 0.05$) are colored dark blue; other detected proteins are shown in light blue. The names of V-ATPase subunits and known interacting proteins are shown in purple, Rtc5 is colored in yellow. Other proteins enriched with $P$ value <0.01 are labeled in black. (E) Model of Rtc5 bound to the V-ATPase. The model was created by docking with HADDOCK, the Alphafold-generated model of Rtc5, and the available V-ATPase structure, with the detected cross-links as restraints. The model shown is the one in best agreement with the cross-link data. The subunits of the V-ATPase are shown in the same color as in the diagram in panel (B) and Rtc5 is shown in yellow. Cross-links that fall below the 35 Å range are shown as red lines, while the crosslinks above this distance are shown as green lines. (F) Distance distribution of the 16 cross-links detected between Rtc5 and the V-ATPase when mapped onto the structure shown in panel (E). 35 Å are considered as the allowed maximum distance restraint of the DSBSO cross-linker.

Taken together, our results show that our cross-linking dataset reproduces the interactions of vacuolar protein complexes in good agreement with known structures, as well as interactions representing regulatory functions, enzyme–substrate pairs, and interactions formed during membrane fusion processes. This speaks for the high quality of the dataset and suggests that many of the novel protein–protein interactions found are likely relevant in vivo.

## The TLDc-domain-containing protein of unknown function Rtc5 is a novel interactor of the vacuolar V-ATPase

In the XL-based protein interaction network, the V-ATPase complex is shown as an interaction hub within the vacuole, and interacts with a number of proteins that have not been previously reported as its binding partners. One of these potential new interactors is the protein Rtc5, which cross-linked to several V-ATPase subunits (Fig. 2A). For reference, a diagram of the structure of the V-ATPase is shown in Fig. 2B. Rtc5 is a protein of unknown function, which was identified in a genetic screen for genetic interactors of a mutant version of Cdc13, an adapter of telomerase, hence its name of restriction of telomere capping (Addinall et al, 2008). Rtc5 is predicted to contain a TLDc domain (Tre2/Bub2/Cdc16, LysM, domain catalytic, Pfam PF07534, Inter-Pro IPR006571, Prosite PS51886, and SMART SM00584). Recent studies described proteins containing this domain from different organisms as interactors of the V-ATPase. Mouse Ncoa7 and Oxr1 were identified as interactors in a proteomics approach using immunoprecipitated V-ATPases from mouse kidneys (Merkulova et al, 2015), and mEAK7 (Tldc1) was observed by cryo-EM data mining in a small proportion of purified porcine kidney V-ATPases (Tan et al, 2022). Finally, *Saccharomyces cerevisiae* Oxr1, which is the only other protein in yeast containing a TLDc domain, was co-purified with a $V_1$ domain of the V-ATPase containing a mutant version of subunit H (Khan et al, 2022).

To confirm the interaction of Rtc5 with the V-ATPase, we performed quantitative (SILAC-based) affinity purification mass spectrometry experiments with C-terminally GFP-tagged versions of the $V_1$ subunit B (Vma2) or the vacuolar $V_O$ subunit a (Vph1) as baits. In both cases, we could observe strong enrichment of V-ATPase subunits, known V-ATPase assembly factors, and Rtc5 (Fig. 2C,D). The fact that we can co-enrich Rtc5 both with Vma2 and with Vph1 indicates that it can interact either with both the $V_O$ and $V_1$ domains or with the assembled V-ATPase.

Since the structure of Rtc5 is not yet available, we performed a structural prediction using AlphaFold (Jumper et al, 2021). Based

on this model, the high-resolution structure of the V-ATPase complex (7FDA), and the 16 cross-links between Rtc5 and V-ATPase, we generated models of the V-ATPase bound to Rtc5 using the HADDOCK web server (Honorato et al, 2021; Van Zundert et al, 2016). The best-scoring model (Fig. 2E) was in good agreement with the cross-linking data: 14 out of 16 cross-links were below the 35 Å distance restraint (Fig. 2E,F; Dataset EV2). Furthermore, one of the two over-length cross-links is Vma5 (subunit C)-K125-Rtc5-K425. Since Vma5 is released during V-ATPase disassembly, our hypothesis is that this cross-link is formed on the intermediates of the assembly or disassembly process of the V-ATPase (Tabke et al, 2014). This hypothesis is in line with the function of Rtc5 in V-ATPase disassembly discussed below.

## Rtc5 localizes to the vacuolar membrane and this localization depends on an assembled V-ATPase

Analysis of the in vivo localization of Rtc5 C-terminally tagged with mNeonGreen (Rtc5-mNG) shows that the protein localizes to the vacuolar membrane, as assessed by co-localization with endocytosed FM4-64 (Fig. 3A).

We then addressed if the interaction with the V-ATPase plays a role in the subcellular localization of Rtc5. Thus, we determined the localization of Rtc5 in strains lacking subunits *VMA4*, *VMA5*, or *VMA11* of the V-ATPase, which results in failure to assemble the V-ATPase complex (Jefferies et al, 2008). In these deletion strains, the localization of Rtc5 from the vacuolar membrane was lost, and the protein was completely cytosolic (Fig. 3A). Furthermore, we deleted *VPH1*, the gene that encodes for the vacuole-specific isoform of subunit a. This deletion should result in lack of most V-ATPases complexes from the vacuole membrane. This also resulted in cytosolic localization of Rtc5, indicating that it specifically requires the assembly of the vacuole-localized V-ATPase (Fig. 3A). To the right of the images, we show the fluorescence intensity of the two fluorophores along lines that traverse the vacuole, shown in yellow in the merged images. This analysis shows that while Rtc5-mNG shows signal peaks that coincide with peaks of FM4-64 in the control strain, i.e., at the vacuolar membrane, this is not the case in the strains that lack assembled V-ATPases (*vma4Δ*, *vma5Δ*, or *vma11Δ*) or the strain that lacks the vacuolar isoform of subunit a (*vph1Δ*).

The V-ATPase is regulated by disassembly of the $V_1$ domain from the $V_O$ domain. The main trigger for this in yeast is the availability of glucose, the lack of which results in disassembly (Parra and Kane, 1998). This model was based on experiments that assessed the co-purification of the two domains or the association

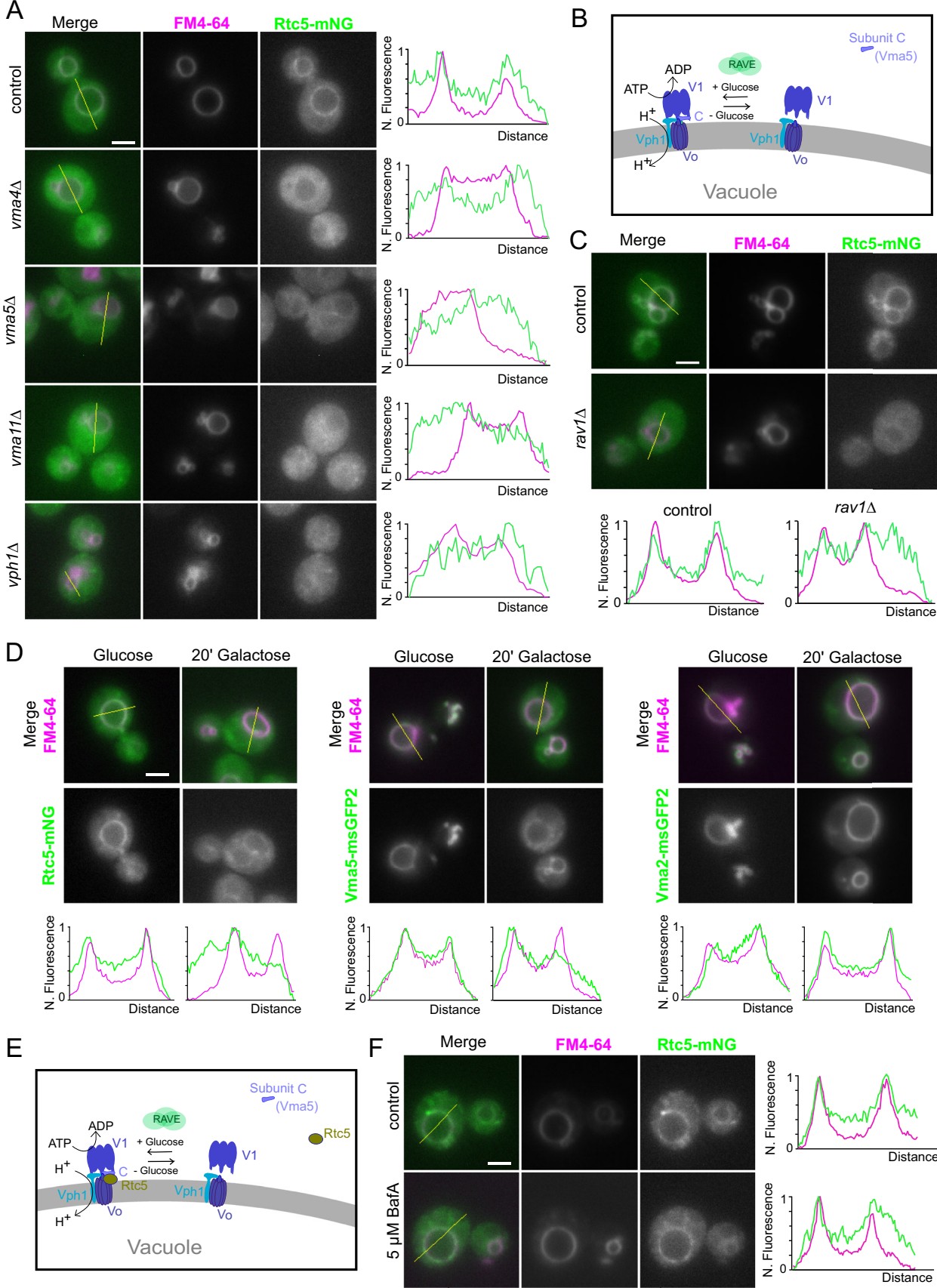

◄ **Figure 3.  Rtc5 localizes to the vacuole membrane dependent on an assembled V-ATPase.**

(A) Fluorescence microscopy analysis of the subcellular localization of Rtc5-mNeonGreen (Rtc5-mNG) in a control strain or strains lacking different subunits of the V-ATPase: Vma4, Vma5, Vma11, or Vph1. The vacuolar membrane is stained with FM4-64. The line profiles to the right show the normalized fluorescence of the FM4-64 staining and the mNeonGreen signal of Rtc5 along the yellow lines shown in the merged images. The scale bar represents 2 μm. (B) Diagram representing the regulatory mechanism of disassembly of the V-ATPase upon glucose depletion. In conditions of high glucose availability, the $V_1$-$V_O$ complex assembles and is active. When glucose is limited, subunit C (Vma5) is released from the vacuole membrane to the cytosol, the interaction between the two domains is weakened, and the complex becomes inactive. The re-assembly of the complex upon glucose becoming available again is aided by a specialized chaperone called the RAVE complex, the main subunit of which is the protein Rav1. (C, D) Fluorescence microscopy analysis of the subcellular localization of Rtc5-mNeonGreen to the vacuole membrane in conditions of V-ATPase disassembly. In (C), the disassembly of the V-ATPase is caused by disrupting the RAVE complex with the deletion of *RAV1*. In (D), cells were shifted to galactose for 20 min. The localization to the vacuole membrane is evidenced by line profile analysis along the lines shown in yellow in the merged images, like in panel (A). The scale bars represent 2 μm. (E) Updated model of V-ATPase disassembly, including that under low-glucose conditions, both subunit C and Rtc5 are released into the cytosol. (F) Fluorescence microscopy analysis of the subcellular localization of Rtc5-mNeonGreen to the vacuole membrane in conditions of V-ATPase inhibition by 30 min of treatment with the V-ATPase inhibitor Bafilomycin A1 (5 μM). Localization to the vacuole membrane is evidenced by line profile analysis along the lines shown in yellow in the merged images. The scale bars represent 2 μm. Source data are available online for this figure.

of the $V_1$ domain with membranes (Parra and Kane, 1998). In vivo analysis of the subcellular localization of different $V_1$ subunits after 20 min of growth in a medium containing galactose as the sole carbon source, showed that it is only subunit C (Vma5) that actually loses the vacuolar localization (Tabke et al, 2014). A model that combines all of these findings is that upon low glucose availability, subunit C leaves the vacuole, and the rest of the complex adopts an inactive conformation in which the interaction between the two domains is weakened and thus does not withstand co-purification (Fig. 3B). A specific chaperone complex called the RAVE complex, the main subunit of which is the protein Rav1, aids the re-assembly of the $V_1$-$V_O$ complex when glucose becomes available again (Seol et al, 2001). In addition, RAVE has a more general role in V-ATPase assembly, and thus the V-ATPase complex has serious structural and functional defects in cells lacking RAVE subunits (Smardon et al, 2002). Therefore, to further test the requirement of an assembled V-ATPase, we tested the localization of Rtc5 upon glucose depletion or upon deletion of Rav1, two conditions that should result in reduced assembly of the V-ATPase.

Figure 3C shows that the deletion of *RAV1* also results in the mislocalization of Rtc5 to the cytosol, further confirming the requirement of an assembled V-ATPase. In addition, shifting cells to a medium containing galactose as the sole carbon source for 20 min, resulted in Vma5 (subunit C) but not subunit B (Vma2) re-localizing to the cytosol (Fig. 3D), as previously reported (Tabke et al, 2014). Under these conditions, Rtc5 also re-located to the cytosol (Fig. 3D). Thus, we would like to update the current model regarding the disassembly of the V-ATPase: under glucose depletion, the interaction between the $V_1$ and $V_O$ domains is weakened, and Subunit C and Rtc5 leave the vacuole membrane (Fig. 3E). Considering that Rtc5 is co-enriched with subunits of both the $V_O$ and $V_1$ domain, that it was cross-linked to the V-ATPase on isolated vacuoles and that it localizes at the vacuole membrane dependent on an assembled V-ATPase, we suggest that Rtc5 interacts with the assembled V-ATPase complex.

Finally, we assessed if V-ATPase activity is required for the localization of Rtc5 to the vacuole. Incubation of cells with 5 μM Bafilomycin A1 for 30 min, conditions that inhibit V-ATPase activity in vivo, did not affect vacuolar localization of Rtc5. This indicates that Rtc5 localization is not dependent on V-ATPase activity (Fig. 3F). Altogether, our results show that Rtc5 requires an assembled V-ATPase, but not an active one, to localize to the vacuole membrane.

## Rtc5 is N-myristoylated, and this modification is required for its localization to the vacuolar membrane

In contrast to the vacuolar localization of C-terminally tagged Rtc5, N-terminally tagged Rtc5 displayed a completely cytosolic signal (Fig. 4A). As before, analysis of the fluorescence signal along the indicated yellow lines that traverse the vacuole shows that while the signal of Rtc5-msGFP2 peaks at the vacuolar membrane, this is not the case for msGFP2-Rtc5. This suggested that a free N-terminus is a requirement for the protein to achieve its subcellular localization. Rtc5 contains a Glycine in position 2, which indicates potential N-myristoylation. Indeed, analysis of the protein sequence with the algorithm developed by (Maurer-Stroh et al, 2002), predicted a robust N-myristoylation sequence.

To confirm the N-myristoylation of Rtc5, we metabolically labeled cells with 12-azidododecanoic acid, an analog of myristic acid (Azido-Myr), which can be coupled to different molecules through a click-chemistry reaction. After lysis, we linked the fatty acid to biotin and pulled-down linked proteins with streptavidin beads. As can be observed in Fig. 4B, Rtc5 was specifically enriched in the labeled samples, confirming that the protein is N-myristoylated in vivo.

We then assessed the role of this modification in the localization of the protein, by mutating the Glycine 2 to Alanine (G2A) in the genome. The subcellular fractionation in Fig. 4C shows that the membrane association of Rtc5(G2A) is strongly reduced. Consistently, Fig. 4D shows that this mutant version localizes completely to the cytosol, indicating that the modification is necessary for the protein to achieve its subcellular localization.

We conclude that Rtc5 is an N-myristoylated protein and that it requires both this modification and the interaction with an assembled V-ATPase to localize to the vacuole membrane.

## Rtc5 and Oxr1 are not required for V-ATPase activity, and their deletion results in higher V-ATPase assembly in vivo

We have now confirmed the interaction of Rtc5 with the V-ATPase. The only other TLDc domain-containing protein of yeast, Oxr1, was not detectable in our XL-MS and pulldown experiments. In a previous study, Oxr1 was co-purified with a $V_1$ domain that contained a mutant version of the H subunit, and its presence prevented the in vitro assembly of this $V_1$ domain onto the $V_O$ domain. Furthermore, the addition of Oxr1 promoted the

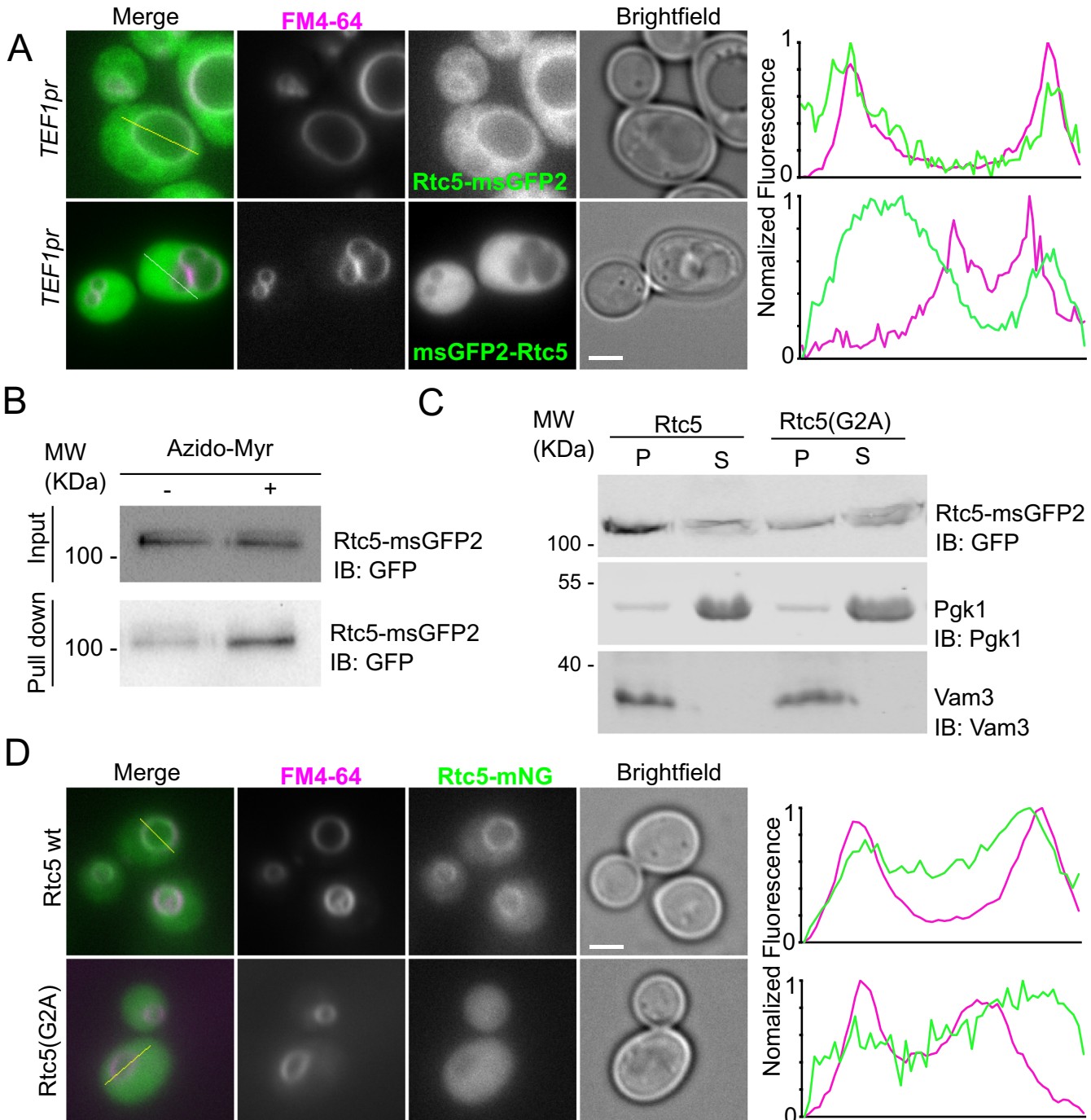

disassembly of the holocomplex (Khan et al, 2022). This is likely the reason why we do not detect Oxr1 in our experiments, which rely on isolated vacuoles and thus would only include $V_1$ domains that are assembled onto the membrane. In addition, Oxr1 is less abundant in yeast cells than Rtc5, according to the protein abundance database PaxDb (Wang et al, 2015). Because of our newly described interaction of Rtc5 with the V-ATPase and the previous report on Oxr1, we decided to characterize how these proteins affect V-ATPase function in vivo.

Rtc5 is a 567-residue-long protein. Analysis of the protein using HHPred (Zimmermann et al, 2018), finds homology to the

structure of porcine Meak7 (PDB ID: 7U8O, (Tan et al, 2022)) over the whole protein sequence (residues 37–559). For both yeast Rtc5 and human Meak7 (Uniprot ID: Q6P9B6), HHPred detects homology of the C-terminal region to other TLDc domain-containing proteins like yeast Oxr1 (PDBID: 7FDE), *Drosophila melanogaster* Skywalker (PDB ID: 6R82), and human NCOA7 (PDB ID: 7OBP), while the N-terminus has similarity to EF-hand domain calcium-binding proteins (PDB IDs: 1EG3, 2CT9, 1S6C6, Fig. 5A). HHPred analysis of the 273-residue long *Saccharomyces cerevisiae* Oxr1, on the other hand, only detects similarity to TLDc domain-containing proteins (PDB IDs: 7U80, 6R82, 7OBP), which

◄ **Figure 4. Rtc5 depends on N-myristoylation to localize to the vacuole membrane.**

(A) Fluorescence microscopy analysis of the localization of Rtc5 under the expression of the strong constitutive *TEF1* promoter when tagged C- or N-terminally with msGFP2. The vacuolar membrane is stained with endocytosed FM4-64. The line profiles to the right show the normalized fluorescence intensity of Rtc5 tagged with msGFP2 and FM4-64 along the yellow lines in the merged images. The scale bar represents 2 μm. (B) Cells expressing Rtc5-msGFP2 under the control of the strong constitutive *TEF1* promoter were labeled with azido-myristate (+) or mock-treated (−). A click chemistry-based conjugation of the azido-myristate with alkyne-biotin was performed in the lysates, and myristoylated proteins were pulled down using a streptavidin matrix. The immunoblot (IB) was performed with a primary antibody against GFP. An additional repetition of this experiment is shown in Appendix Fig. S2A. (C) Analysis of membrane association of Rtc5 and the Rtc5(G2A) mutant. A subcellular fractionation was performed using lysates from strains expressing C-terminally msGFP2 tagged Rtc5 or the Rtc5(G2A) mutant. Pgk1 is shown as a cytosolic marker protein, and Vam3 as an integral membrane protein marker. Next to each membrane, the protein recognized by the primary antibody used for the immunoblot is indicated (IB). Two additional repetitions of this experiment are shown in Appendix Fig. S2B,C. (D) Fluorescence microscopy analysis of strains expressing C-terminally mNeonGreen (mNG) tagged Rtc5 and the Rtc5(G2A) mutant. The vacuole membranes are stained with endocytosed FM4-64. Localization to the vacuole membrane is evidenced by line profile analysis across the organelle, along the yellow lines shown in the merged image. The scale bar represents 2 μm. Source data are available online for this figure.

spans the majority of the sequence of the protein (residues 71–273, Fig. 5A). The overall sequence identity between Oxr1 and Rtc5 is 24% according to a ClustalOmega alignment within Uniprot. The Alphafold model that we generated for Rtc5 is in good agreement with the available partial structure of yeast Oxr1 (7FDE) (root mean square deviation (RMSD) of 3.509 Å) (Fig. EV1A), indicating the proteins are structurally very similar, in the region of the TLDc domain. Taken together, these analyses suggest that Oxr1 belongs to a group of TLDc domain-containing proteins consisting mainly of just this domain like the splice variants Oxr1-C or NCOA7-B in humans (NP_001185464 and NP_001186551, respectively), while Rtc5 belongs to a group containing an additional N-terminal EF-hand-like domain and a N-myristoylation sequence, like human Meak7 (Finelli and Oliver, 2017) (Fig. 5A).

As a first approach for addressing the role of these proteins, we tested growth phenotypes related to V-ATPase function in strains lacking them. The V-ATPase is not essential for viability in yeast cells, and mutants lacking subunits of this complex grow similarly to a wt strain in acidic media, but display a growth defect at near-neutral pH (Nelson and Nelson, 1990). In addition, the proton gradient across the vacuole membrane generated by the V-ATPase energizes the pumping of metals into the vacuole, as a mechanism of detoxification. Thus, increasing concentrations of divalent cations, such as calcium and zinc, generate conditions in which growth is increasingly reliant on V-ATPase activity (Förster and Kane, 2000; MacDiarmid et al, 2002, 2000; Kane, 2006).

Thus, we tested if cells lacking *RTC5* or *OXR1* displayed impaired growth in media with near-neutral pH, with or without the addition of divalent cations. Figure 5B shows that while a strain lacking subunit E of the V-ATPase (*vma4Δ*) has a strong growth defect in neutral media containing zinc or calcium, deletion of neither *RTC5, OXR1*, or both together affects growth under these conditions. Consistently, none of these strains showed an altered vacuolar pH when measured with the BCECF ratiometric stain (Fig. 5C). Since glucose availability is one of the main regulators of V-ATPase function, we performed the same experiments in media containing galactose as the carbon source. The results were comparable to the ones in glucose: deletion of *RTC5, OXR1* or both has no effect on growth in a neutral pH medium with or without the addition of calcium or zinc or on vacuolar pH when measured with BCECF (Fig. EV1B,C). These experiments show that neither Oxr1 nor Rtc5 are required for the function of the V-ATPase.

To address the role of these proteins further, we analyzed how the proteome of vacuoles is affected when the proteins are missing using the recently developed vacuole proteomics technique (Eising et al, 2019). We found that vacuoles purified from cells lacking either *OXR1* or *RTC5* have higher amounts of subunits of the peripheral $V_1$ domain of the V-ATPase (Fig. 5D,E). Subunits Vma1 (A), Vma4 (E), Vma2 (B), Vma13 (H), Vma10 (G), Vma5 (C), Vma8 (D), and Vma7 (F), all components of the $V_1$ domain, were enriched significantly in both experiments. In contrast, Vma6 (d), a subunit of the $V_O$ domain, was detected but showed no significant enrichment (Fig. 5D). Performing the same experiments but switching which strain was labeled with heavy and light amino acids gave consistent results (Appendix Fig. S3A,B). Furthermore, this effect was not caused by the cells having higher levels of V-ATPase subunits, as an experiment addressing changes in the cellular proteome revealed that these were unaffected in the absence of *RTC5* or *OXR1* (Appendix Fig. S4A,B). Together, these data indicate that in cells lacking either Oxr1 or Rtc5, there are higher levels of assembled V-ATPase at the vacuole.

To address the situation in intact cells, we observed the in vivo localization of subunit C, Vma5, the only subunit of the $V_1$ domain that changes its subcellular localization, becoming cytosolic upon glucose depletion (Tabke et al, 2014). Our results are consistent with what was published, and we can quantify this decrease in the vacuolar localization of Vma5 after 20 min of growth in a medium containing galactose as the only carbon source, by making a ratio between the fluorescence at the vacuolar membrane and the cytosol (Fig. 5F,G). Under these conditions, which promote V-ATPase disassembly, Vma5 has a higher vacuolar enrichment in cells that lack either Oxr1 or Rtc5 (Fig. 5F,G).

Taken together, our results suggest that both yeast TLDc domain-containing proteins promote an in vivo state of lower assembly of the V-ATPase, in line with the previous report that in vitro Oxr1 promotes V-ATPase disassembly (Khan et al, 2022).

## Overexpression of Rtc5 or Oxr1 results in mild defects in vacuole acidification and decreased localization of Vma5 to the vacuole membrane

Since deletion of either *RTC5* or *OXR1* resulted in increased assembly of the V-ATPase, we tested the effect of overexpressing them. Overexpression of the proteins by placing them under the control of the strong constitutive *TEF1* promoter resulted in growth comparable to the control strain in neutral pH, even with the addition of 3 mM $ZnCl_2$ (Fig. 6A). However, upon conditions in which cells rely strongly on V-ATPase activity because of high concentrations of $ZnCl_2$ at pH = 7.5, cells overexpressing Oxr1 showed a mild growth impairment (Fig. 6A). This growth defect was more pronounced in strains that carried additionally the

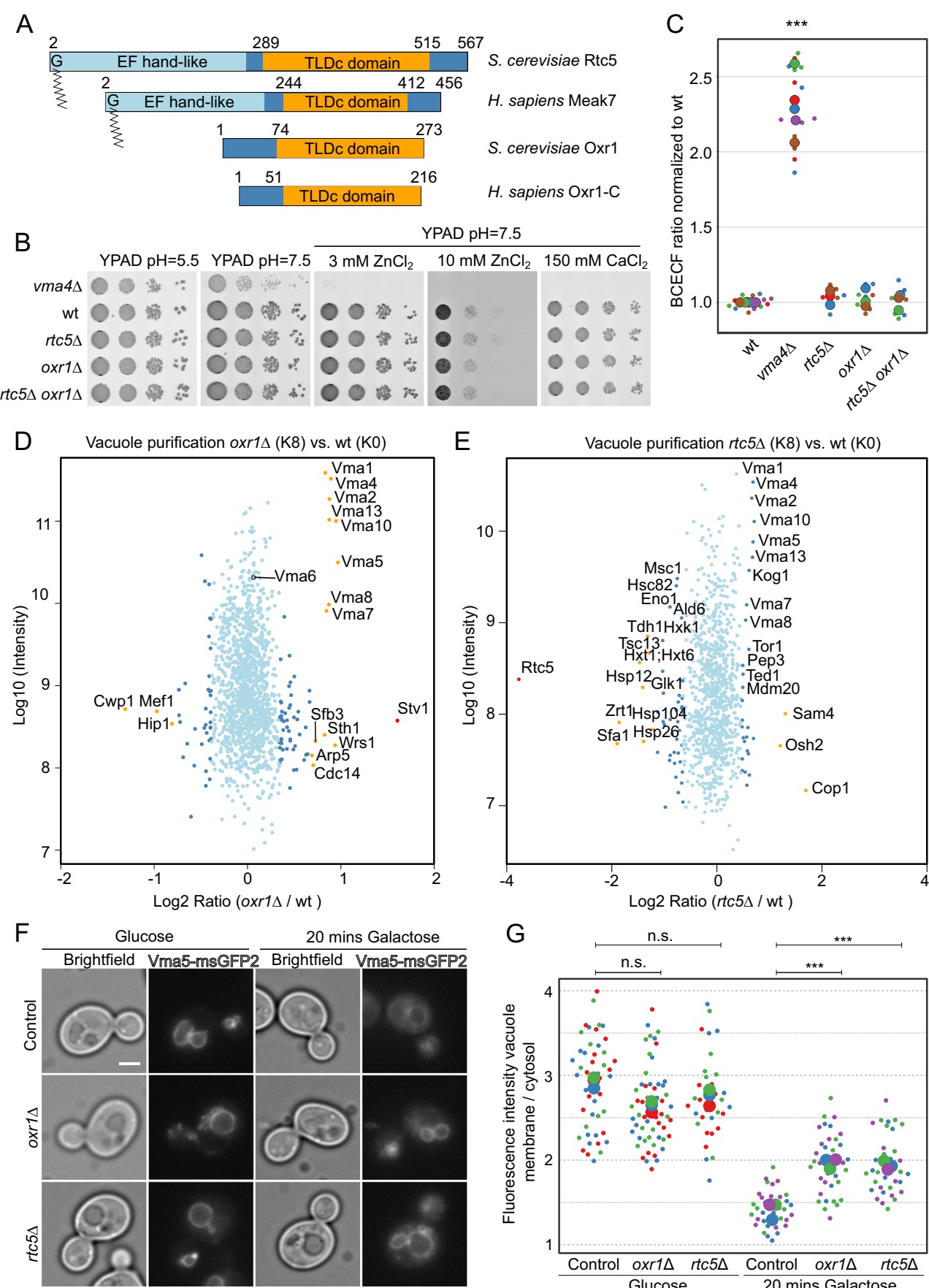

**Figure 5.  Rtc5 and Oxr1 are not necessary for V-ATPase function, and their deletion results in higher V-ATPase assembly in vivo.**

(A) Diagram of the domain organization of the two yeast TLDc domain-containing proteins, Oxr1 and Rtc5, together with two human proteins with a similar domain organization. *S. cerevisiae* Rtc5 and *H. sapiens* Meak7 are N-myristoylated proteins, which contain an N-terminal EF-hand-like domain and a C-terminal TLDc domain. *S. cerevisiae* Oxr1 and the splice variant Oxr1-C of *H. sapiens* consist mainly of just their TLDc domains. (B) A wt strain or strains lacking *VMA4, OXR1 or RTC5*, or both *OXR1 and RTC5* were spotted as serial dilutions on media containing glucose with pH = 5.5, pH = 7.5, or pH = 7.5 and 3 mM $ZnCl_2$, 10 mM $ZnCl_2$, or 150 mM $CaCl_2$. (C) Analysis of vacuolar acidity via BCECF staining in a wt strain, a strain lacking *VMA4, OXR1, RTC5*, or both *RTC5 and OXR1*. The experiments were performed with cultures grown in a medium containing glucose and pH = 5.5. For each strain, at least three independent experiments were performed, each containing three biological replicates. For each sample, the fluorescence emission of BCECF at 538 nm was measured when excited at 440 nm or 485 nm, and a ratio between these two values was calculated. The ratio was normalized to the average value for the wt strain in that experiment. The different colors in the graph indicate independent experiments, the smaller dots are biological replicates, and the larger circles represent the averages of each independent experiment. Statistical analysis was performed with a one-way ANOVA and a Tukey post hoc test. The *vma4Δ* strain was significantly different from the wt strain (***$P$ value <0.001), all other strains are not significantly different from the wt strain ($P$ value >0.05). (D, E) SILAC-based vacuole proteomics of cells lacking either *OXR1* (D) or *RTC5* (E) compared with the wt strain. Log10 of the detected protein intensities are plotted against Log2 of the heavy/light SILAC ratios. Significant outliers based on a two-group, two-tailed Student's *t*-test are color-coded in red ($P$ value <1e − 14), orange ($P$ value <0.0001), or dark blue ($P$ value <0.05); other identified proteins are shown in light blue. Corresponding experiments performed by switching the heavy and light labeling of the strains are shown in Appendix Fig. S3 A,B. (F, G) Cells in which Subunit C (Vma5) was tagged with msGFP2 were grown in a glucose-containing medium and imaged in the presence of glucose or after shifting them for 20 min to a medium containing galactose as the sole carbon source. The scale bar represents 2 μm. The distribution of Vma5 between the two compartments was quantified by using a ratio between the mean fluorescence intensity in a line along the vacuole membrane and the average of the mean fluorescence intensity in three circular regions in the cytosol. The different colors in the graph indicate independent experiments. The smaller circles represent individual cells, and the bigger circles represent the average for each independent experiment. Statistical comparison was performed using a one-way ANOVA and a Tukey post hoc test among the experimental means within each condition (glucose or galactose, n.s not significant $P$ value >0.05; ***$P$ value <0.001). Source data are available online for this figure.

overexpression of Rtc5 (Fig. 6A). This suggests that both proteins promote an in vivo condition with slightly lower V-ATPase activity. We observe no steady-state defect in the vacuole acidity of cells grown in the presence of glucose with BCECF (Fig. 6B), which correlates with the fact that the growth defects observed are only observable under highly stringent conditions with high concentrations of $ZnCl_2$. Performing the same experiments in a medium containing galactose as the carbon source showed similar results (Figs. 6C and EV2A).

Similar to the above, we decided to address the assembly state of the V-ATPase in intact cells, by analyzing the localization of Vma5 to the vacuole membrane in glucose-grown cells or after shift to medium containing galactose for 20 min. In the presence of glucose, we observed that cells overexpressing either Rtc5 or Oxr1 have lower levels of Vma5 on the vacuole membrane (Fig. 6D,E). These results further confirm that these proteins promote lower levels of V-ATPase assembly in vivo.

## Rtc5 and Oxr1 counteract the function of the RAVE complex

Our results so far indicate that both TLDc domain-containing proteins promote an in vivo state of lower V-ATPase assembly. However, cells overexpressing Oxr1 or Rtc5 only display a growth impairment under conditions that are strongly reliant on V-ATPase activity (Fig. 6A,C). Consistently, we did not observe less assembly of the vacuolar V-ATPase in vacuole proteomics experiments of cells overexpressing either Rtc5 or Oxr1 (Fig. EV2,C). One possible explanation for this is that the increased levels of Oxr1 or Rtc5 are counteracted by regulatory mechanisms.

We thus tested if overexpression of Rtc5 or Oxr1 would have a more pronounced effect in a sensitized background lacking Rav1, a critical subunit of the RAVE complex, which functions as a chaperone for the assembly of the V-ATPase (Smardon et al, 2002; Smardon and Kane, 2007). Indeed, cells overexpressing Oxr1 or Rtc5 from the strong constitutive *TEF1* promoter additionally to a *RAV1* deletion showed a growth defect in high-pH media containing $ZnCl_2$, to a much further level than the deletion of

*RAV1* alone (Fig. 7A,B). This effect was also observable by addressing vacuole acidity in vivo with BCECF, which showed a higher pH for *TEF1pr-OXR1 rav1Δ* and *TEF1pr-RTC5 rav1Δ* cells than for *rav1Δ* cells (Fig. 7C). In this sensitized background, overexpression of Oxr1 caused disassembly of the V-ATPase, detectable by vacuole proteomics analysis (Fig. 7D; Appendix Fig. S5A). The overexpression of Rtc5 in the same background did not cause a significant decrease in the amount of $V_1$ domain subunits at the vacuole (Appendix Fig. S5B,C), even though an effect on growth and vacuolar acidity was observable (Fig. 7B,C). Furthermore, the deletion of *OXR1* on top of *RAV1* deletion reduced the growth defect in media containing $ZnCl_2$ caused by *RAV1* deletion (Fig. 7E). Deletion of *RTC5* in this background, also had a positive effect on growth, although this was very minor (Fig. 7F). Analysis of the vacuolar acidity of these strains with BCECF showed results consistent with the growth phenotypes (Fig. 7G). Taken together, these results show that both TLDc-domain-containing proteins favor V-ATPase disassembly in vivo, and counteract the function of the RAVE complex.

Since only the vacuole-localized pool of the V-ATPase is regulated by reversible dissociation (Kawasaki-Nishi et al, 2001; Smardon et al, 2014), the overexpression of the TLDc-containing proteins should cause specifically a decrease of assembled V-ATPases at this organelle. We thus hypothesized that their overexpression should have a negative impact on growth when combined with the lack of the Golgi-localized V-ATPase isoform, similar to the growth defect caused by a double deletion of the two isoforms of subunit a (Manolson et al, 1994). Indeed, overexpression of either Rtc5 or Oxr1 resulted in increased growth defects in the context of *STV1* deletion (Fig. 7H,I).

The genetic interactions of *RTC5* and *OXR1* with *RAV1* allowed us to test the functionality of the tagged constructs. A genetic interaction occurs when the combination of two mutations results in a different phenotype from that expected from the addition of the phenotypes of the individual mutations. For example, deletion of *OXR1* or *RTC5* has no impact on growth in neutral pH media containing $ZnCl_2$ in a control background but improves the growth of *rav1Δ* strains (Fig. 7E,F), so this is a positive genetic interaction.

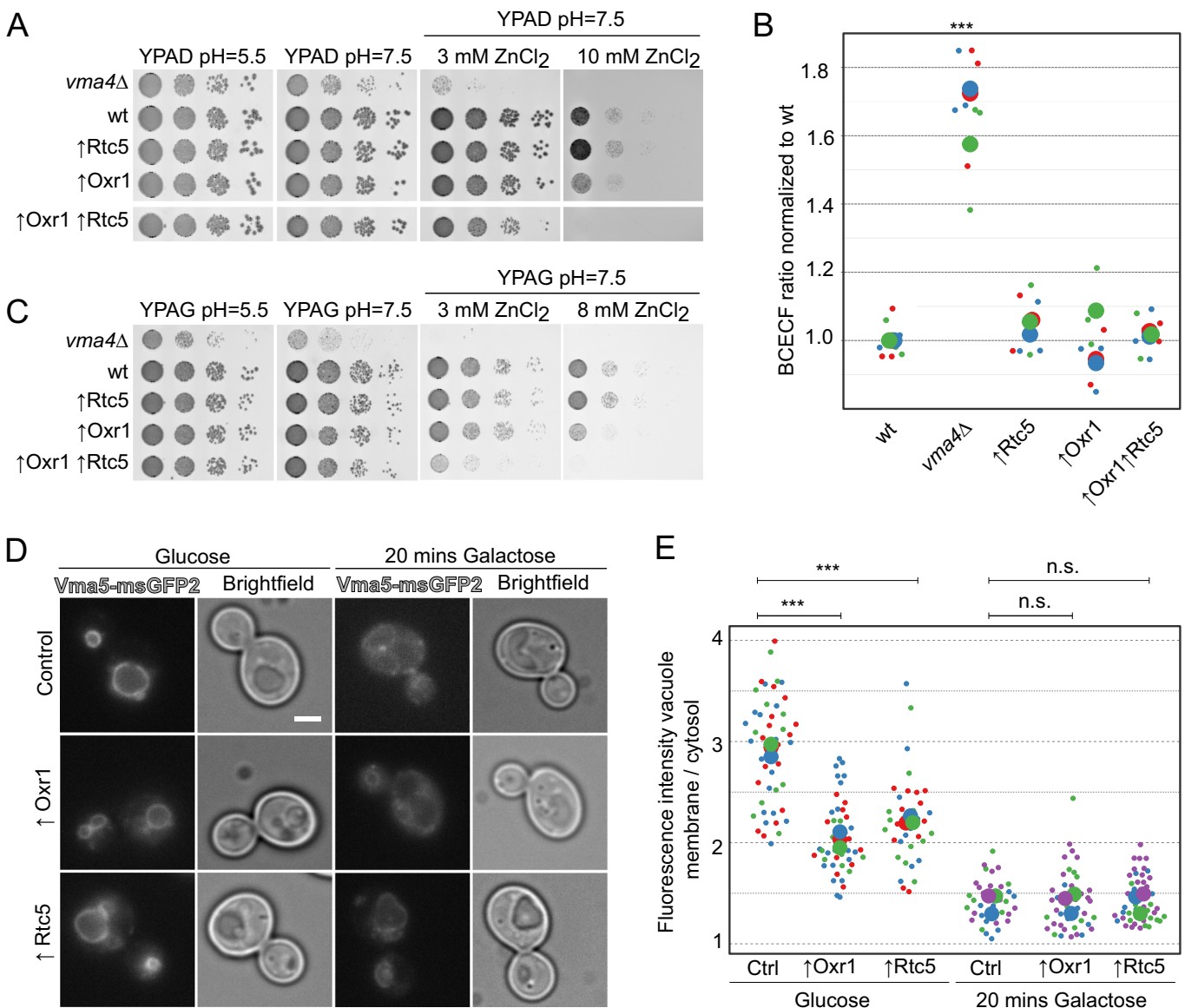

**Figure 6. Overexpression of Rtc5 or Oxr1 results in mild defects in vacuole acidification and less localization of Vma5 to the vacuole.**

(A) A wt strain, a strain lacking *VMA4*, or strains overexpressing Rtc5, Oxr1, or both under the control of the strong constitutive *TEF1* promoter were spotted as serial dilutions on glucose media of the indicated pH, with or without the addition of divalent cations as indicated in the figure. (B) Analysis of vacuolar acidity via BCECF staining in a wt strain, a strain lacking *VMA4*, and strains overexpressing either Rtc5, Oxr1, or both under the control of the strong constitutive *TEF1* promoter. The experiments were performed with cultures grown in a medium containing glucose and pH = 5.5. For each strain, at least three independent experiments were performed, each containing three biological replicates. For each sample, the fluorescence emission of BCECF at 538 nm was measured when excited at 440 or 485 nm, and a ratio between these two values was calculated. The ratio was normalized to the average value for the wt strain in that experiment. The different colors in the graph indicate independent experiments, the smaller dots are biological replicates, and the larger circles represent the averages of each independent experiment. Statistical analysis was performed with a one-way ANOVA and a Tukey post hoc test. The vma4Δ strain was significantly different from the wt strain (***P value <0.001), all other strains are not significantly different from the wt strain (P value >0.05). (C) A wt strain, a strain lacking *VMA4*, or strains overexpressing Rtc5, Oxr1, or both under the control of the strong constitutive *TEF1* promoter were spotted as serial dilutions on galactose media of the indicated pH, with or without the addition of divalent cations as indicated in the Figure. (D, E). Cells in which Subunit C (Vma5) was tagged with msGFP2 were grown in glucose-containing medium and imaged in this same medium or after shifting them for 20 min to a medium containing galactose as the sole carbon source. The distribution of Vma5 between the two compartments was quantified by using a ratio between the mean fluorescence intensity in a line along the vacuole membrane and the average of the mean fluorescence intensity in three circular regions in the cytosol. The different colors in the graph indicate independent experiments. The smaller circles represent individual cells, and the bigger circles represent the average for each independent experiment. Statistical comparison was performed using a one-way ANOVA and a Tukey post hoc test among the experimental means (***P value <0.001; n.s. not significant P value >0.05). The scale bar represents 2 µm. Source data are available online for this figure.

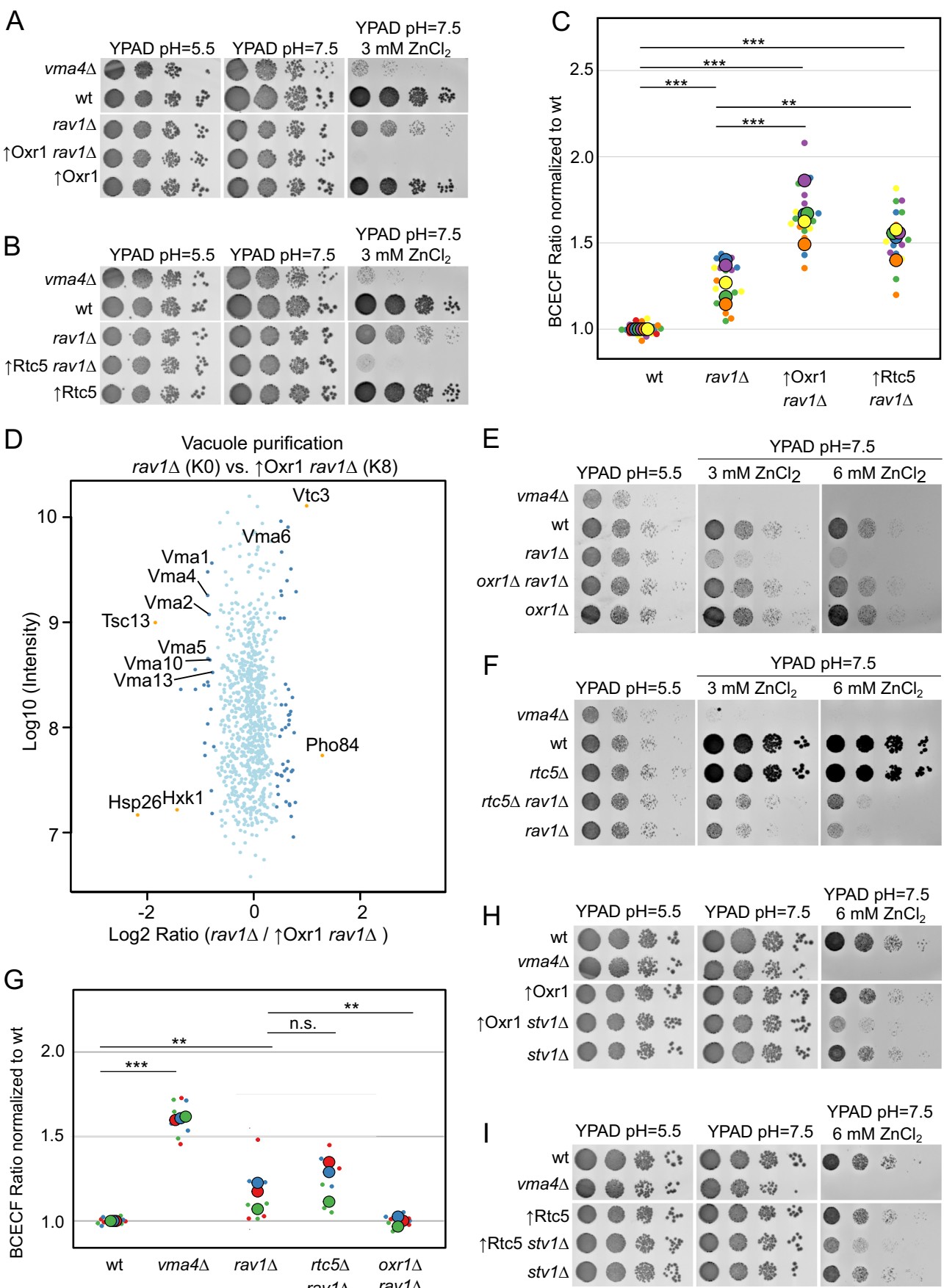

◄ **Figure 7. Rtc5 and Oxr1 counteract the function of the RAVE complex.**

(A, B) Isogenic strains with the indicated modifications in the genome were spotted as serial dilutions in media with pH = 5.5, 7.5, or 7.5 and containing 3 mM ZnCl$_2$. (C) Analysis of vacuolar acidity via BCECF staining in a wt strain, a strain lacking *RAV1*, and strains lacking *RAV1* and overexpressing either Rtc5 or Oxr1 under the control of the strong constitutive *TEF1* promoter. Five independent experiments were performed, each containing three biological replicates. For each experiment, the BCECF ratio was normalized to the average value of the wt strain in that experiment. The different colors in the graph indicate independent experiments, the smaller dots are biological replicates, and the larger circles represent the averages of each independent experiment. Statistical analysis was performed with a one-way ANOVA and a Tukey post hoc test (***P value <0.001; **P value <0.01; ***P value <0.001). (D) SILAC-based vacuole proteomics of *rav1Δ* compared to *rav1Δ* cells that overexpress Oxr1. Log10 of the detected protein intensities are plotted against Log2 of the heavy/light SILAC ratios. Significant outliers based on a two-group, two-tailed Student´s t-test are color-coded in red (*P* value <1e − 14), orange (*P* value <0.0001), or dark blue (*P* value <0.05); other identified proteins are shown in light blue. The same experiment but switching the heavy and light labeling of the strains is shown in Appendix Fig. S5A. (E, F) Isogenic strains with the indicated genomic modifications were spotted as serial dilutions on media with pH = 5.5, or pH = 7.5 with 3 mM and 6 mM ZnCl$_2$. (G) Analysis of vacuolar acidity via BCECF staining in a wt strain, a strain lacking *RAV1*, and strains lacking *RAV1* and lacking in addition either *RTC5* or *OXR1*. Three independent experiments were performed, each containing three biological replicates. For each experiment, the BCECF ratio was normalized to the average value of the wt strain in that experiment. The different colors in the graph indicate independent experiments, the smaller dots are biological replicates, and the larger circles represent the averages of each independent experiment. Statistical analysis was performed with a one-way ANOVA and a Tukey post hoc test (n.s not significant *P* value >0.05, **P value <0.01, ***P value <0.001). (H, I) Isogenic strains with the indicated genomic modifications were spotted as serial dilutions on media with pH = 5.5, pH = 7.5, or pH = 7.5 with the addition of 6 mM ZnCl$_2$. Source data are available online for this figure.

On the other hand, overexpression of either Rtc5 or Oxr1 results in a growth defect in a background lacking Rav1 in neutral media containing ZnCl$_2$, a negative genetic interaction (Fig. 7A,B). We found that Rtc5 tagged in the C-terminus, as used for experiments throughout this manuscript, is functional (Fig. EV3A,B). In contrast, Oxr1 tagged in the C-terminus with either msGFP2 or 2xmNeonGreen is not functional (Fig. EV3C,D). Both these constructs show a cytosolic localization (Fig. EV3E,F), but as these are not functional proteins, this result should be interpreted with caution. Oxr1 is annotated as localized to mitochondria, based on microscopy analysis of C-terminal tagged constructs (Elliott and Volkert, 2004). Since we now show that these constructs are likely not functional, this localization should be re-addressed.

## Oxr1 is required for the retention of Stv1 in pre-vacuolar compartments

In addition to the results discussed previously, the vacuolar proteomics experiments showed that vacuoles of cells lacking *OXR1* have increased amounts of Stv1 (Fig. 5D and Appendix Fig. S3A). Stv1 is one of the two yeast isoforms of subunit a of the V$_O$ domain, which localizes to endosomes and the Golgi complex (Manolson et al, 1994; Banerjee and Kane, 2017; Kawasaki-Nishi et al, 2001). We wondered if this effect was caused by mislocalization of Stv1 from the Golgi complex to the vacuole in this mutant, and addressed this by fluorescence microscopy. Indeed, in cells lacking *OXR1*, Stv1-mNeonGreen (Stv1-mNG) shows a clear localization to the vacuole membrane, in contrast to wt and *rtc5Δ* cells (Fig. 8A). Stv1-mNG is functional as assessed by the negative genetic interaction of Stv1 with Vph1 (Manolson et al, 1994) (Fig. EV3G). We quantified the re-localization of Stv1-mNG using Mander´s overlap coefficients with the vacuole membrane marker Pfa3-HaloTag and the late-Golgi/ trans-Golgi network marker Sec7-2xmKate2 (Smotrys et al, 2005; Hou et al, 2005; Franzusoff et al, 1991; Day et al, 2018). Both Overlap coefficients between Stv1 with Pfa3 increased significantly in cells lacking *OXR1* compared to a control strain, but were unaffected by the deletion of *RTC5* (Fig. 8B). In addition, deletion of *OXR1* produced a strong decrease in the fraction of Stv1 overlapping with Sec7, consistent with a fraction of the protein re-localizing to a different compartment, the vacuole (M2, Fig. 8C). The fraction of Sec7 overlapping with Stv1 did not diminish (M1, Fig. 8C), indicating that late-Golgi compartments are still positive for Stv1 in the absence of Oxr1.

However, the intensity of Stv1 in these compartments was decreased in cells lacking Oxr1, indicative of a smaller concentration of the protein (Fig. 8D). The levels of the protein in the whole cell, in contrast, were unaffected (Fig. 8E). This last result also shows that the re-localization of Stv1 to the vacuole is not due to higher expression levels, which have already been reported to result in Stv1 localization to the vacuole (Finnigan et al, 2012).

The steady-state localization of Stv1 is achieved by cycles of retrograde transport mediated by the Retromer complex (Finnigan et al, 2011). However, the effect we observe on Stv1 is specific, and not caused by the lack of a functional Retromer pathway, as other Retromer cargo proteins, like Vps10 or Tlg2, were not enriched in vacuoles of cells lacking *OXR1* in our vacuolar proteomics experiments shown before (Fig. 5D and Appendix Fig. S3A). The dots representing these proteins are labeled in Fig. EV4A,B. We confirmed this by addressing the subcellular localization of two Retromer cargo proteins, Vps10 and Kex2, in *oxr1Δ* cells and *vps26Δ* cells, lacking a functional Retromer. While both proteins are mislocalized to the vacuole in *vps26Δ* cells, the deletion of *OXR1* has no effect on their subcellular localization (Fig. EV4C,D).

The localization of Stv1 to the Golgi/Endosomes is mediated by the interaction of its cytosolic N-terminus with PI(4)P, and a truncated form of the protein involving only the first 452 amino acids localizes to this compartment (Banerjee and Kane, 2017). We thus wondered if the localization of this construct would also be affected by the lack of *OXR1*. Indeed, cells lacking Oxr1 mislocalized Stv1(1–452)-mNeonGreen to the vacuole membrane (Fig. 8F). This is observable by a significant increase of the overlap coefficients between Stv1(1–452) and Pfa3 in cells lacking *OXR1* (Fig. 8G). This result suggests that the mislocalization of Stv1 in cells lacking Oxr1 is mediated by interfering with the specificity of the interaction of the N-terminus of Stv1 with Golgi membranes.

We wondered if the re-localization of Stv1 to the vacuole could complement the lack of the vacuole-specific subunit a, Vph1. Indeed, cells lacking both *VPH1* and *OXR1* have a milder growth defect in high-pH media containing zinc or calcium as the single deletion of *VPH1* (Fig. 8H). This indicates that the re-localization of Stv1 to the vacuole in *oxr1Δ* cells is able to complement partially the lack of Vph1 on the vacuole, as was also observed for overexpression of Stv1 (Manolson et al, 1994). The deletion of *RTC5*, which does not cause re-localization of Stv1, had the opposite effect (Appendix Fig. S6A), and neither the deletion of *RTC5* or *OXR1* had a positive genetic interaction with the deletion of *STV1* (Appendix Fig. S6B,C).

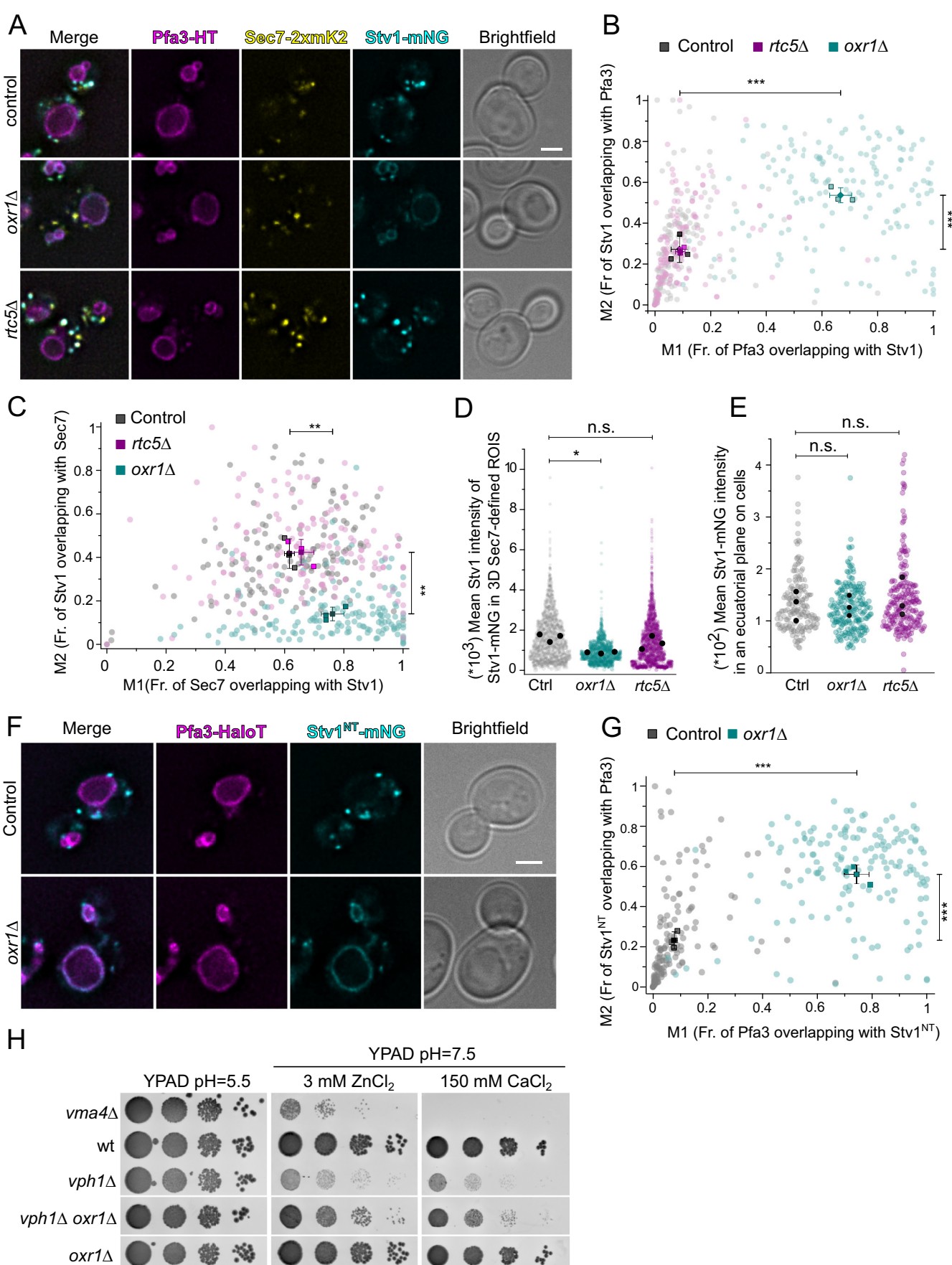

◄ **Figure 8. Oxr1 is required for the retention of Stv1 in pre-vacuolar compartments.**

(A–C) Fluorescence microscopy analysis of the subcellular localization of Stv1 in the absence of Oxr1 and Rtc5. Cells expressing C-terminally mNeonGreen tagged Stv1 (Stv1-mNG) in a control strain and in *OXR1* and *RTC5* deletion strains were imaged live by fluorescence microscopy. Sec7 was C-terminally tagged with 2xmKate2 (Sec7-2xmK2) as a late-Golgi marker, and Pfa3 was tagged with the HaloTag (Pfa3-HT) and stained with JFX650 as a vacuole membrane marker. Panel (A) shows representative images, and the scale bar represents 2 µm. Panels (B) and (C) show a co-localization analysis of Stv1-mNG with Pfa3-HT (B) or Sec7-2xmK2 (C) using Mander's coefficients M1 and M2 for the overlap of the two signals. Each circle represents a single cell, the squares represent the average of each of three independent experiments, and the diamonds the overall average with error bars representing standard deviation. The statistical comparison was performed by a one-way ANOVA among the means for each experiment, followed by a Tukey post hoc test. The comparisons shown in the graph are between control and *oxr1Δ* cells (***P value <0.001, **P value <0.01), the difference between wt and *rtc5Δ* cells was non-significant in all cases (P value >0.05). (D, E) Analysis of the intensity of Stv1-mNG signal in late-Golgi compartments (D) or in whole cells (E). Analysis of the intensity of Stv1-mNG in the same experiment shown in panel (A). The mean intensity of the Stv1-mNG signal was measured in regions of interest (ROIs) representing the whole cell in an equatorial plane or in 3D ROIs representing late-Golgi compartments defined as structures positive for Sec7-2xmK2 signals. Each colored circle represents a single cell, and each black circle represents the mean of each of the three independent experiments. Statistical comparisons were performed by a one-way ANOVA among the means for each experiment, followed by a Tukey post hoc test (*P value <0.05; n.s. not significant P value >0.05). (F, G) Fluorescence microscopy analysis of Stv1(1–452)-mNG localization in a control strain and in a strain lacking *OXR1*, together with Pfa3-HT as a vacuole membrane marker. Panel (F) shows representative images, with a scale bar representing 2 µm. Panel G shows co-localization analysis using Mander's M1 and M2 coefficients as described for panels (B) and (C). The statistical comparison was performed by a one-way ANOVA among the means for each experiment, followed by a Tukey post hoc test (***P value <0.001) (H) A wt strain, a *VMA4* deletion strain, *VPH1* and *OXR1* deletion strains and a *VPH1 OXR1* double deletion strain were spotted as serial dilutions on high-pH media with and without 3 mM $ZnCl_2$ and 150 mM $CaCl_2$. Source data are available online for this figure.

## Discussion

In this work, we have used XL-MS to characterize vacuolar protein complexes and protein–protein interactions. Our dataset was able to recapitulate the interactions among subunits of protein complexes with high fidelity to the known structures, as well as other known interactions, including proteins described as regulators of other proteins, formation of SNARE complexes, and enzyme–substrate pairs. This suggests that the generated dataset is an important resource to study the protein–protein interactions of vacuolar proteins, including many cross-links that could represent novel interactions. Among these, we focused on the identification of the uncharacterized protein Rtc5 as a novel interactor of the V-ATPase.

Our experiments addressed the role of Oxr1 and Rtc5 with respect to the V-ATPase in vivo and showed that both proteins promote the disassembly of the complex. In the presence of the RAVE complex, overexpressing the proteins only produced a minor acidification defect, detectable in conditions that strongly rely on V-ATPase function, like growth in neutral media in the presence of a high concentration of zinc. However, vacuole proteomics allowed us to observe more assembled complexes in the absence of these proteins. This shows that in the wt strain during growth in the exponential phase in media containing glucose, the V-ATPase complex is not fully assembled. These findings reinforce previous observations that the assembly of the V-ATPase is not an all-or-nothing phenomenon, but rather it is able to sample a range of states (Parra and Kane, 1998). In addition, this indicates that Oxr1 and Rtc5 are part of a disassembly mechanism that takes place under standard growth conditions, and not only under specific stresses, like oxidative stress. In the absence of the RAVE complex, the V-ATPases of the vacuole are unstable and only loosely associated (Smardon and Kane, 2007), and under these conditions, the overexpression of either TLDc protein was enough to produce a further disassembly of the complex and a strong growth defect in neutral pH and presence of metals.

An intriguing observation is that Rtc5 is able to interact with the assembled $V_O$-$V_1$ V-ATPase and localizes to the vacuole membrane in vivo through this interaction, while Oxr1 binds to the $V_1$ domain in a way that inhibits the assembly of the holocomplex (Fig. 9) (Khan et al, 2022). Mammalian mEAK7 (Tldc1), on the other hand,

also binds the assembled complex (Tan et al, 2022). Both mEAK7 and Rtc5 are modified with myristic acid in the N-terminus, and contain an additional N-terminal EF hand-like domain, thus they might represent a subfamily of TLDc-domain-containing proteins with a different mode of binding to the complex than Oxr1. Despite this difference, our data shows that both Rtc5 and Oxr1 promote an in vivo state of lower V-ATPase assembly, but the effects caused by Oxr1 were always stronger. A plausible explanation is that Rtc5 is always bound to the V-ATPase, but is regulated in vivo by a yet unknown mechanism to access a state where it promotes disassembly under specific conditions. Therefore, Rtc5 overexpression has milder phenotypes on V-ATPase disassembly because it needs to overcome this regulatory switch (Fig. 9). Interestingly, it has been shown that there is a fraction of V-ATPase complexes that do not disassemble (Parra and Kane, 1998). This suggests that there could be sub-populations of the complex with differential regulations, for instance, complexes bound and not-bound to Rtc5 (Fig. 9). Given the crucial role of the V-ATPases in many processes and the many signals that determine assembly state (glucose, pH, and osmotic shock), it is not a surprise that different mechanisms may converge regulating both assembly and disassembly to achieve optimal V-ATPase activity levels.

Finally, our experiments showed that cells lacking Oxr1 re-localize the Golgi-specific V-ATPase subunit Stv1 to the vacuole (Fig. 9). This unexpected result, which was revealed by vacuole proteomics, highlights the advantage of taking comprehensive approaches rather than targeted analysis of proteins that are expected to be affected under a specific condition. The subcellular localization of Stv1 to the Golgi complex is mediated by active recycling from late endosomes by the retromer complex (Kawasaki-Nishi et al, 2001) as well as direct interaction of its N-terminal domain with the marker phosphoinositide of this compartment, PI(4)P (Banerjee and Kane, 2017). Both acute inactivation of the Golgi-localized PI(4) Kinase, Pik1, or mutation of the involved region in Stv1, result in re-localization of Stv1 to the vacuole (Banerjee and Kane, 2017; Finnigan et al, 2011). We now uncover Oxr1 as an additional factor required for the retention of Stv1 in the Golgi complex. Other retromer cargo proteins were not affected by the deletion of *OXR1*, indicating that the effect of Oxr1 is independent of this pathway. Interestingly, the localization of the truncated N-terminal domain of Stv1 to the Golgi complex also requires Oxr1. Since this truncated form cannot be assembled into the

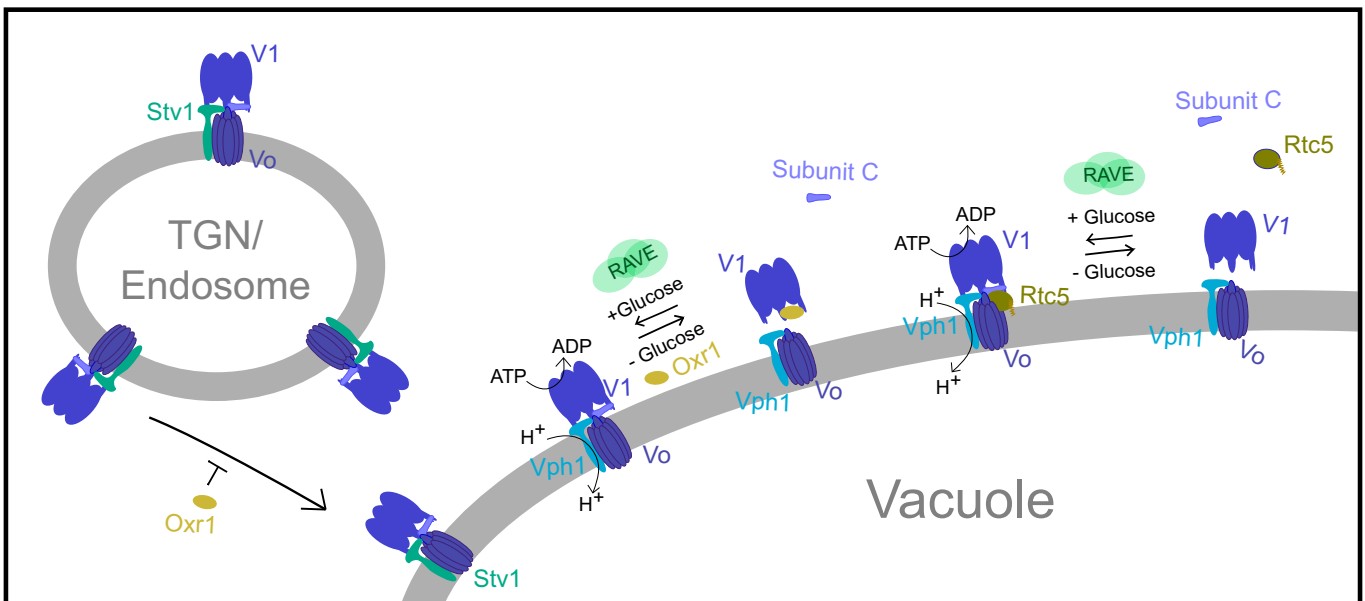

**Figure 9.  Diagram summarizing the in vivo localization and function of Rtc5 and Oxr1 with respect to the V-ATPase.**

Rtc5 localizes to the vacuole membrane based on its N-terminal myristoylation and interaction with the assembled V-ATPase complex. Both Oxr1 and Rtc5 favor the disassembly of the V-ATPase, counteracting the role of the RAVE complex. Finally, Oxr1 is necessary for the retention of Stv1-containing V-ATPases in the late-Golgi or endosomal compartments.

V-ATPase, this suggests that Oxr1 affects the localization of this domain in particular and not of Stv1-containing V-ATPase complexes. An interesting possibility that remains to be addressed is whether Oxr1 affects the PI4P content either in the Golgi or on the vacuoles, and affects Stv1 localization in this way. A plausible scenario would be that the subcellular localization of Stv1 is regulated in vivo, as a way to fine-tune the pH homeostasis of Golgi/Endosome compartments, and that such a mechanism is controlled by Oxr1.

Accumulating evidence supports that TLDc domains are V-ATPase interacting modules. Recently all mammalian proteins containing this domain were shown to be interactors of the V-ATPase (Eaton et al, 2021; Merkulova et al, 2015; Tan et al, 2022), and we now confirm the role of yeast Oxr1 on the V-ATPase and show a role for the only other yeast TLDc domain-containing protein for the first time (Khan et al, 2022). This suggests that V-ATPase binding is the evolutionarily conserved role of the TLDc domain. For the yeast proteins, we show a role in V-ATPase disassembly for both of them, confirming the previous in vitro experiments with Oxr1 (Khan et al, 2022). Whether this activity is conserved in all TLDc proteins of other organisms remains to be seen. A compelling hypothesis is that the diversification of the family through evolution resulted in proteins that connect the V-ATPase, as a machinery with a central role in cellular homeostasis, with different proteins or cellular processes. In this respect, other previously reported functions, such as the role of EAK7 in lifespan determination in *Caenorhabditis elegans* and of mEAK7 in TORC1 signaling need to be revisited to understand how they relate to V-ATPase binding (Nguyen et al, 2018; Alam et al, 2010). In particular, the role of these proteins in oxidative stress protection is also conserved, and future research should address the connection between these two aspects of this protein family.

## Methods

### Yeast strains

*Saccharomyces cerevisiae* strains were based on either BY4741 or SEY6210. Genetic manipulations were done via homologous recombination of cassettes amplified via PCR, as described in (Janke et al, 2004). The genotypes of all yeast strains used in this study are listed in Table EV1. The Rtc5(G2A) mutant was generated in the genome using CRISPR-Cas9 as described in (Generoso et al, 2016). For this, 500 ng of the plasmid pAGM 164 and the oligonucleotide oAGM 347 were transformed in a BY4741 strain. Plasmid pAGM 164 was generated by PCR amplification of the plasmid pCU5003 using the primers oAGM 345 and oAGM 346. The plasmid was confirmed by sequencing it. The selected *S. cerevisiae* transformant was confirmed by making a PCR of the relevant genomic region with primers oAGM 034 and oAGM 035 and sequencing it. Plasmids and oligonucleotides used in this study are listed in Table EV2.

### Growth tests on solid media

For seriated dilution growth assays, the wild-type or isogenic mutant yeast cells were grown to saturation in liquid YPAD (Yeast extract 2%, peptone 1%, glucose 2%, adenine 40 mg/l) pH 5.5. Cultures were diluted 1:100 and grown for 5–6 h until the exponential phase. Cultures were adjusted to an $OD_{600}$ of 0.25, and seriated 1:10 dilutions were performed. The different dilutions were spotted on solid (2% agar) YPAD or YPAG (same composition, 2% galactose instead of glucose) media of the indicated pH, with or without the addition of $ZnCl_2$ or $CaCl_2$ at

different concentrations. Plates were incubated at 30 °C for 1 to 3 days and imaged using a Bio-Rad ChemiDoc MP imaging System. The figures were assembled with the conditions that best showed differences among the tested strains. All the growth experiments shown were performed at least 3 times, and one representative example is shown.

## Crosslinking mass spectrometry of isolated vacuoles

### Vacuole isolation and cross-linking

Vacuoles were isolated as described in (Haas, 1995) from 12 l of culture. Buffers were modified by using 10 mM HEPES/KOH pH = 7.4 instead of 10 mM PIPES/KOH pH = 6.8. The whole amount of obtained vacuoles was combined, and protease inhibitors were added (1 mM phenylmethylsulfonyl fluoride (PMSF), 0.1 mg/ml leupeptin, 0.5 mg/ml pepstatin A, and 0.1 mM Pefabloc). 1 mM of the cross-linker Azide-A-DSBSO was added, and the samples were incubated with end-over-end rotation for 20 min at room temperature, followed by the addition of 20 mM Tris/HCl and 30 min incubation at room temperature to quench the reaction.

### Digestion

Cross-linked vacuoles were digested in solution. Proteins were denatured by incubation in 8 M urea in 50 mM tetraethylammonium bromide (TEAB), reduced with 5 mM dithiothreitol (DTT) for 60 min at 37 °C and alkylated with 40 mM chloroacetamide at room temperature for 30 min in the dark. Proteins were digested with Lysyl endopeptidase C (Wako) at an enzyme-to-protein ratio of 1:75 (w/w) at 37 °C for 4 h. After diluting with 50 mM TEAB to a final concentration of 2 M urea, the digestion was continued with trypsin (Serva) at an enzyme-to-protein ratio of 1:100 (w/w) at 37 °C overnight. Peptides were desalted with Sep-Pak C18 cartridges (Waters) and dried in a SpeedVac.

### Enrichment for cross-linked peptides

The digested cross-linked vacuolar peptides were enriched on dibenzocyclooctyne (DBCO) coupled sepharose beads (Click Chemistry). The peptides were resuspended in PBS to a final concentration of 1 mg/ml, and then the prewashed beads were added for incubation overnight at room temperature. The bead and peptide ratio was 10 µl beads (20 µl slurry) per 0.6 mg peptide. After incubation, the beads were washed once with water, and then incubated with 0.5% SDS at 37 °C for 15 min. The beads were washed sequentially three times with each of the following solutions: 0.5% SDS, 8 M urea in 50 mM TEAB bromide, 10% acetonitrile, and finally, twice with water. The washing volume was 500 µl. The cross-linked peptides were eluted with 100 µl 10% trifluoroacetic acid at 25 °C for 2 h and dried in a SpeedVac.

### High-pH reverse-phase (HPH) fractionation

The enriched peptides were fractionated by high-pH chromatography using a Gemini C18 column (Phenomenex) on an Agilent 1260 Infinity II system. A 90 min gradient was applied and 24 fractions were collected, dried under a speed vacuum, and subjected to LC/MS analysis.

### LC-MS analysis

LC-MS analysis of cross-linked and HPH-fractionated peptides was performed using an UltiMate 3000 RSLC nano-LC system coupled online to an Orbitrap Fusion Lumos mass spectrometer (Thermo Fisher Scientific). Reversed-phase separation was performed using a 50 cm analytical column (in-house packed with Poroshell 120 EC-C18, 2.7 µm, Agilent Technologies) with a 180 min gradient. Cross-link acquisition was performed using an LC-MS2 method. The following parameters were applied: MS resolution 120,000; MS2 resolution 60,000; charge state 4–8 enable for MS2; stepped HCD energy 19, 25, 32 with FAIMS voltages set to −50, −60, and −75.

### Data analysis

Raw data were converted into .mgf file in Proteome Discoverer (version 2.4). Data analysis was performed using XlinkX standalone (Liu et al, 2017) with the following parameters: minimum peptide length = 6; maximal peptide length = 35; missed cleavages = 3; fix modification: Cys carbamidomethyl = 57.021 Da; variable modification: Met oxidation = 15.995 Da; Azide-A-DSBSO cross-linker = 308.0038 Da (short arm = 54.0106 Da, long arm = 236.0177 Da); precursor mass tolerance = 10 ppm; fragment mass tolerance = 20 ppm. MS2 spectra were searched against the UniProt yeast database. Results were reported at 2% FDR at a unique lysine-lysine connection level in Dataset EV1. The raw data were publicly available through the PRIDE repository (https://www.ebi.ac.uk/pride/; accession code PXD046792).

### Structural mapping

The following structures were used for the structural mapping: PI3K complex (5DFZ), EGO complex (6JWP), V-ATPase complex (7FDA), AP-3 complex (7P3Y), TORC1 complex (7PQH), VTC complex (7YTJ), HOPS complex (7ZU0), and SEA complex (8ADL). Cross-links were mapped onto these selected structures using Chimera X 1.3 (Goddard et al, 2018; Pettersen et al, 2021). Cα-Cα distances were determined using the measuring function in Pymol v.2.5.2 (Schrodinger LLC), and a list of the distances measured for all crosslinks is provided in Dataset EV2.

### Data visualization

A protein–protein interaction (PPI) network was constructed with the Cytoscape software (version 3.8.2), and the visualization of residue-to-residue connections was performed by the plugin software XlinkCyNET (Lima et al, 2021)

## Structural modeling

Structure modeling was performed by the HADDOCK web portal (https://wenmr.science.uu.nl/haddock2.4/) following its tutorial. Briefly, for the V-ATPase complex, the multiple chains with overlapping numbering was renumbered by R clean.pdb package, and then the predicted structure of Rtc5 and PDB structure of the V-ATPase complex were uploaded to the web. The cross-linked residues were selected as active residues directly involved in the interaction, and the cross-links data were uploaded as unambiguous restraints to the web. Then the output prediction models were used for structural mapping.

## Fluorescence microscopy and image analysis

Cells were grown to logarithmic phase in yeast extract peptone medium containing glucose (YPAD). The vacuolar membrane was stained by the addition of 30 µM FM4-64 dye (Thermo Fisher

Scientific) for 30 min at 30 °C with shaking, followed by washing and incubation without dye for 30 min under the same conditions. The lumen of the vacuoles was stained by the addition of 20 μM 7-Amino-4-Chlormethylcumarin (CMAC, Invitrogen) for 15 min at 30 °C with shaking, followed by two washing steps. Strains containing proteins tagged with the HaloTag were incubated with 2 μM of JFX650-HaloTag Ligand (kindly provided by the Lavis laboratory in Janelia Research Campus) for 15 min at 30 °C with shaking, followed by eight washes with 1 mL synthetic medium supplemented with essential amino acids (SDC). For experiments involving a shift to galactose media, the cells were harvested by centrifugation, washed twice with YPAG media incubated in YPAG media for 20 min, and imaged in SGC (synthetic media containing galactose).

Cells were imaged live in SDC or SGC with an Olympus IX-71 inverted microscope equipped with 100x NA 1.49 and 60x NA 1.40 objectives, a sCMOS camera (PCO, Kelheim, Germany), an InsightSSI illumination system, 4',6-diamidino-2-phenylindole, GFP, mCherry, and Cy5 filters, and SoftWoRx software (Applied Precision, Issaquah, WA). Z-stacks were used with 200, 250, or 350 nm spacing for constrained-iterative deconvolution with the SoftWoRX software. Image processing and quantification was done with ImageJ (Schneider et al, 2012). One plane of the z-stack is shown in the figures.

For the analysis of the co-localization of Stv1-mNeonGreen and Stv1(1–452)-mNeonGreen with vacuoles (Pfa3-HaloTag) and late-Golgi/TGN (Sec7-2xmKate) the images were processed as follows. In each channel, we performed background subtraction using the background subtraction function of Image J, and each channel was thresholded with the thresholding function of Image J and the Otsu algorithm. Regions of interest were generated for each cell, using the YeastMate plugin (Bunk et al, 2022). The thresholded images were used to calculate Manders M1 and M2 coefficient between Stv1-mNeonGreen and Sec7-2xmKate or Pfa3-HaloTag for each cell, using the JaCoP Plugin (Bolte and Cordelières, 2006), BIOP version. The graphs show the M1 and M2 coefficients for each cell, and the averages for each of the three experiments, as well as an overall average. Statistical comparisons were made using the averages for independent experiments.

For the analysis of the distribution of Vma5-msGFP2 between the vacuole membrane and the cytosol, cells were analyzed in an equatorial plane. A line profile along the vacuole membrane was drawn using the FM4-64 channel, and the mean fluorescence intensity in the GFP channel was measured. In addition, the mean fluorescence intensity was measured in three circular regions in the cytosol and averaged. A ratio between the fluorescence in the vacuole and in the cytosol was calculated. The experiment was performed at least three times for each condition (glucose and 20 min galactose), and at least ten cells were analyzed for every strain in every experiment. The statistical analysis was performed using the experimental means for each strain, using a one-way ANOVA to compare the strains within each growth condition.

## Metabolic labeling with an azido-myristate, click chemistry reaction with alkyne-biotin, and pulldown of labeled proteins

Cells were grown in YPAD media to the logarithmic phase. 10 OD$_{600}$ units of cells were harvested twice for each strain, and cells were resuspended with 20 ml fresh media. About 25 μM myristic acid-azide (Thermo Fisher Scientific; labeled) was added to

one of the two samples, and dimethyl sulfoxide (DMSO; unlabeled) was added to the other. Cells were incubated in the dark at 30 °C with agitation for 4 h. 10 OD$_{600}$ units of cells were harvested for each condition and resuspended with 300 μl buffer (50 mM Tris pH = 8, 1% TX-100, 1 mM PMSF, 0.1 mg/ml leupeptin, 0.5 mg/ml pepstatin A, 0.1 mM Pefabloc). 200 μl of acid-washed glass beads were added, and cells were mechanically lysed twice in a FastPrep device (6 m/sec for 40 s; MP Biomedicals), with a 5-min incubation on ice in between. The cell lysate was transferred to a new tube, and the beads were washed with an additional 600 μl buffer, which was combined with the lysate. The cell lysate was centrifuged at 20,000 × g for 20 min and 4 °C. 25 μl slurry of GFP-Trap beads (Chromotek) were equilibrated with buffer and the lysate was added to the beads. The samples were incubated for 30 min at 4 °C. The beads were centrifuged for 1 min at 300 × g and the supernatant (flow-through) was taken of the beads. The beads were washed twice with 1 ml buffer (centrifugation 1 min, 300 × g and 4 °C). The proteins were eluted by the addition of 25 μl Tris pH = 8, 2% SDS, and boiling of the sample at 95 °C for 5 min. About 25 μl of the flow-through were added to the eluate as a carrier protein for the protein precipitations; this sample was used for the click-chemistry reaction as follows.

The 50 μl of sample were combined with 100 μl Click-iT reaction buffer (Thermo Fisher Scientific) containing the azide detection reagent alkyne-biotin and 10 μl water. Copper (II) Sulfate, Click-iT reaction buffer additive 1, and Click-iT reaction buffer additive 2 from the Click-iT™ Protein Reaction Buffer Kit (Thermo Fisher Scientific) were added to the sample according to the manufacturer's instructions. The sample was incubated with end-over-end rotation for 20 min at 4 °C. The protein in the sample was precipitated using chloroform:methanol. This involved the subsequent addition of 600 μl Methanol, 150 μl Chloroform, mixing of sample, and addition of 400 μl water, with mixing by vortexing after every addition. The sample was centrifuged for 5 min at 20,000 × g and 4 °C. The upper phase was removed and 450 μl of Methanol was added, followed by centrifugation for 5 min at 20,000 × g and 4 °C. The supernatant was discarded, and the pellet was washed with 450 μl Methanol. The sample was centrifuged for 5 min at 20,000 × g and 4 °C, and the pellet was air-dried until completely dry. The pellet was resuspended in 30 μl Buffer B (4% SDS, 50 mM TrisHCl pH 7.4, 5 mM EDTA) with agitation at 37 °C, followed by the addition of 70 μl Buffer C (50 mM TrisHCl pH 7.4, 5 mM EDTA, 150 mM NaCl, 0.2% Triton X-100). After mixing of sample, 10 μl were removed as the input sample. 1400 μl of Buffer C was added to the sample, and combined with 40 μl of slurry high-capacity streptavidin beads (Thermo Fisher Scientific; previously washed three times with Buffer C). The sample and beads were incubated for 1 h at room temperature while rotating end-over-end. The beads were washed four times with Buffer C and centrifuged at 1500 × g for 30 s and 4 °C. The proteins were removed from the beads by the addition of 40 μl of sample buffer with 100 mM DTT and heating at 95 °C for 10 min. The samples were analyzed via SDS-PAGE and Western Blot.

## Subcellular fractionation

Cells were grown to the logarithmic phase in YPAD media. 50 OD$_{600}$ units of cells were harvested and resuspended in 500 μl lysis buffer (PBS, 0.5 mM PMSF, 0.1 mg/ml leupeptin, 0.5 mg/ml pepstatin A, 0.1 mM Pefabloc). 300 μl acid-washed glass beads were added, and

cells were mechanically lysed twice with a FastPrep device (6 m/s for 40 s; MP Biomedicals) with a 5-min incubation on ice in between. The cell lysate was centrifuged for 5 min at $400 \times g$ and 4 °C, followed by transfer of the supernatant to a new tube. The protein concentration of each lysate was measured by the Bradford method (Bio-Rad) and the lysates were diluted to the same concentration. This lysate corresponds to the input sample. The lysate was centrifuged at $20,000 \times g$ and 4 °C for 20 min. The supernatant was removed, and the pellet was resuspended in an equivalent amount of buffer. The input, supernatant and pellet samples were mixed with 4x sample buffer and heated up at 95 °C for 10 min and analyzed via SDS-PAGE and Western blot, loading equivalent amounts.

## SDS-PAGE and Western blot

Proteins were separated using SDS-PAGE in 10% Bis-Tris acrylamide/bisacrylamide gels and transferred to a nitrocellulose membrane (GE Healthcare). The membranes were blocked for 30 min with PBS 5% milk and incubated with the first antibody for 1 h at room temperature or at 4 °C overnight with gentle shaking. The membranes were washed three times with PBS for 5 min, and once with TBS-Tween (0.5% (v/v) Tween 20), followed by incubation with a 1:20,000 dilution of a fluorescent-dye-coupled secondary antibody (Thermo Fisher Scientific) for 1 h at room temperature. Antibodies were diluted in PBS 5% milk. For the detection of the fluorescent signal, a LiCor Odyssey infrared fluorescence scanner was used. The antibodies used are listed in Table EV3.

## SILAC-based GFP-Trap pull downs

Lysine auxotrophic strains were grown in yeast nitrogen base (YNB) medium containing either 30 mg/ml L-lysine or 30 mg/ml $^{13}$C6;$^{15}$N2- L-lysine (Cambridge Isotope Laboratories, USA). Cells were grown to logarithmic phase and up to 350 OD$_{600}$ units were harvested of labeled cells. Pellets were divided into two cryo vials and 750 µl of lysis buffer (20 mM HEPES pH 7.4, 150 mM KoAC, 1 mM MgCl2, 5% Glycerol, 1% GDN, 1 mM PMSF, protease inhibitor cocktail FY (Serva) 1/100) were added. 500 µl acid-washed glass beads were added and cells were mechanically lysed twice in a FastPrep device (6 m/s for 40 s; MP Biomedicals), with a 5-min incubation on ice in between. The lysate was transferred to a new tube and the glass beads were washed with an additional 250 µl of buffer, which was combined with the lysate. The lysate was centrifuged at $12,000 \times g$ for 20 min and 4 °C. The protein concentration of the supernatant was measured in quadruplicates using the Bradford method (Bio-Rad). 12.5 µl slurry GFP-Trap beads (Chromotek) were equilibrated with lysis buffer and the same amount of protein from each lysate was added to the beads, and filled up to 1500 µl with buffer. Samples were incubated at 4 °C for 15 min, rotating end-over-end. The beads were centrifuged at $300 \times g$ for 1 min and washed twice with 1 ml lysis buffer and four times with 1 ml buffer without detergent. The samples were processed for mass spectrometry with the iST 96x Kit (Preomics) according to manufacturer instructions, using LysC as the protease.

## Vacuole and whole-cell proteomics

Vacuole proteomic analysis was done as described in (Eising et al, 2019). Briefly, lysine auxotrophic strains were grown in yeast

nitrogen base (YNB) medium containing either 30 mg/ml L-lysine or 30 mg/ml $^{13}$C6;$^{15}$N2- L-lysine (Cambridge Isotope Laboratories, USA). Cells were grown to logarithmic phase and the same amount of OD units of the control strain and the mutant strain were harvested and combined. Vacuoles were isolated as described in (Haas, 1995) and protein concentration was measured by the Bradford method (Bio-Rad). Samples were diluted to 200 µg/ml in 1 ml with 0% Ficoll (10 mM PIPES/KOH pH = 6.8, 0.2 M Sorbitol). If the protein concentration was lower, the whole amount of harvested vacuoles was used. 250 µl of cold 100% Trichloroacetic acid was added to the samples, and they were incubated for 10 min at 4 °C. The samples were centrifuged for 5 min at 14,000 rpm, 4 °C. The pellet was washed twice with 200 µl ice-cold acetone and then dried at 55 °C until completely dry. Samples were processed for mass spectrometry with the iST 96x Kit (Preomics) according to manufacturer instructions, using LysC as the protease.

For whole-cell proteomics, lysine auxotrophic strains were grown in yeast nitrogen base (YNB) medium containing either 30 mg/ml L-lysine or 30 mg/ml $^{13}$C6;$^{15}$N2- L-Lysine (Cambridge Isotope Laboratories, USA). Cells were grown to logarithmic phase and 1 OD unit of each strain was harvested. The cells were centrifuged at 4000 rpm for 5 min at 4 °C, followed by washing of cells with 500 µl ice-cold water. The cells were centrifuged at 4000 rpm for 5 min at 4 °C, and the samples were processed for mass spectrometry with the iST 96x Kit (Preomics) according to manufacturer instructions, using LysC as the protease.

## Mass spectrometry analysis of vacuole and whole cell proteomics, and GFP-Trap pull downs

### Mass spectrometry analysis

For mass spectrometry analysis, reversed-phase chromatography was performed on a Thermo Ultimate 3000 RSLCnano system connected to a Q ExactivePlus mass spectrometer (Thermo Fisher Scientific) through a nano-electrospray ion source. For peptide separation, 50 cm PepMap C18 easy spray columns (Thermo Fisher Scientific) with an inner diameter of 75 µm were used and kept at a temperature of 40 °C. The peptides were eluted from the column with a linear gradient of acetonitrile from 10 to 35% in 0.1% formic acid for 118 min at a constant flow rate of 300 nl/min, followed by direct electrospraying into the mass spectrometer. The mass spectra were acquired on the Q-Exactive Plus in a data-dependent mode to automatically switch between full scan MS and up to ten data-dependent MS/MS scans. The maximum injection time for full scans was 50 ms, with a target value of 3,000,000 at a resolution of 70,000 at m/z = 200. The ten most intense multiply charged ions (z = 2) from the survey scan were selected with an isolation width of 1.6 Th and fragments with higher energy collision dissociation with normalized collision energies of 27 (Olsen et al, 2007). Target values for MS/MS were set at 100,000 with a maximum injection time of 80 ms at a resolution of 17,500 at m/z = 200. To avoid repetitive sequencing, the dynamic exclusion of sequenced peptides was set at 30 s.

### Data processing

The resulting MS and MS/MS spectra were analyzed using MaxQuant (version 1.6.0.13, https://www.maxquant.org/) utilizing the integrated ANDROMEDA search algorithms (Cox and Mann, 2008; Cox et al, 2011). The peak lists were compared against local

databases for *S. cerevisiae* (obtained from the *Saccharomyces Genome* database, Stanford University), with common contaminants added. The search included carbamidomethylation of cysteine as a fixed modification and methionine oxidation, N-terminal acetylation, and phosphorylation as variable modifications. The maximum allowed mass deviation was 6 ppm for MS peaks and 20 ppm for MS/MS peaks. The maximum number of missed cleavages was two. The false discovery rate was 0.01 on both the peptide and the protein level. The minimum required peptide length was six residues. Proteins with at least two peptides (one of them unique) were considered identified. The re-quant option of MaxQuant was disabled. The calculations and plots were performed with the R software package (https://www.r-project.org/). All resulting protein group files are included in Dataset EV3.

### BCECF staining

Triplicate cultures of every strain were grown to logarithmic phase in a synthetic medium, supplemented with essential amino acids (SDC). 3 $OD_{600}$ units of cells were harvested and resuspended in 30 μl of fresh medium containing 50 μM BCECF, AM (2', 7'-Bis-(2-Carboxyethyl)-5-(and-6)-Carboxyfluorescein, Acetoxymethyl-Ester; Thermo Fisher Scientific), followed by an incubation at 30 °C for 30 min with shaking. The cells were washed twice. The fluorescence signal was measured using a SpectraMax iD3 plate reader (Molecular Devices, LLC). The emission at 538 nm was measured with excitation at 440 and 485 nm, and this was used to calculate the ratio between them. For every experiment, the ratio was normalized to the average of the 3 wt samples.

## Data availability

The raw data for the vacuole cross-linking mass spectrometry experiment is publicly available through the PRIDE repository (https://www.ebi.ac.uk/pride/; accession code PXD046792): https://www.ebi.ac.uk/pride/archive?keyword=PXD046792. The protein group files for all vacuole proteomics, whole cell proteomics and pulldown experiments are compiled in Dataset EV3.

## Peer review information

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

## Acknowledgements

We are thankful to Christian Ungermann for providing yeast strains and for the critical reading of the manuscript. We thank Stefan Walter for his technical assistance with Mass spectrometry experiments. Fluorescence microscopy experiments were carried out in the iBiOS facility of Osnabrück University. Mass spectrometry experiments not involving cross-linking were carried out in the Mass Spectrometry facility of the Biology School of Osnabrück University. The JFX650-HaloTag Ligand was kindly provided by the Lavis Laboratory of the HHMI Janelia Research Campus.

## Author contributions

**Samira Klössel**: Formal analysis; Investigation; Visualization; Methodology; Writing—review and editing. **Ying Zhu**: Data curation; Investigation; Visualization; Methodology; Writing—review and editing. **Lucia Amado**: Investigation; Methodology; Writing—review and editing. **Daniel D Bisinski**: Investigation; Methodology; Writing—review and editing. **Julia Ruta**: Investigation; Methodology; Writing—review and editing. **Fan Liu**: Conceptualization; Supervision; Funding acquisition; Methodology; Project administration; Writing—review and editing. **Ayelén González Montoro**: Conceptualization; Supervision; Funding acquisition; Visualization; Methodology; Writing—original draft; Project administration; Writing—review and editing.

## Funding

This work was funded by a DFG individual collaborative grant to AGM and FL (GO 3313-1-1 and LI 3260/5-1). JR is supported by the European Research Council (ERC) Starting Grant (ERC-STG No. 949184). Open Access funding enabled and organized by Projekt DEAL.

## Disclosure and competing interests statement

The authors declare no competing interests.

# Expanded View Figures

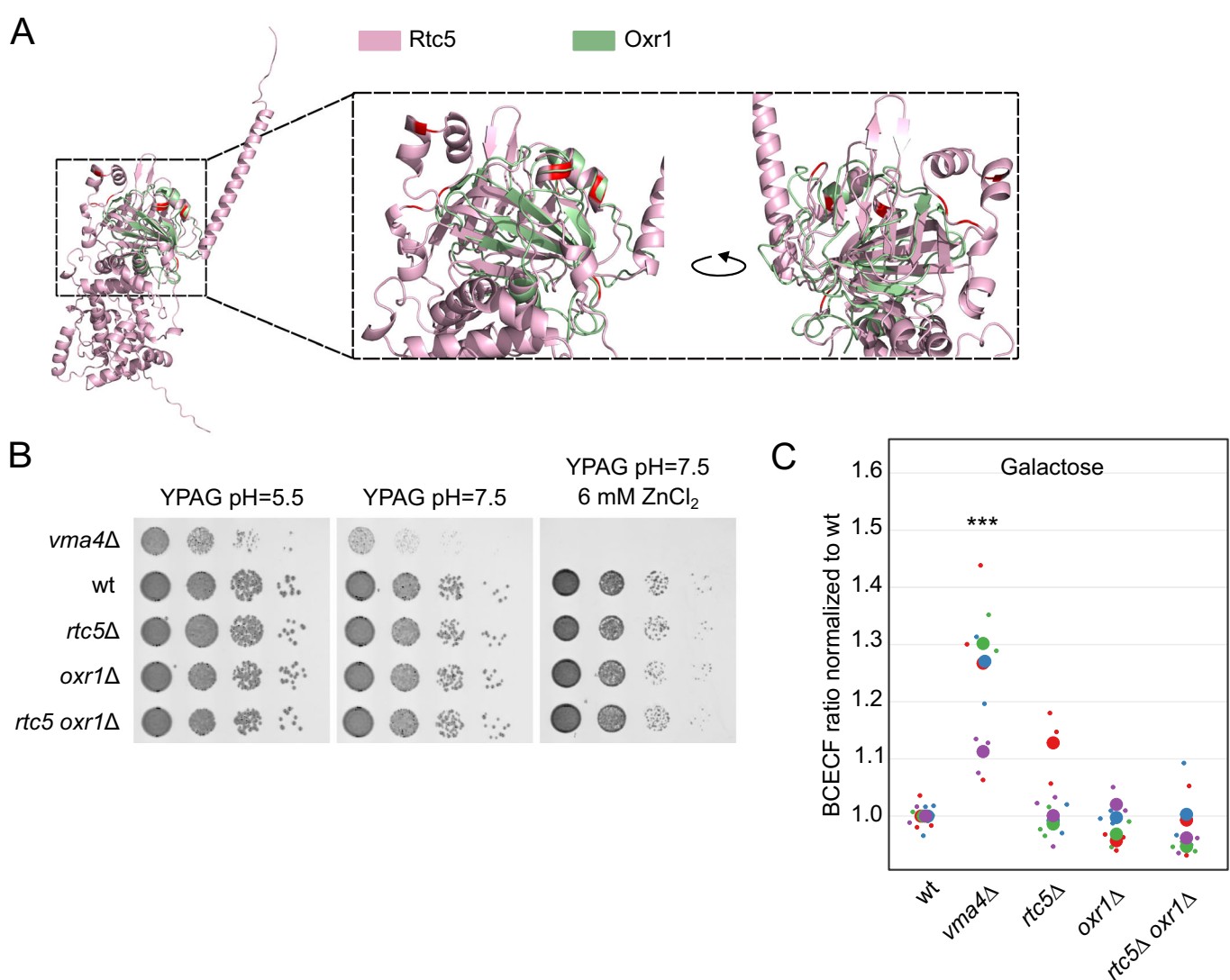

**Figure EV1.** **Alphafold model of Rtc5 and growth phenotypes of *rtc5Δ* and *oxr1Δ* strains in medium containing galactose as the carbon source.**

(A) Comparison of the AlphaFold model generated for Rtc5 with the available structure of Oxr1, the only two TLDc domain-containing proteins of *Saccharomyces cerevisiae*. (B) A wt strain or strains lacking *VMA4*, *OXR1* or *RTC5*, or both *OXR1* and *RTC5* were spotted as serial dilutions on media containing galactose as the carbon source with pH = 5.5, pH = 7.5, or pH = 7.5 and 6 mM $ZnCl_2$. (C) Analysis of vacuolar acidity via BCECF staining in a wt strain, a strain lacking *VMA4*, *OXR1*, *RTC5*, or both *RTC5* and *OXR1*. The experiments were performed with cultures grown in a medium containing galactose and pH = 5.5. For each strain, at least three independent experiments were performed, each containing three biological replicates. For each sample, the fluorescence emission of BCECF at 538 nm was measured when excited at 440 or 485 nm, and a ratio between these two values was calculated. The ratio was normalized to the average value for the wt strain in that experiment. The different colors in the graph indicate independent experiments, the smaller dots are biological replicates and the larger circles represent the averages of each independent experiment. Statistical analysis was performed with a one-way ANOVA and a Tukey post hoc test. The *vma4Δ* strain was significantly different from the wt strain (***$P$ value <0.001), all other strains are not significantly different from the wt strain ($P$ value >0.05).

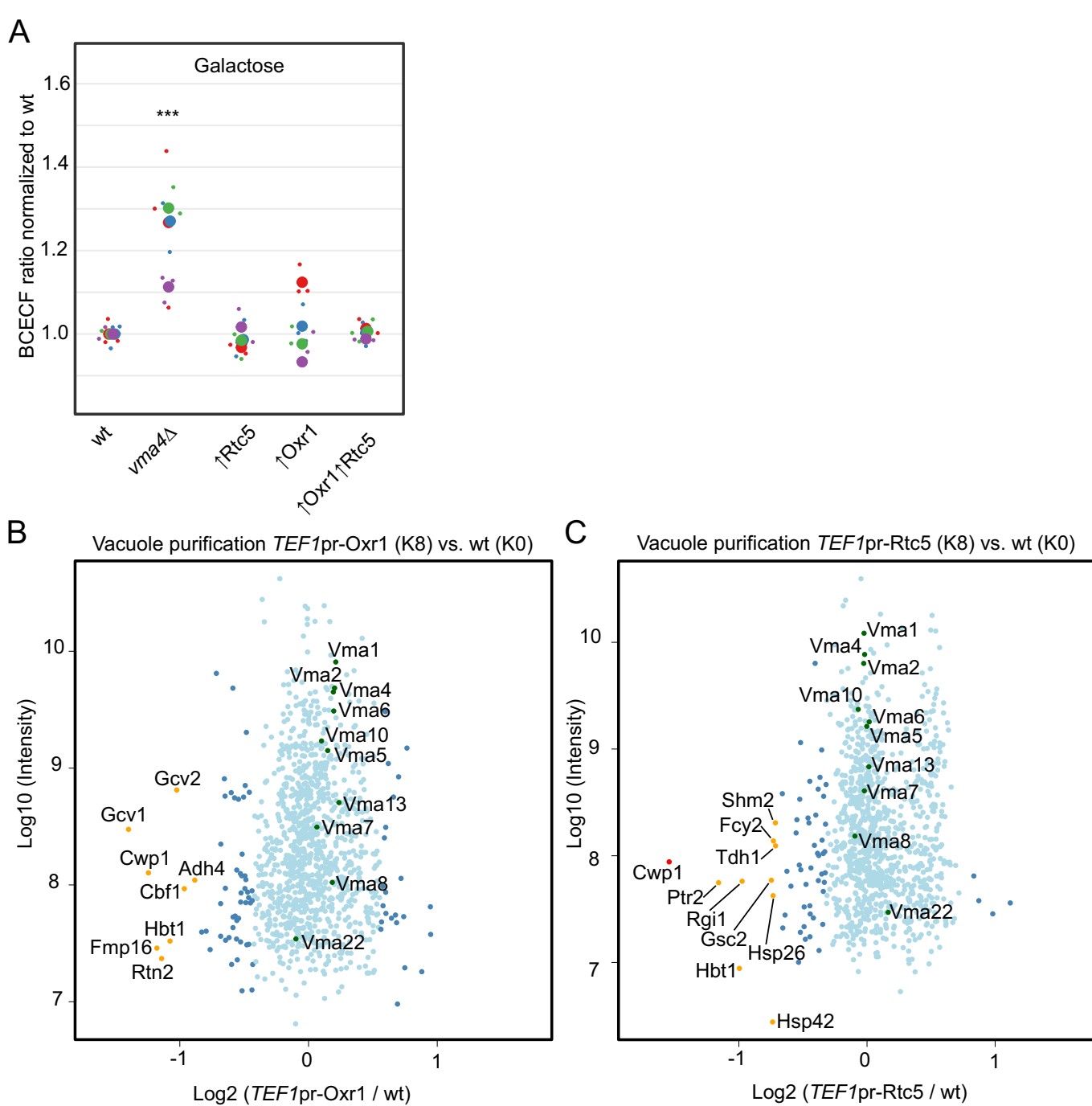

**Figure EV2. Vacuolar acidity when grown in galactose and vacuolar proteomics of strains overexpressing Rtc5 or Oxr1.**

(A) Analysis of vacuolar acidity via BCECF staining in a wt strain, a strain lacking *VMA4*, overexpressing Oxr1, Rtc5, or both Rtc5 and Oxr1. The experiments were performed with cultures grown in a medium containing galactose and pH = 5.5. Four independent experiments were performed, each containing three biological replicates. For each sample, the fluorescence emission of BCECF at 538 nm was measured when excited at 440 or 485 nm, and a ratio between these two values was calculated. The ratio was normalized to the average value for the wt strain in that experiment. The different colors in the graph indicate independent experiments, the smaller dots are biological replicates, and the larger circles represent the averages of each independent experiment. Statistical analysis was performed with a one-way ANOVA and a Tukey post hoc test. The *vma4Δ* strain was significantly different from the wt strain (***$P$ value <0.001), all other strains are not significantly different from the wt strain ($P$ value >0.05). (B, C) SILAC-based vacuole proteomics of cells overexpressing either Oxr1 (B) or Rtc5 (C) compared with the wt strain. Log10 of the detected protein intensities are plotted against Log2 of the heavy/light SILAC ratios. Significant outliers are color-coded in red ($P$ <1e − 14), orange ($P$ < 0.0001), or dark blue ($P$ < 0.05); other identified proteins are shown in light blue. V-ATPase subunits are labeled and shown as green dots. Statistical comparison is based on a two-group, two-tailed Student´s *t*-test. In panel C the range chosen for the X-axis excludes the dot representing Rtc5, so that the individual dots are clearly visible. This protein showed a Log2 (normalized H/L ratio) of 4.109945 and a Log10 (intensity) 9.433689846.

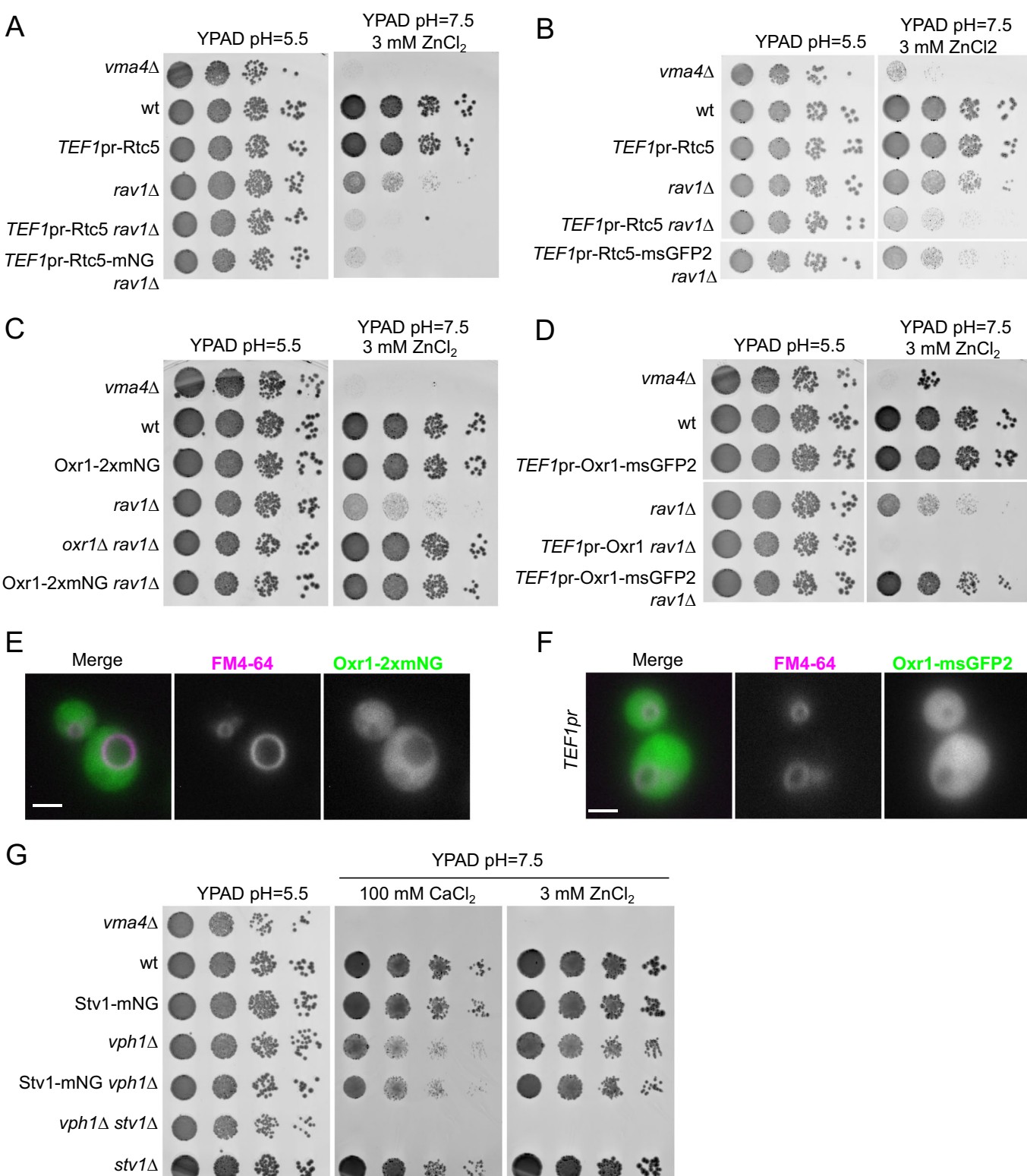

◀ **Figure EV3. C-terminally tagged Rtc5 and Stv1 are functional, C-terminally tagged Oxr1 is not.**

(A, B) Rct5-mNeonGreen and Rtc5-msGFP2 are functional. Serial dilutions of strains with the indicated genotypes were spotted on YPAD media pH = 5.5 or YPAD media pH = 7.5 containing 3 mM $ZnCl_2$. (C) Oxr1-2xmNeonGreen is not functional. Serial dilutions of strains with the indicated genotypes were spotted on YPAD media pH = 5.5 or YPAD media pH = 7.5 containing 3 mM $ZnCl_2$. (D) Oxr1-msGFP2 is not functional. Serial dilutions of strains with the indicated genotypes were spotted on YPAD media pH = 5.5 or YPAD media pH = 7.5 containing 3 mM $ZnCl_2$. (E) Oxr1-2xmNeonGreen shows a cytosolic localization. Fluorescence microscopy images of cells expressing Oxr1-2xmNeonGreen (2xmNG) and endocytosed FM4-64 as a vacuolar marker. The scale bar represents 2 μm. (F) Overexpressed Oxr1-msGFP2 shows a cytosolic localization. Fluorescence microscopy images of cells expressing Oxr1-msGFP2 under the control of the strong constitutive *TEF1* promoter and endocytosed FM4-64 as a vacuolar marker. The scale bar represents 2 μm. (G) Stv1-mNeonGreen is functional. Strains with the indicated genotypes were spotted as seriated dilutions in YPAD medium pH=5.5 and YPAD medium pH = 7.5 containing either 100 mM $CaCl_2$ or 3 mM $ZnCl_2$.

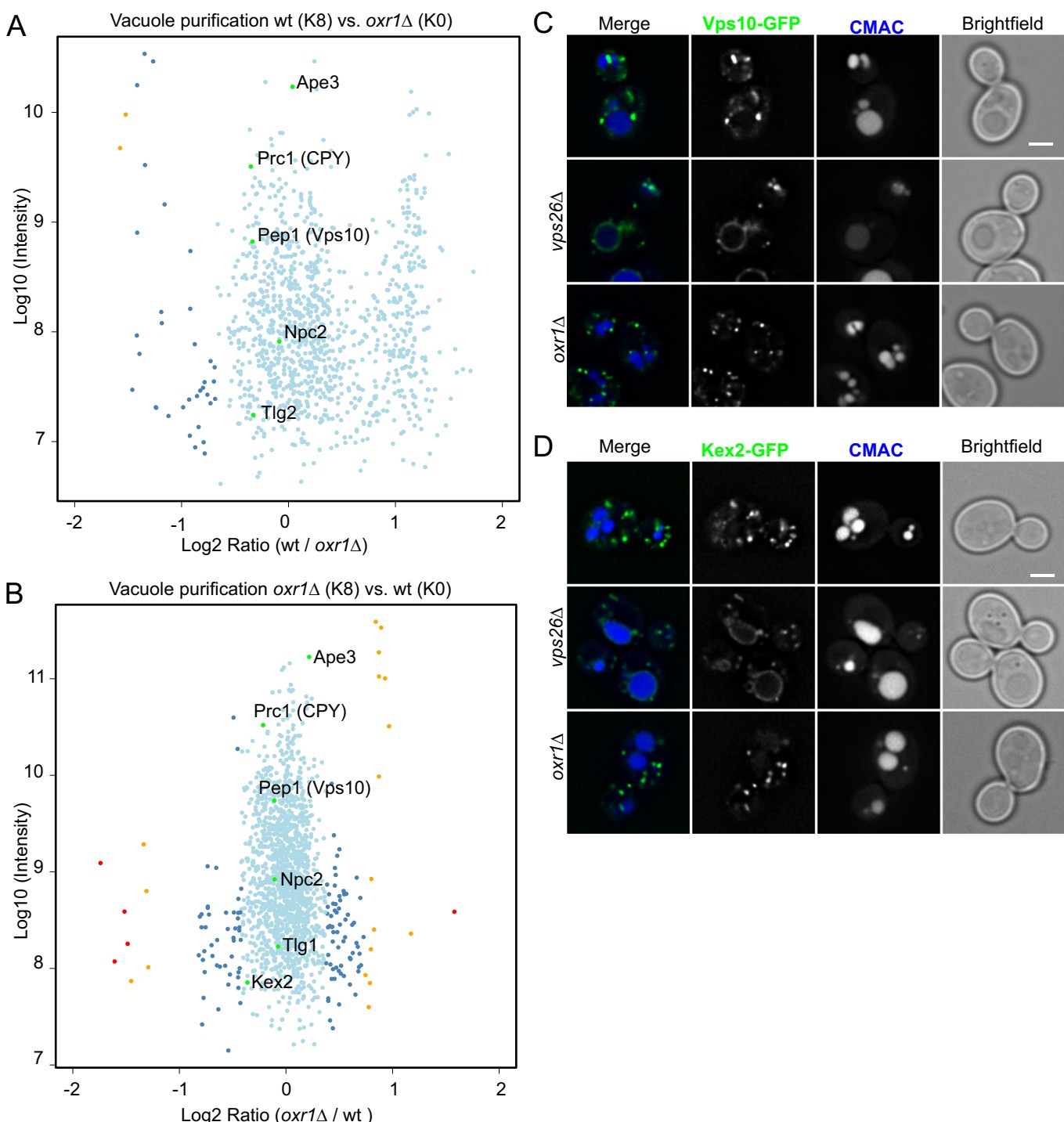

**Figure EV4. The localization of cargo proteins of the Retromer pathway is not affected by deletion of OXR1.**

(A, B) The abundance of Retromer cargo proteins in the vacuole is not affected by the deletion of *OXR1*. The experiment in (A) is the same experiment as in Appendix Fig. S3A and the experiment in panel (B) is the same experiment as the one in Fig. 5D. SILAC-based vacuole proteomics of cells lacking *OXR1* compared with the wt strain. Log10 of the detected protein intensities are plotted against Log2 of the heavy/light SILAC ratios. Significant outliers are color-coded in red (*P* value <1e − 14), orange (*P* value <0.0001), or dark blue (*P* value <0.05); other identified proteins are shown in light blue. Statistical comparison is based on a two-group, two-tailed Student´s *t*-test. Retromer cargo proteins were labeled and the dots are shown in green. (C, D) Retromer cargo proteins do not re-localize to the vacuole in strains lacking *OXR1*. Fluorescence microscopy analysis of Vps10-GFP or Kex2-GFP and vacuole lumen stained with CMAC, in wt cells, cells lacking the Retromer complex subunit *VPS26* or strains lacking *OXR1*. The scale bar represents 2 μm.

