## [Peer Review File · The EMBO Journal]

Yeast TLDC domain proteins regulate the assembly state and subcellular localization of the V-ATPase

Samira Klössel, Ying Zhu, Lucia Amado, Daniel Bisinski, Julia Ruta, Fan Liu, and Ayelén González Montoro

Corresponding author: Ayelén González Montoro (ayelen.gonzalez.montoro@uos.de)

Review Timeline:

Transferred from Review Commons:	18th Jan 24
Editorial Decision:	24th Feb 24
Revision Received:	6th Mar 24
Accepted:	8th Mar 24

Editor: Kelly Anderson

Transaction Report:

This manuscript was transferred to The EMBO Journal following peer review at Review Commons.

Review #1

1. Evidence, reproducibility and clarity:

Evidence, reproducibility and clarity (Required)

Klössel et al. explore the role of the TLDC domain-containing proteins Oxr1p and Rtc5p in *Saccharomyces cerevisiae*. They performed cross-linking mass spectrometry and detected the interaction of Rtc5p with V-ATPase. TLDC domains have previously been found to serve as V-ATPase interacting domains. The authors find that both Oxr1p and Rtc5p induce dissociation of V-ATPase in vivo, an activity that was previously established for Oxr1p in vitro. They propose that this activity counteracts the activity of the V-ATPase assembling RAVE complex. They also find that Oxr1p is necessary for late Golgi retention of the Golgi form of the V-ATPase (i.e. containing the Stv1p isoform of subunit a). It is a little surprising that Oxr1p binding to V-ATPase was not detected by the cross-linking mass spectrometry, although the authors argue that this absence may be owing to the abundance of the proteins, which sounds reasonable.

Suggestions:

1. The authors observed that knockout of Rtc5p or Oxr1p does not affect vacuolar pH. If Rtc5p and Oxr1p both cooperate to dissociate V-ATPase, the authors may wish to characterize the effect of a $\Delta Rtc5p\Delta Oxr1p$ double knockout on vacuolar pH.
2. The manuscript would benefit from a well-labelled diagram showing the subunits of V-ATPase (e.g. in Figure 2D).
3. The images of structures, especially in Figure 1-Supplement 1B, are not particularly clear and could be improved (e.g. by removing shadows or using transparency).
4. The authors should clearly describe the differences between Rtc5p and Oxr1p in terms of protein length, sequence identity, domain structure, etc.

Minor:

1. The "O" in VO should be capitalized.
2. In Figure 4 supplement 1, the labels "I", "S", and "P" should be defined.
3. Please clarify what is meant by "switched labelling"
4. The meaning of the sentence "Indeed, this was the case for both of them" is ambiguous.
5. For Figure 1-Supplement 1B it is hard to see the crosslink distances.
6. The statement "The effects of Oxr1 are greater than those caused by Rtc5" requires more context. Is there a way of quantifying this effect for the reader?
7. The phrase "negative genetic interaction" should be clarified.
8. In the sentence "Isogenic strains with the indicated modifications in the genome were spotted as serial dilutions in media with pH=5.5, pH=7.5 or pH=7.5 and containing 3 mM ZnCl₂", "where" should be "were".
9. Figure 2D: the authors should consider re-coloring these models, as it is challenging to distinguish Rtc5p from the V-ATPase.

2. Significance:

Significance (Required)

The vacuolar protein interaction map alone from this manuscript is a nice contribution to the literature. Experiments establishing colocalization of Rtc5p to the vacuole are convincing, as is dependence of this association on the presence of assembled V-ATPase. Similarly, experiments related to myristoylation are convincing. The observed mislocalization of V-ATPases that contain Stv1p to the vacuole (which is also known to occur when Vph1p has been knocked out) upon knockout of Oxr1p is also extremely interesting.

Overall, this is an interesting manuscript that contributes to our understand of TLDC proteins.

3. How much time do you estimate the authors will need to complete the suggested revisions:

Estimated time to Complete Revisions (Required)

(Decision Recommendation)

Less than 1 month

Yes

Review #2

1. Evidence, reproducibility and clarity:

Evidence, reproducibility and clarity (Required)

Using cross-linking proteomics, Klössel et al. identify the yeast TLDC domain protein Rtc5 as a novel interactor of yeast V-ATPase and characterize functions for Rtc5 and the TLDC domain protein Oxr1 in V-ATPase assembly and localization.

****Major points:****

1. The evidence of Oxr1 and Rtc5 as V-ATPase disassembly factors is circumstantial. The authors base their interpretation primarily on increased V1 (but not Vo) at purified vacuoles from Oxr1- or Rtc5-deleted strains, which does not directly address disassembly. Of course, the results regarding Oxr1 confirm detailed disassembly experiments with the purified protein complex (PMID 34918374), but on their own are open to other interpretations, e.g. suppression of V-ATPase assembly. Of note, the authors emphasize that they provide first evidence of the *in vivo* role of Oxr1, but monitor V1 recruitment with purified vacuoles and do not follow V-ATPase assembly in intact cells.
2. Oxr1 and Rtc5 have very low sequence similarity. It would be helpful if the authors provided more detail on the predicted structure of the putative TLDC domain of Rtc5 and its relationship to the V-ATPase - Oxr1 structure. Is Rtc5 more closely related to established TLDC domain proteins in other organisms?
3. The authors conclude vacuolar recruitment of Rtc5 depends on the assembled V-ATPase, based on deletion of different V1 and Vo domain subunits. However, these genetic manipulations likely cause a strong perturbation of vacuolar acidification; indeed, the images show drastically altered vacuolar morphology. To strengthen their conclusion, it would be helpful to show that Rtc5 recruitment is not blocked by inhibition of vacuolar acidification, and that conversely it is blocked by deletion of *rav1*.

2. Significance:

Significance (Required)

This is an interesting paper that confirms and extends previous findings on TLDC domain proteins as a novel class of proteins that interact with and regulate the V-ATPase in eukaryotes. The title seems to exaggerate the findings a bit, as the authors do not investigate V-ATPase (dis)assembly directly and only phenotypically describe altered subcellular localization of the Golgi V-ATPase in Oxr1-deleted cells. A recent structural and biochemical characterization of Oxr1 as a V-ATPase disassembly factor (PMID 34918374) somewhat limits the novelty of the results, but the function of Oxr1 in regulating subcellular V-ATPase localization and the identification of a second potential TLDC domain protein in yeast provide relevant insights into V-ATPase regulation. This paper will be of interest to cell biologists and biochemists working on lysosomal biology, organelle proteomics and V-ATPase regulation.

3. How much time do you estimate the authors will need to complete the suggested revisions:

Estimated time to Complete Revisions (Required)

(Decision Recommendation)

Between 1 and 3 months

No

Review #3

1. Evidence, reproducibility and clarity:

Evidence, reproducibility and clarity (Required)

****Summary****

In this manuscript, the authors used a proteomics approach to comprehensively study yeast vacuole protein-protein interactions using cross-linking mass-spectrometry (XL-MS). They identified 16694 interactions between 2051 proteins. Many known vacuolar protein complexes were found and used as positive controls, confirming the high quality of the dataset, however, no negative controls were reported, and this issue is raised in the 'Major comments' section. The authors then focused on one particular previously unknown protein-protein interaction between the TLDC-domain containing protein of unknown function Rtc5 and the vacuolar-type proton ATPase, V-ATPase, which acidifies yeast vacuoles. The methods and results regarding Rtc5 discovery as a novel interactor of the V-ATPase, Rtc5 myristoylation, and its V-ATPase-dependent localization to the vacuole membrane are convincing. The authors then moved on to study the in vivo function of Rtc5 as well as Oxr1, the only other TLDC-domain-containing protein in yeast. Interestingly, they did not originally detect Oxr1 in their protein-protein interaction studies, apparently due to its very low abundance in yeast. However, they found that deletion of either RTC5 or OXR1 in vivo resulted in more assembled V-ATPase at the yeast vacuole and this effect was stronger in *oxr1*Δ cells. However, RTC5/OXR1 deletion or

overexpression in parental yeast strains did not affect either vacuolar pH (a readout of functional V-ATPase) or yeast growth, including growth under specific conditions (neutral pH, in the presence of high concentrations of calcium or zinc), which is used to reveal a conditional lethal phenotype of unfunctional V-ATPase (the Vma⁻ phenotype). Since they did not observe any in vivo phenotype in parental yeast strains, they subsequently studied the effects of RTC5/OXR1 deletion and overexpression in the 'sensitized' rav1Δ strain, lacking a specific assembly factor of V-ATPase, Rav1, one of the subunits of RAVE complex. In this strain, RTC5/OXR1 overexpression resulted in less acidic vacuolar pH and reduced growth of double mutant cells, compared to the single rav1Δ mutant. In addition, overexpression of Oxr1, but not Rtc5, caused disassembly of the V-ATPase in rav1Δ cells, noteworthy this effect was not detectable in the parent strain with intact Rav1p.

Finally, they found that in oxr1Δ cells there is more Stv1 in the vacuole and concluded that Oxr1 is necessary for the retention of Stv1 containing V-ATPase at the vacuole. However, the mechanism seems to be complicated and remains to be elucidated.

In summary, an impressive variety of methods from a technologically advanced XL-MS to classical yeast growth assays were used to identify Rtc5 interaction with V-ATPase and analyze its functional role in vivo in yeast, making the conclusions well justified overall.

****Major comments****

Re: A cross-linking mass spectrometry map of vacuolar protein interactions (results)

While XL-MS is a very powerful method, it is a high-throughput approach and there should be some kind of negative control in these experiments. In cross-linking experiments, non-cross-linked samples are usually used as negative controls. What was the negative control in cross-linking mass-spectrometry experiments here? If there was no negative control, how the specificity of interactions was evaluated? Maybe the authors analyzed the dataset for highly improbable interactions and found very few of them? In addition, the high purity of vacuole preparation is critical. How was it assessed by the authors? All this is important to know to use this dataset as a reliable resource in the future.

Re: Rtc5 and Oxr1 counteract the function of the RAVE complex (results)

Taken together, data, presented in this section of the manuscript, provide strong evidence that Rtc5 and Oxr1 negatively regulate V-ATPase activity, counteracting the V-ATPase assembly, facilitated by the activity of the RAVE complex. However, the complete deletion of the major RAVE subunit Rav1p was required to observe this effect in vivo in yeast. The other way to induce V-ATPase disassembly in yeast is glucose deprivation. It will be interesting to study if there is a synergistic effect between glucose deprivation and RTC5/OXR1 deletion on V-ATPase assembly, vacuolar pH, and growth of single oxr1Δ, rtc5Δ or double oxr1Δrtc5Δ mutants (OPTIONAL). Glucose deprivation is a more physiologically relevant condition than a deletion of an entire gene.

Re: Figure 6 - supplement 1. The title is relevant to panel D only, it should be renamed to reflect the results of the disassembly of V-ATPase in rav1Δ mutant strains, while results about the

stv1Δ-based strains (Panel D) should be shown together with similar experiments in Figure 7 - supplement 2 for clarity.

Re: Figure 7 - supplement 1, Panel A. The proper assay to show that Stv1-mNeonGreen is functional is to express it in double mutant vph1Δstv1Δ to see if the growth defect is reversed. In addition, the vph1Δ growth defect is not changed (improved or worsened) in the presence of Stv1-mNeonGreen, so it means that the expression of Stv1-mNeonGreen does not further compromise the V-ATPase function, but it does not mean that it improves its function.

Re: Figure 7 - supplement 2. This figure should be combined with Fig. 6- suppl 1, panel D as also mentioned above. The figure seems to lack some labels, and conclusions are not accurate as discussed below. However, this data provides important additional information about relationships between isoform-specific subunits of V-ATPase Vph1 and Stv1 and both Rtc5 and Oxr1 and should be repeated if it is not done yet to have a better idea about these relationships. Panel B: Based on this picture, deletion of RTC5 has a negative genetic interaction with the deletion of VPH1, since double deletion mutant vph1Δ rtc5Δ grows worse than each individual mutant. Although it also means that there is no positive interaction, it is not the same. Panel C: Same as for panel B. Based on this picture, the deletion of OXR1 has a weak negative genetic interaction with the deletion of STV1, since double deletion mutant stv1Δ oxr1Δ grows worse than each individual mutant at 6 mM ZnCl₂. In addition, there is no label for the media in the middle panel, is it just YPAD pH=7.5, without the addition of any metals? Why there is no growth assay in the presence of CaCl₂, like in panels A and B? Panel D: Same as for panels B and C. Based on this picture, deletion of RTC5 has a negative genetic interaction with the deletion of STV1, since double deletion mutant stv1Δ rtc5Δ grows worse than each individual mutant at 6 mM ZnCl₂. There is no label in the middle panel (growth conditions) and no growth assay data in the presence of CaCl₂.

Re: Figure 7 - supplement 2, continued. How many times all these experiments were repeated? These experiments should be repeated at least 3 times, which is especially necessary for the experiments in panel C, because the effects are borderline. If results are reproducible and statistically significant, although small, the conclusion should be changed from "no positive genetic interactions" to "negative genetic interactions", which is more precise and informative. However, these results will be then in contradiction with the results from Figure 6 - Supplement 1, panel D, showing negative genetic interaction between the overexpression of Rtc5 or Oxr1 and deletion of Stv1, since both deletion and overexpression of Rtc5 or Oxr1 would have negative genetic interactions with Stv1. In addition, apparently, there is no data about genetic interaction between the overexpression of Rtc5 or Oxr1 and the deletion of Vph1. All this needs clarification, therefore repeating these experiments is essential. In conclusion, while genetic interactions between RTC5/OXR1 and RAV1 are straightforward, they seem to be more complex with STV1/VPH1.

Re: Methods. There is no description of yeast serial dilution growth assay at all. In addition, why the specific media (neutral pH, in the presence of high concentrations of calcium or zinc) was used is not explained either in the results or methods. Appropriate references should be included, for example, PMID: 2139726, PMID: 1491236.

****Minor comments****

Yeast proteins are named with "p" at the end, such as "Rtc5p".

Re: Introduction. In the introduction it should be indicated that Rtc5 was originally discovered as a "restriction of telomere capping 5", using screening of temperature-sensitive *cdc13-1* mutants combined with the yeast gene deletion collection [PMID: 18845848]. A couple of sentences should be written about the RAVE complex and its role in V-ATPase assembly.

Re: The TLDC domain-containing protein of unknown function Rtc5 is a novel interactor of the vacuolar V-ATPase (results)

1) It is important to understand, that Oxr1 was co-purified before with the V1 domain of V-ATPase from a certain mutant strain, not wild-type yeast [PMID: 34918374]. It may explain why the authors did not identify it in their original protein-protein interactions screen here.

2) It is a wrong conclusion that because Rtc5 was co-purified with both V1 and V0 domain subunits it interacts with the assembled V-ATPase, this does not exclude a possibility that Rtc5 also interacts with separate V1 sector or separate V0 sector of V-ATPase.

Re: Figure 1, Panel C. Is it possible to show individual proteins in different colors for clarity? Panel D. How were cross-link distances measured? It is not obvious if you are not an expert in the field and it is not described in the methods.

Re: Figure 1 - Supplement 1, Panel A. What scientific information are we getting from this picture? Panel B. Why are these complexes shown separately from the complexes in Figure 1, panel C? Also, can individual proteins be colored differently here as well?

Re: Figure 3. It will be nice to show the localization of the untagged protein as well if antibodies are available (OPTIONAL).

Re: Figure 4. Why different tags were used in panels A (GFP), C (msGFP2) and D (mNeonGreen)? Panels B and C. Were Rtc5 fusions detected using anti-GFP antibodies? The authors should have full-size Western blots available, not just cut-out bands, as some journals and reviewers require them for publication.

Re: Figure 4 - Supplement 1, Panel A. Does "-" and "+" mean +/- Azido-Myr? Panel B. There is no blot with a membrane protein marker (Vam3 or Vac8), it should be included.

Re: Figure 5. The title does not describe all results in this figure and should be modified accordingly. Panel C. Statistical significance value for *** should be indicated in the legend. It is not clear how many times yeast growth assays were repeated. Usually, all experiments should be done in triplicates or more.

Re: Figure 5 - supplement 1. No title

Re: Figure 5 - supplement 2. No title

Re: Figure 6. There is a typo on the second lane in the legend: "...the genome were", not "...the genome where". Panel C. Why the analysis of BCECF vacuole staining of double mutants *oxr1Δrav1Δ* and *rtc5Δrav1Δ* is not shown? Was it done at all?

Re: Figure 6 - Supplement 2. Why were two different tags (2xmNG and msGFP2) used? Did the authors study N-terminally tagged Oxr1? Was it functional? Panel B. Results for the untagged TEF1pr-Oxr1 overexpression are not shown, thus tagged and untagged proteins can't be compared. Are they available?
What is the promoter for the expression of 2xmNG fusion constructs?

Re: Methods. Were vacuoles prepared differently for XL-MS and SILAC-based vacuole proteomics (there are different references) and why? Methods for XL-MS and quantitative SILAC-based proteomics can be placed together for clarity.
What is CMAC dye? Why was it used to stain the vacuolar lumen?
Some abbreviations (TEAB, ACN) are not explained. What is 0% Ficoll?

****Referees cross-commenting****

I agree with both reviewers, although I think that it is a pretty novel finding because while I was familiar with Oxr1 data I did not realize until now that there is a second protein in yeast. I think it is because homology between Oxr1 and Rtc5 is really low. I also agree that they should study more about what happens with V0 subunits.

2. Significance:

Significance (Required)

Field of expertise keywords:

Protein-protein interactions, V-ATPase, TLDC

The vacuolar-type proton ATPase, V-ATPase, is the key proton pump, that hydrolyses ATP and uses this energy to pump protons across membranes. Amazingly, this proton pump and its function are conserved in eukaryotes from yeast to mammals. While V-ATPase structure and function have been studied for more than 30 years in various organisms, its regulation is not completely understood. The very recent discoveries of two new V-ATPase interacting proteins in yeast, first Oxr1 (OXidative Resistance 1), and now Rtc5 (Restriction of Telomere Capping 5), both the only two members of TLDC (The Tre2/Bub2/Cdc16 (TBC), lysin motif (LysM), domain catalytic) proteins in yeast, provide new insights in V-ATPase regulation in yeast, and because the interaction is conserved in mammals its relevance to mammalian V-ATPases regulation as well.

TLDC proteins are best known for their role in protection from oxidative stress, in particular in yeast and in the nervous system in mammals. The discovery of the novel Rtc5-V-ATPase interaction points to the role of V-ATPase not only in protection from oxidative stress but also in

restriction of telomere capping in yeast and most likely higher species. The studies of other species also highlight the possible conserved role of V-ATPase in lifespan determination and Torc1 signaling, mediated through these interactions. Thus, the discovery of this new functionally important interaction between the second TLDC family member in yeast, Rtc5, and V-ATPase will shed light on the molecular mechanisms of all these essential biological processes and pathways.

In addition, because the authors performed a comprehensive proteomics protein-protein interaction study of the purified yeast vacuole it provides a valuable resource for all researchers who study vacuoles and/or related to them lysosomes.

The follow-up functional studies using the *rav1Δ* strain clearly demonstrated that Rtc5 and Oxr1 disassemble V-ATPase and counteract the function of V-ATPase assembly RAVE complex in vivo in yeast. Thus, they are essentially the first discovered endogenous eukaryotic protein inhibitors of V-ATPase. Moreover, because the authors obtained the evidence that Oxr1 is the regulator of the specific subunit isoform of V-ATPase Stv1p in vivo in yeast, it suggests that different TLDC proteins may regulate different specific V-ATPase subunit isoforms in cell- and tissue-specific manner in higher eukaryotes. The mechanism of this isoform-specific regulation in yeast and other species needs further investigation in the future.

Because of the conservation of the TLDC-V-ATPase interactions, all this information can be extrapolated to higher species, all the way to humans, in whom genetic mutations in various TLDC proteins are known to cause devastating diseases and syndromes.

3. How much time do you estimate the authors will need to complete the suggested revisions:

Estimated time to Complete Revisions (Required)

(Decision Recommendation)

Cannot tell / Not applicable

Yes

Full Revision

Manuscript number: RC-2023-02140

Corresponding author(s): Ayelén González Montoro

1. General Statements [optional]

We are thankful to the reviewers for the time and effort invested in assessing our manuscript and for their suggestions to improve it. We have now considered the points raised by them, carried out additional experiments, and modified the text and figures to address them. We feel that the new experiments and modifications have been able to solve all concerns raised by the reviewers and have improved the manuscript substantially, strengthening and extending our conclusions.

The main modifications include:

- We have extended the analysis of the overexpression strains to highly stringent conditions, which revealed a mild acidification defect for the strain overexpressing Oxr1. In addition, we have included in our analysis a strain in which both proteins are overexpressed, which resulted in a further growth defect.
- We have analyzed the recruitment of Rtc5 to the vacuole under additional conditions: deletion of the main subunit of the RAVE complex *RAVI*, medium containing galactose as the sole carbon source and pharmacological inhibition of the V-ATPase. These experiments allowed us to strengthen and extend our conclusions regarding the requirements for Rtc5 targeting to the vacuole.
- We have analyzed V-ATPase disassembly in intact cells, by addressing the localization to the vacuole of subunit C (*Vma5*) in glucose and galactose-containing medium. The results strengthen our conclusion that both Rtc5 and Oxr1 promote an *in vivo* state of lower V-ATPase assembly.
- We have extended our analyses of V-ATPase function to medium containing galactose as a carbon source, since glucose availability is one of the main regulators of V-ATPase function *in vivo*. The results are consistent with what we observed in glucose-containing medium.
- We have included a diagram of the structure of the V-ATPase for reference.
- We have included a diagram and a paragraph describing Oxr1 and Rtc5 regarding protein length and domain architecture and comparing them to other TLDC domain-containing proteins.
- We have made changes to the text and figures to improve clarity and accuracy, including a methods section that was missing.

We include below a point-by-point response to the reviewers' comments.

Reviewer #1 (Evidence, reproducibility and clarity (Required)):

Suggestions:

1. The authors observed that knockout of Rtc5p or Oxr1p does not affect vacuolar pH. If Rtc5p and Oxr1p both cooperate to dissociate V-ATPase, the authors may wish to characterize the effect of a $\Delta Rtc5p\Delta Oxr1p$ double knockout on vacuolar pH.

The double mutant $\Delta rtc5\Delta oxr1$ was already included in the original manuscript (the growth test is shown in Figure 5 B and the BCECF staining is shown in Figure 5C). This strain behaved like wt in both of these assays. Of note, what we observe for the deletion strains is increased assembly (Figure 5 D - G), so we expect that it would be hard to observe a difference in vacuole acidity or growth in the presence of metals.

Therefore, we have now also included a strain with the double overexpression of Oxr1 and Rtc5. Since overexpression of the proteins results in decreased assembly, it is more likely that this strain will show impaired growth under conditions that strongly rely on V-ATPase activity. Indeed, we observed that the overexpression of Oxr1 alone resulted in a slight growth defect in media containing high concentrations of $ZnCl_2$ and the double overexpression strain showed an even further defect (Figure 6 A and C).

2. The manuscript would benefit from a well-labelled diagram showing the subunits of V-ATPase (e.g. in Figure 2D).

We agree with the reviewer and we have now added a diagram of the structure of the V-ATPase labeling the different subunits in Figure 2B.

3. The images of structures, especially in Figure 1-Supplement 1B, are not particularly clear and could be improved (e.g. by removing shadows or using transparency).

We are thankful to the reviewer for this suggestion. To improve the clarity of the structures in Figure 1 C and Figure 1 – Supplement 1A, we are now presenting the different subunits in the structures with different shades of blue and grey.

4. The authors should clearly describe the differences between Rtc5p and Oxr1p in terms of protein length, sequence identity, domain structure, etc.

We are thankful for this suggestion and we have now included a diagram of the domain architecture and protein length of Rtc5 and Oxr1, comparing with two human proteins containing

Full Revision

a TLDC domain in Figure 5A. In addition, we have added the following paragraph describing the features of the proteins.

“Rtc5 is a 567 residue-long protein. Analysis of the protein using HHPred (Zimmermann et al., 2018), finds homology to the structure of porcine Meak7 (PDB ID: 7U80, (Zi Tan et al., 2022)) over the whole protein sequence (residues 37-559). For both yeast Rtc5 and human Meak7 (Uniprot ID: Q6P9B6), HHPred detects homology of the C-terminal region to other TLDC domain containing proteins like yeast Oxr1 (PDBID: 7FDE), Drosophila melanogaster Skywalker (PDB ID: 6R82), and human NCOA7 (PDB ID: 7OBP), while the N-terminus has similarity to EF-hand domain calcium-binding proteins (PDB IDs: 1EG3, 2CT9, 1S6C6, Figure 5A). HHPred analysis of the 273 residue long Saccharomyces cerevisiae Oxr1, on the other hand, only detects similarity to TLDC domain containing proteins (PDB IDs: 7U80, 6R82, 7OBP), which spans the majority of the sequence of the protein (residues 71-273). The overall sequence identity between Oxr1 and Rtc5 is 24% according to a ClustalOmega alignment within Uniprot. The AlphaFold model that we generated for Rtc5 is in good agreement with the available partial structure of Oxr1 (7FDE) (root mean square deviation (RMSD) of 3.509Å) (Figure 5 - S1 A), indicating they are structurally very similar, in the region of the TLDC domain. Taken together, these analyses suggest that Oxr1 belongs to a group of TLDC domain-containing proteins consisting mainly of just this domain like the splice variants Oxr1-C or NCOA7-B in humans (NP_001185464 and NP_001186551, respectively), while Rtc5 belongs to a group containing an additional N-terminal EF-hand-like domain and a N-myristoylation sequence, like human Meak7 (Finelli & Oliver, 2017) (Figure 5 A).”

Minor:

1. The "O" in VO should be capitalized.

This has been corrected.

2. In Figure 4 supplement 1, the labels "I", "S", and "P" should be defined.

This has been clarified in the figure legend.

3. Please clarify what is meant by "switched labelling"

This refers to the SILAC vacuole proteomics experiments, for which yeast strains are grown in medium containing either L-Lysine or $^{13}\text{C}_6$; $^{15}\text{N}_2$ - L-Lysine to produce normal ('light') or heavy isotope-labeled ('heavy') proteins. This allows comparing two conditions. To increase the robustness of the comparisons, the experiments are done twice with both possible labeling schemes (condition A – light, condition B – heavy + condition A – heavy + condition B – light), which is commonly described as switched labeling or label switching.

We have exchanged the original sentence in the manuscript for:

“Performing the same experiments but switching which strain was labeled with heavy and light amino acids gave consistent results.”

4. The meaning of the sentence "Indeed, this was the case for both of them" is ambiguous.

We have now replaced this sentence with the following:

“Indeed, overexpression of either Rtc5 or Oxr1 resulted in increased growth defects in the context of Stv1 deletion (Figure 7 H and I).”

5. For Figure 1-Supplement 1B it is hard to see the crosslink distances.

We have updated this figure to improve the visibility of the cross-links. In addition, we now include a supplemental table (supplemental table 5) with a list of the Ca- Ca distances measured for all the crosslinks we mapped onto high-resolution structures.

6. The statement "The effects of Oxr1 are greater than those caused by Rtc5" requires more context. Is there a way of quantifying this effect for the reader?

We agree that this sentence was too general and vague. The effects caused by one or the other protein depend on the condition and the assay. We have thus deleted this sentence, and we think it is better to refer to the description of the individual assays performed.

7. The phrase "negative genetic interaction" should be clarified.

We have included in the text the following explanation of genetic interactions:

“A genetic interaction occurs when the combination of two mutations results in a different phenotype from that expected from the addition of the phenotypes of the individual mutations. For example, deletion of OXR1 or RTC5 has no impact on growth in neutral pH media containing zinc in a control background but improves the growth of RAV1 deletion strains (Figure 7 E and F), so this is a positive genetic interaction. On the other hand, overexpression of either Rtc5 or Oxr1 results in a growth defect in a background lacking Rav1 in neutral media containing zinc, a negative genetic interaction.”

8. In the sentence "Isogenic strains with the indicated modifications in the genome where spotted as serial dilutions in media with pH=5.5, pH=7.5 or pH=7.5 and containing 3 mM ZnCl₂", "where" should be "were".

This has been corrected.

Full Revision

9. Figure 2D: the authors should consider re-coloring these models, as it is challenging to distinguish Rtc5p from the V-ATPase.

We have changed the coloring of this structure and added a diagram of the V-ATPase structure with the same coloring scheme to improve clarity.

Reviewer #1 (Significance (Required)):

The vacuolar protein interaction map alone from this manuscript is a nice contribution to the literature. Experiments establishing colocalization of Rtc5p to the vacuole are convincing, as is dependence of this association on the presence of assembled V-ATPase. Similarly, experiments related to myristoylation are convincing. The observed mislocalization of V-ATPases that contain Stv1p to the vacuole (which is also known to occur when Vph1p has been knocked out) upon knockout of Oxr1p is also extremely interesting. Overall, this is an interesting manuscript that contributes to our understand of TLDc proteins.

We are thankful to the reviewer for their appreciation of the significance of our work, including the interactome map of the vacuole as a resource and the advances on the understanding of the regulation of the V-ATPase by TLDc domain-containing proteins.

Reviewer #2 (Evidence, reproducibility and clarity (Required)):

Major points:

1. The evidence of Oxr1 and Rtc5 as V-ATPase disassembly factors is circumstantial. The authors base their interpretation primarily on increased V1 (but not Vo) at purified vacuoles from Oxr1- or Rtc5-deleted strains, which does not directly address disassembly. Of course, the results regarding Oxr1 confirm detailed disassembly experiments with the purified protein complex (PMID 34918374), but on their own are open to other interpretations, e.g. suppression of V-ATPase assembly. Of note, the authors emphasize that they provide first evidence of the *in vivo* role of Oxr1, but monitor V1 recruitment with purified vacuoles and do not follow V-ATPase assembly in intact cells.

We are thankful to the reviewer for pointing this out. We did not want to express that the molecular activity of the proteins is the disassembly of the complex, as our analyses include *in vivo* and *ex vivo* experiments and do not directly address this. We rather meant that both proteins promote an *in vivo* state of lower assembly of the V-ATPase. We have modified the wording throughout the manuscript to be clearer about this.

In addition, we have added new experiments to monitor V-ATPase assembly in intact cells, as suggested by the reviewer. Previous work has shown that in yeast, only subunit C leaves the vacuole membrane under conditions that promote disassembly, while the other subunits remain

at the vacuole membrane (Tabke et al 2014). Our own experiments agree with what was published (Figure 3 D). We have thus monitored Vma5 localization to the vacuole under glucose or after shift to galactose containing media in cells lacking or overexpressing Rtc5 or Oxr1. We observed that cells overexpressing either TLDC domain protein show lower levels of Vma5 recruitment to the vacuole in glucose (Figure 6 D and E). Additionally cells lacking either Rtc5 or Oxr1 contain higher levels of Vma5 at the vacuole after 20 minutes in galactose medium (Figure 5 F and G). Thus, these results re-inforce our conclusions that Rtc5 and Oxr1 promote states of lower assembly.

2. Oxr1 and Rtc5 have very low sequence similarity. It would be helpful if the authors provided more detail on the predicted structure of the putative TLDC domain of Rtc5 and its relationship to the V-ATPase - Oxr1 structure. Is Rtc5 more closely related to established TLDC domain proteins in other organisms?

We have now included a diagram of the domain architecture of Rtc5 and Oxr1, and comparison to the features of other TLDC domain containing proteins in Figure 5 A, as well as a paragraph describing them:

“Rtc5 is a 567 residue-long protein. Analysis of the protein using HHPred (Zimmermann et al., 2018), finds homology to the structure of porcine Meak7 (PDB ID: 7U80, (Zi Tan et al., 2022)) over the whole protein sequence (residues 37-559). For both yeast Rtc5 and human Meak7 (Uniprot ID: Q6P9B6), HHPred detects homology of the C-terminal region to other TLDC domain containing proteins like yeast Oxr1 (PDBID: 7FDE), Drosophila melanogaster Skywalker (PDB ID: 6R82), and human NCOA7 (PDB ID: 7OBP), while the N-terminus has similarity to EF-hand domain calcium-binding proteins (PDB IDs: 1EG3, 2CT9, 1S6C6, Figure 5A). HHPred analysis of the 273 residue long Saccharomyces cerevisiae Oxr1, on the other hand, only detects similarity to TLDC domain containing proteins (PDB IDs: 7U80, 6R82, 7OBP), which spans the majority of the sequence of the protein (residues 71-273). The overall sequence identity between Oxr1 and Rtc5 is 24% according to a ClustalOmega alignment within Uniprot. The AlphaFold model that we generated for Rtc5 is in good agreement with the available partial structure of Oxr1 (7FDE) (root mean square deviation (RMSD) of 3.509Å) (Figure 5 - S1 A), indicating they are structurally very similar, in the region of the TLDC domain. Taken together, these analyses suggest that Oxr1 belongs to a subfamily of TLDC domain-containing proteins consisting mainly of just this domain like the splice variants Oxr1-C or NCOA7-B in humans (NP_001185464 and NP_001186551, respectively) , while Rtc5 belongs to a subfamily containing an additional N-terminal EF-hand-like domain and a N-myristoylation sequence, like human Meak7 (Finelli & Oliver, 2017) (Figure 5 A).”

3. The authors conclude vacuolar recruitment of Rtc5 depends on the assembled V-ATPase, based on deletion of different V1 and Vo domain subunits. However, these genetic manipulations likely cause a strong perturbation of vacuolar acidification; indeed, the images show drastically altered vacuolar morphology. To strengthen their conclusion, it would be helpful to show that Rtc5 recruitment is not blocked by inhibition of vacuolar acidification, and that conversely it is blocked by deletion of rav1.

We are thankful to the reviewer for this insightful suggestion and we have now performed both experiments suggested. The experiment regarding *rav1Δ* is now Figure 3C, and we observed that this mutation also disrupts Rtc5 localization to the vacuole. In addition, we decided to include an experiment showing the subcellular localization of Rtc5 after shifting the cells to galactose containing medium for 20 minutes, as a physiologically relevant condition that results in disassembly of the complex (Figure 3D). We observed that under these conditions Rtc5 re-localizes to the cytosol. This result is particularly interesting given that in yeast only subunit C (but not other V₁ subunits) re-localizes to the cytosol under these conditions. In addition, the experiment using Bafilomycin A to inhibit the V-ATPase shows that Rtc5 is still localized at the vacuole membrane under conditions of V-ATPase inhibition (Figure 3 F). Taken together these results allowed us to strengthen our original interpretation that Rtc5 requires an assembled V-ATPase for its localization and extend it to the fact that the V-ATPase does not need to be active.

Reviewer #2 (Significance (Required)):

This is an interesting paper that confirms and extends previous findings on TLDC domain proteins as a novel class of proteins that interact with and regulate the V-ATPase in eukaryotes. The title seems to exaggerate the findings a bit, as the authors do not investigate V-ATPase (dis)assembly directly and only phenotypically describe altered subcellular localization of the Golgi V-ATPase in Oxr1-deleted cells. A recent structural and biochemical characterization of Oxr1 as a V-ATPase disassembly factor (PMID 34918374) somewhat limits the novelty of the results, but the function of Oxr1 in regulating subcellular V-ATPase localization and the identification of a second potential TLDC domain protein in yeast provide relevant insights into V-ATPase regulation. This paper will be of interest to cell biologists and biochemists working on lysosomal biology, organelle proteomics and V-ATPase regulation.

We thank the reviewer for the assessment of our work, and for recognizing the novel insights that we provide. Regarding the previous biochemical work on Oxr1 and the V-ATPase, we have clearly cited this work in the manuscript. In our opinion, our results complement and extend this article, showing that the function in disassembly is relevant *in vivo*. Additionally, this is only one of five major points of the article, the other four being

- The interactome map of the vacuole as a resource
- The identification of Rtc5 as a second yeast TLDC domain containing protein and interactor of the V-ATPase.
- The identification of the role of Rtc5 in V-ATPase assembly.
- The identification of the role of Oxr1 in Stv1 subcellular localization.

We believe these additional points add important insights to researchers interested in lysosomes, the V-ATPase, intracellular trafficking and TLDC-domain containing proteins.

Reviewer #3 (Evidence, reproducibility and clarity (Required)):

Major comments

1) Re: A cross-linking mass spectrometry map of vacuolar protein interactions (results)

While XL-MS is a very powerful method, it is a high-throughput approach and there should be some kind of negative control in these experiments. In cross-linking experiments, non-cross-linked samples are usually used as negative controls. What was the negative control in cross-linking mass-spectrometry experiments here? If there was no negative control, how the specificity of interactions was evaluated? Maybe the authors analyzed the dataset for highly improbable interactions and found very few of them?

We fully agree that it is crucial to ensure the specificity of the interactions detected by XL-MS. To achieve this, one needs to control (1) the specificity of the data analysis (i.e. that the recorded mass spectrometry data are correctly matched to cross-linked peptides from the sequence database) and (2) the biological specificity (i.e. that the cross-linking captured natively occurring interactions).

To ascertain that criterion (1) is met, cross-link identifications are filtered to a pre-defined false-discovery rate (FDR) – an approach that the XL-MS field adopted from mass spectrometry-based proteomics. As a result, low-confidence identifications (e.g. cross-linked peptides that are only supported by a few signals in a given mass spectrum) are removed from the dataset. FDR filtering in XL-MS is a rather complex matter as it can be done at different points during data analysis and the optimal FDR cut-off depends on the specific scientific question at hand (for more details see for example Fischer and Rappsilber, *Anal Chem*, 2017). Generally speaking, an overly restrictive FDR cut-off would remove a lot of correct identifications, thereby greatly limiting the sensitivity of the analysis. On the other hand, a too relaxed FDR cut-off would dilute the correct identifications with a high number of false-positives, which would impair the robustness and specificity of the dataset. While many XL-MS study control the FDR on the level of individual spectrum matches, we opted for a 2% FDR cut-off on the level of unique residue pairs, which is more stringent (see Fischer and Rappsilber, *Anal Chem*, 2017). Our FDR parameters are described in the Methods section (Cross-linking mass spectrometry of isolated vacuoles - Data analysis). Of note, we have made all raw mass spectrometry data publicly available through the PRIDE repository (<https://www.ebi.ac.uk/pride/>; accession code PXD046792; login details during peer review: Username = reviewer_pxd046792@ebi.ac.uk, Password = q1645ITP). This will allow other researchers to re-analyze our data with the data analysis settings of their choice in the future.

To ascertain that criterion (2) is met, we mapped the identified cross-links onto existing high-resolution structures of vacuolar protein complexes. Taking into account the length of our cross-linking reagent, the side-chain length of the cross-linkable amino acids (i.e. lysines), and a certain degree of in-solution flexibility, cross-links can reasonably occur between lysines with a mutual C α -C α distance of up to 35 Å. Using this cut-off, the lysine-lysine pairs in the high-resolution structures we studied can be split into possible cross-linking partners (C α -C α distance < 35 Å)

and improbable cross-linking partners ($C\alpha$ - $C\alpha$ distance > 35 Å). Of all cross-links we could map onto high-resolution structures, 95.2% occurred between possible cross-linking partners. In addition, our cross-links reflect numerous known vacuolar protein interactions that have not yet been structurally characterized. These lines of evidence increase our confidence that our XL-MS approach captured genuine, natively occurring interactions. These analyses are described in more detail in the first Results sub-section (“A cross-linking mass spectrometry map of vacuolar protein interactions”).

In addition, the high purity of vacuole preparation is critical. How was it assessed by the authors?

We disagree that the purity of the vacuole preparation is critical for this analysis to be valid. The accuracy of the protein-protein interactions detected will depend on their preservation during sample preparation until the sample encounters the cross-linker, and the data analysis, as described above. The experiment would have been equally valid if performed on whole cell lysates without any enrichment of vacuoles, but the coverage of vacuolar proteins would have likely been very low. For this reason, we decided to use the vacuole isolation procedure to obtain better coverage of the proteins of this particular organelle. The use of the Ficoll gradient protocol (Haas, 1995) was based on that it is a protocol that yields strong enrichment of proteins annotated with the GO Term “vacuole” (Eising et al, 2019) and that it preserves the functionality of the organelle, as evidenced by its use for multiple functional assays (vacuole-vacuole fusion (Haas, 1995), autophagosome-vacuole fusion (Gao et al, 2018), polyphosphate synthesis by the VTC complex (Desfougères et al, 2016), among others).

2) Re: Rtc5 and Oxr1 counteract the function of the RAVE complex (results)

Taken together, data, presented in this section of the manuscript, provide strong evidence that Rtc5 and Oxr1 negatively regulate V-ATPase activity, counteracting the V-ATPase assembly, facilitated by the activity of the RAVE complex. However, the complete deletion of the major RAVE subunit Rav1p was required to observe this effect *in vivo* in yeast. The other way to induce V-ATPase disassembly in yeast is glucose deprivation. It will be interesting to study if there is a synergistic effect between glucose deprivation and RTC5/OXR1 deletion on V-ATPase assembly, vacuolar pH, and growth of single *oxr1* Δ , *rtc5* Δ or double *oxr1* Δ *rtc5* Δ mutants (OPTIONAL). Glucose deprivation is a more physiologically relevant condition than a deletion of an entire gene.

We would like to point out that an effect on assembly is observed without deleting the RAVE complex: deletions of Oxr1 or Rtc5 resulted in increased V-ATPase assembly *in vivo* in the presence of glucose and of the RAVE complex (Figures 5 D and E). We have now also added the experiments showing that the overexpression strains have a mild growth defect under conditions that force cells to strongly rely on V-ATPase activity (Figures 6 A and C).

Nevertheless, we agree that addressing the effect of changing the levels of Oxr1 and Rtc5 under low-glucose conditions is an interesting physiologically relevant question. We have now included growth assays and BCECF staining in medium containing galactose as the carbon source

(Figures 5 – Supplement 1 B and C, and Figure 6 C and Figure 6- Supplement 1A). In addition, we have addressed the vacuolar localization of Vma5 in medium containing glucose or after shifting to medium containing galactose for 20 minutes, as a proxy for V-ATPase disassembly in intact cells (Figure 5 F and G, Figure 6 D and E). Taken together, these analyses reinforce our conclusions that both Rtc5 and Oxr1 promote an *in vivo* state of lower V-ATPase assembly, based on the following observations:

- Higher localization of Vma5 to the vacuole after 20 mins in galactose in cells lacking Oxr1 or Rtc5 (Figure 5 F and G).
- Lower localization of Vma5 to the vacuole in medium containing glucose in cells overexpressing Oxr1 or Rtc5 (Figure 6 D and E).
- Growth defect of the strain overexpressing Oxr1 in medium containing galactose with pH = 7.5 and zinc chloride, with a further growth defect caused by additional overexpression of Rtc5 (Figure 6 C).

3) Re: Figure 6 - supplement 1. The title is relevant to panel D only, it should be renamed to reflect the results of the disassembly of V-ATPase in *rav1Δ* mutant strains, while results about the *stv1Δ*-based strains (Panel D) should be shown together with similar experiments in Figure 7 - supplement 2 for clarity.

We have shifted the Panel D from the original Figure 6 – Supplement 1 to the main Figure (now Figure 7 – H and I). Regarding the title of the Figure, whether Supplemental Figures have titles or not will depend on the journal where the manuscript is published. For now, we have removed all titles from supplemental figures, as they are conceived to complement the main Figures.

4) Re: Figure 7 - supplement 1, Panel A. The proper assay to show that Stv1-mNeonGreen is functional is to express it in double mutant *vph1Δstv1Δ* to see if the growth defect is reversed. In addition, the *vph1Δ* growth defect is not changed (improved or worsened) in the presence of Stv1-mNeonGreen, so it means that the expression of Stv1-mNeonGreen does not further compromise the V-ATPase function, but it does not mean that it improves its function.

It is clear from the experiment suggested by the reviewer that they think that we have expressed Stv1-mNeonGreen from a plasmid. This was not the case, Stv1 was C-terminally tagged with mNeonGreen in the genome. It is thus the only expressed version in the strain. The experiment we have performed is thus equivalent to the one suggested by the reviewer, but for genomically expressed variants. For reference, the genotypes of all the strains used can be found in Supplemental Table 1.

5) Re: Figure 7 - supplement 2. This figure should be combined with Fig. 6- suppl 1, panel D as also mentioned above. The figure seems to lack some labels, and conclusions are not accurate as discussed below. However, this data provides important additional information about relationships between isoform-specific subunits of V-ATPase Vph1 and Stv1 and both Rtc5 and Oxr1 and should be repeated if it is not done yet to have a better idea about these relationships.

Panel B: Based on this picture, deletion of *RTC5* has a negative genetic interaction with the deletion of *VPH1*, since double deletion mutant *vph1Δ rtc5Δ* grows worse than each individual mutant. Although it also means that there is no positive interaction, it is not the same.

Indeed, there is a negative genetic interaction between the deletion of *RTC5* and *VPH1*. We have replaced the growth tests in this figure (Figure 8 – Supplement 2 A in the new manuscript) to show this negative genetic interaction better. This effect is reproducible, as shown in the repetitions of the experiments.

Panel C: Same as for panel B. Based on this picture, the deletion of *OXR1* has a weak negative genetic interaction with the deletion of *STV1*, since double deletion mutant *stv1Δ oxr1Δ* grows worse than each individual mutant at 6 mM $ZnCl_2$.

Panel D: Same as for panels B and C. Based on this picture, deletion of *RTC5* has a negative genetic interaction with the deletion of *STV1*, since double deletion mutant *stv1Δ rtc5Δ* grows worse than each individual mutant at 6 mM $ZnCl_2$. There is no label in the middle panel (growth conditions) and no growth assay data in the presence of $CaCl_2$.

However, these results will be then in contradiction with the results from Figure 6 - Supplement 1, panel D, showing negative genetic interaction between the overexpression of *Rtc5* or *Oxr1* and deletion of *Stv1*, since both deletion and overexpression of *Rtc5* or *Oxr1* would have negative genetic interactions with *Stv1*.

For both Panels C and D (Now Figure 8 - Supplement 2 B and C). The effect pointed out by the reviewer (slightly stronger growth defect for the double mutants than for the single mutants) is very mild. We have attempted to make it more evident by assessing growth in medium with higher and lower concentrations of zinc and this was not possible. This is in contrast with the very clear positive genetic interaction that we observe between the deletion of *OXR1* and *VPH1* (Now Figure 8 H). This is the reason that we decided to report the lack of a positive genetic interaction instead of the presence of a negative one, as we do not want to draw conclusions based on results that are borderline detectable.

In addition, there is no label for the media in the middle panel, is it just YPAD pH=7.5, without the addition of any metals?

Indeed, the media is YPAD pH=7.5, without the addition of any metals. The line drawn above several images based on this media indicated this. Since this form of labeling appears to be confusing, we have now replaced it and placed the label directly above the image.

Why there is no growth assay in the presence of $CaCl_2$, like in panels A and B?

Every growth test shown in the manuscript was performed including growth in YPD pH=5,5 as a control of a permissive condition for lack of V-ATPase activity, and then in YPD pH=7,5 including a broad range of Zinc Chloride and Calcium chloride concentrations. From all these pictures, the conditions where the differences among strains were clearly visible were chosen to assemble the

figures. Conditions that did not provide any information for that particular experiment were not included in the figure to avoid making them unnecessarily large and crowded.

Re: Figure 7 - supplement 2, continued. How many times all these experiments were repeated? These experiments should be repeated at least 3 times, which is especially necessary for the experiments in panel C, because the effects are borderline. If results are reproducible and statistically significant, although small, the conclusion should be changed from "no positive genetic interactions" to "negative genetic interactions", which is more precise and informative.

All growth tests shown in the manuscript were repeated at least three times for the conditions shown. We are thankful to the reviewer for pointing out that this was not mentioned, and we have added this to the methods section. We have assembled a file with all repetitions of the shown growth tests and added it at the end of this file. In doing so, these are already available for the public. These repetitions show that all effects reported are reproducible. We will then discuss with the editors of the journal where this manuscript is published about the necessity of including it with the final article.

Regarding reporting the lack of a positive genetic interaction vs. a negative one, we have discussed this above. Shortly, for Panel B (Figure 8 – Supplement 2 A in the new manuscript) we have changed the conclusion to “negative genetic interaction” as adjusting the zinc chloride concentration allowed us to show this clearly and reproducibly, as shown by the repetitions of the experiments. For panels C and D (Now Figure 8 - Supplement 2 B and C), the effect is really mild and barely detectable, even when we tried a wide range of zinc chloride concentrations. For this reason, we would prefer to maintain the “no positive genetic interaction” conclusion.

Re: Methods. There is no description of yeast serial dilution growth assay at all. In addition, why the specific media (neutral pH, in the presence of high concentrations of calcium or zinc) was used is not explained either in the results or methods. Appropriate references should be included, for example, PMID: 2139726, PMID: 1491236.

We apologize for the oversight of the missing methods section, which we have now included.

Regarding the explanation of the media used, the following section was already a part of the results section, before the description of the first growth test:

“The V-ATPase is not essential for viability in yeast cells, and mutants lacking subunits of this complex grow similarly to a wt strain in acidic media. However, when cells grow at near-neutral pH or in the presence of divalent cations such as calcium and zinc, the mutants lacking V-ATPase function show a strong growth impairment (Kane et al, 2006).”

We have now replaced this with the following, more complete version:

“As a first approach for addressing the role of these proteins, we tested growth phenotypes related to V-ATPase function in strains lacking or overexpressing them. The V-ATPase is not essential for viability in yeast cells, and mutants lacking subunits of this complex grow similarly to a wt strain in acidic media, but display a growth defect at near-neutral pH the mutants (Nelson & Nelson, 1990). In addition, the proton gradient across the vacuole membrane generated by the V-ATPase energizes the pumping of metals into the vacuole, as a mechanism of detoxification. Thus, increasing concentrations of divalent cations such as calcium and zinc, generate conditions in which growth is increasingly reliant on V-ATPase activity (Förster & Kane, 2000; MacDiarmid et al, 2002; Kane, 2006).”

MINOR COMMENTS

Yeast proteins are named with "p" at the end, such as "Rtc5p".

This nomenclature rule is falling into disuse during the last decades, as the use of capitals vs lowercase and italics allows to distinguish between genes proteins and strains (*OXR1* = gene, Oxr1 = protein, *oxr1* Δ = strain). As an example, I include a list of the latest papers by some of the major yeast labs around the world, all of which use the same nomenclature as we do (in alphabetical order). This list even includes some work in the field of the V-ATPase.

- Alexey Merz, USA. PMID: 33225520
- Benoit Kornmann, UK. PMID: 35654841
- Christian Ungermann, Germany. PMID: 37463208
- Claudio de Virgilio, Switzerland. PMID: 36749016
- Daniel E. Gottschling, USA. PMID: 37640943
- David Teis, Austria. PMID: 32744498
- Elizabeth Conibear, Canada. PMID: 35938928
- Fulvio Reggiori, Denmark. PMID: 37060997
- J Christopher Fromme, USA. PMID: 37672345
- Maya Schuldiner, Israel. PMID: 37073826
- Patricia Kane, USA. PMID: 36598799
- Scott Emr, USA. PMID: 35770973
- W Mike Henne, USA. PMID: 37889293
- Yoshinori Ohsumi, Japan. PMID: 37917025

In addition, we would prefer to keep the nomenclature that we already use, to keep consistency with other published articles from our lab.

Re: Introduction. In the introduction it should be indicated that Rtc5 was originally discovered as a "restriction of telomere capping 5", using screening of temperature-sensitive *cdc13-1* mutants

combined with the yeast gene deletion collection [PMID: 18845848]. A couple of sentences should be written about the RAVE complex and its role in V-ATPase assembly.

We are thankful for this suggestion and we have now included both pieces of information in the introduction.

“The re-assembly of the V_1 onto the V_0 complex when glucose becomes again available, is aided by a dedicated chaperone complex known as the RAVE complex, which also likely has a general role in V-ATPase assembly (Seol et al, 2001; Smardon et al, 2002, 2014).”

“In our cross-linking mass spectrometry interactome map of isolated vacuoles we found that the only other TLDC-domain containing protein of yeast, Rtc5, is a novel interactor of the V-ATPase. Rtc5 is a protein of unknown function, originally described in a genetic screen for genes related to telomere capping (Addinall et al, 2008)”

Re: The TLDC domain-containing protein of unknown function Rtc5 is a novel interactor of the vacuolar V-ATPase (results)

1) It is important to understand, that Oxr1 was co-purified before with the V_1 domain of V-ATPase from a certain mutant strain, not wild-type yeast [PMID: 34918374]. It may explain why the authors did not identify it in their original protein-protein interactions screen here.

The structural work on the V_1 domain bound to Oxr1 (Khan et al, 2022) showed that the binding of Oxr1 prevented V_1 from assembling onto the V_0 . Since our experiments rely on the purification of vacuoles, they should contain mainly only V_1 assembled onto the V_0 , and not the free soluble V_1 . This is likely the reason that we do not detect Oxr1, in addition to it being less abundant. We have clarified this now in the manuscript and added the fact that Oxr1 was co-purified with a V_1 containing a mutant version of the H subunit.

“In a previous study, Oxr1 was co-purified with a V_1 domain containing a mutant version of the H subunit, and its presence prevented the in vitro assembly of this V_1 domain onto the V_0 domain and promoted disassembly of the holocomplex (Khan et al., 2022). This is likely the reason why we do not detect Oxr1 in our experiments, which rely on isolated vacuoles and thus would only include V_1 domains that are assembled onto the membrane. In addition, Oxr1 is less abundant in yeast cells than Rtc5 according to the protein abundance database PaxDb (Wang et al, 2015).”

2) It is a wrong conclusion that because Rtc5 was co-purified with both V_1 and V_0 domain subunits it interacts with the assembled V-ATPase, this does not exclude a possibility that Rtc5 also interacts with separate V_1 sector or separate V_0 sector of V-ATPase.

Full Revision

We agree with the reviewer that the co-purification of Rtc5 with both V_1 and V_0 domain subunits does not necessarily mean that it interacts with the assembled V-ATPase. Thus, we have modified the text in this part to:

“The fact that we can co-enrich Rtc5 both with Vma2 and with Vph1 indicates that it can interact either with both the V_0 and V_1 domains or with the assembled V-ATPase.”

However, other results throughout the manuscript can be taken into account to strengthen this idea:

1. Rtc5 requires an assembled V-ATPase to localize to the vacuole membrane, and thus seems not to interact with free V_0 domains, which would be available when we delete V_1 subunits or in medium containing galactose.

2. Rtc5 becomes cytosolic in galactose-containing media. This would indicate that it also does not interact with free V_1 domains, which are still localized to the vacuole membrane under these conditions.

Taken together with the pull-downs, these results suggest that Rtc5 interacts with the assembled V_1 - V_0 V-ATPase. Thus, we have included the following sentence after Figure 3, which shows the subcellular localization experiments.

“Taking into account that Rtc5 is co-enriched with subunits of both the V_0 and V_1 domain, and that it localizes at the vacuole membrane dependent on an assembled V-ATPase, we suggest that Rtc5 interacts with the assembled V-ATPase complex.”

Re: Figure 1, Panel C. Is it possible to show individual proteins in different colors for clarity? Panel D. How were cross-link distances measured? It is not obvious if you are not an expert in the field and it is not described in the methods.

We have modified Figure 1 C and Figure 1 – Supplement 1B (now Figure 1 – Supplement 1 A) to present the different subunits in the structures with different shades of blue and grey.

Furthermore, we have clarified the distance measurement approach in the methods section and in the legend of Fig 1D: “Ca-Ca distances were determined using the measuring function in Pymol v.2.5.2 (Schrodinger LLC).”

Re: Figure 1 - Supplement 1,

Panel A. What scientific information are we getting from this picture?

This panel was just a visual representation of the complexity of the protein network we obtained. Indeed, there was no specific scientific message, so we have decided to remove this panel from the revised manuscript.

Panel B. Why are these complexes shown separately from the complexes in Figure 1, panel C? Also, can individual proteins be colored differently here as well?

We did not want to overload Fig 1C, so we decided to show some of the protein complexes in Fig 1 – Supplement 1B. The most important information is the histogram showing that 95% of the mapped cross-links fall within the expected length range, and this is shown in the main Figure (Figure 1D). As stated above, we have adjusted the subunit coloring in Figure 1 C to improve clarity.

Re: Figure 3. It will be nice to show the localization of the untagged protein as well if antibodies are available (OPTIONAL).

Unfortunately, there are no available antibodies for either Rtc5 or Oxr1. This hinders us from detecting the endogenous untagged proteins. We would like to point out that we have been very careful in showing which tagged proteins are functional (C-terminally tagged Rtc5) and which are not (C-terminally tagged Oxr1), so that the reader can know how to interpret the localization data.

Re: Figure 4. Why different tags were used in panels A (GFP), C (msGFP2) and D (mNeonGreen)?

In general, we prefer to use mNeonGreen as a tag for microscopy experiments because it is brighter and more stable, and msGFP2 as a tag for experiments involving Western blots because we have better antibodies available. There was a mistake in the labeling, and actually, all constructs labeled as GFP were msGFP2. We have now corrected this. Of note, we have tested the functionality of both tagged version (mNeonGreen and msGFP2).

Panels B and C. Were Rtc5 fusions detected using anti-GFP antibodies?

Indeed, Rtc5-msGFP2 was detected with an anti-GFP antibody. We have now indicated next to each Western blot membrane the primary antibody used. In addition, all antibodies are detailed in Supplemental Figure 3.

The authors should have full-size Western blots available, not just cut-out bands, as some journals and reviewers require them for publication.

For all western blots, we always showed a good portion of the membrane and not cut-out bands. The cropping was performed to avoid making figures unnecessarily large. The whole membranes are of course available and will be included in an “extended data file” if required by the journal.

Re: Figure 4 - Supplement 1, Panel A. Does "-" and "+" mean +/- Azido-Myr?

Indeed. We have now added this label to the figure.

Panel B. There is no blot with a membrane protein marker (Vam3 or Vac8), it should be included.

We have replaced this western blot for a different repetition of this experiment in which a membrane protein marker was included. Of note, the two other repetitions of the experiment shown (Figure 4 – Supplement 1 panel C and Figure 4 panel C) also include both a membrane protein marker and a soluble protein marker.

Re: Figure 5. The title does not describe all results in this figure and should be modified accordingly.

The original data from Figure 5 is now separated into Figures 5 and 6 because of the additional experiments included during revisions. We have modified the Figure titles to be descriptive of the overall message of the Figures.

Panel C. Statistical significance value for *** should be indicated in the legend.

This has been indicated in the Figure legend.

It is not clear how many times yeast growth assays were repeated. Usually, all experiments should be done in triplicates or more.

All shown growth tests were performed at least three times for the conditions shown. We have now indicated this in the materials and methods section. In addition, we now provide in this response a file with all repetitions of growth tests, which will be appended to the article if deemed necessary by the editors.

Re: Figure 5 - supplement 1. No title

Re: Figure 5 - supplement 2. No title

Whether the supplemental Figures should have a title or not will depend on the style of the journal where the manuscript is finally published. The current idea of the supplemental Figures is that they complement the corresponding main Figure. For this reason, we have removed all titles from supplemental Figures.

Re: Figure 6. There is a typo on the second lane in the legend: "...the genome were", not "...the genome where".

This has been corrected.

Panel C. Why the analysis of BCECF vacuole staining of double mutants *oxr1Δrav1Δ* and *rtc5Δrav1Δ* is not shown? Was it done at all?

We had not included this piece of data, as we thought that the genetic interaction of *RTC5* and *OXR1* and *rav1Δ* was sufficiently well supported with the included data (growth tests in combination with the deletion, growth tests in combination with the overexpression, vacuole proteomics in combination with overexpression, and BCECF staining in combination with the overexpression). Because of the request of the reviewer, we have now included this experiment as Figure 7 G.

Re: Figure 6 - Supplement 2. Why were two different tags (2xmNG and msGFP2) used?

We tried both tags to see if one of them would be functional. Unfortunately, they both resulted in non-functional proteins, as shown by the corresponding growth tests.

Did the authors study N-terminally tagged Oxr1? Was it functional?

We have tagged Oxr1 N-terminally, and this unfortunately resulted in a protein that was not completely functional. We show below the localization of N-terminally mNeon-tagged Oxr1, under the control of the *TEF1* promoter. The protein appears cytosolic (Panel A) but is not completely functional (Panel B). The localization of Oxr1 had already been misreported by using a tagged version that we now show to be non-functional. For this reason, we preferred not to include this data in the manuscript, to avoid again including in the literature subcellular localizations that correspond to non-functional or partially functional proteins.

Panel B. Results for the untagged TEF1pr-Oxr1 overexpression are not shown, thus tagged and untagged proteins can't be compared. Are they available? What is the promoter for the expression of 2xmNG fusion constructs?

Oxr1-2xmNG was C-terminally tagged in the genome, which means that the promoter is the endogenous one, it was not modified. For this reason, the correct controls are a strain expressing Oxr1 at endogenous levels (the wt strain) and a strain lacking Oxr1. Both controls were included in the Figure, and in all repetitions made of this experiment. For reference, all the genotypes of the strains used are found in Supplemental Table 1.

Re: Methods. Were vacuoles prepared differently for XL-MS and SILAC-based vacuole proteomics (there are different references) and why? Methods for XL-MS and quantitative SILAC-based proteomics can be placed together for clarity.

The basis for the method of vacuole purification is the same, from (Haas, 1995). This reference was included in both protocols that include vacuole purifications. However, modifications of this method were performed to fit the crosslinking method (higher pH, no primary amines) or to fit the SILAC labeling (combination of two differentially labeled samples in one purification). The reference for the vacuole proteomics (Eising et al 2022) corresponds to a paper in which the SILAC-based comparison of vacuoles from different mutant strains was optimized, and includes not only the vacuole purification but the growth conditions and downstream processing of the vacuoles.

Since both the SILAC-based vacuole proteomics and the XL-MS are multi-step methods, containing numerous parameters including the sample preparation, processing for MS, MS run and data analysis, we would prefer to keep them separate. We think this would allow a person attempting to reproduce these methods to go through them step by step.

What is CMAC dye? Why was it used to stain the vacuolar lumen?

We apologize for this oversight, we have included the definition of CMAC as 7-Amino-4-Chlormethylcumarin. It is a standard-used organelle marker for the lumen of the vacuole.

Some abbreviations (TEAB, ACN) are not explained.

We apologize for this oversight. We have now replaced these abbreviations with the full names of the compounds in the article.

What is 0% Ficoll?

We used the term 0% Ficoll, because this is the name given to the buffer in the original Haas 1995 paper on vacuole purifications. However, we agree that the term is misleading and we have now added the composition of the buffer (10 mM PIPES/KOH pH=6.8, 0.2 M Sorbitol).

Reviewer #3 (Significance (Required)):

The vacuolar-type proton ATPase, V-ATPase, is the key proton pump, that hydrolyses ATP and uses this energy to pump protons across membranes. Amazingly, this proton pump and its function are conserved in eukaryotes from yeast to mammals. While V-ATPase structure and function have been studied for more than 30 years in various organisms, its regulation is not completely understood. The very recent discoveries of two new V-ATPase interacting proteins in yeast, first Oxr1 (OXidative Resistance 1), and now Rtc5 (Restriction of Telomere Capping 5), both the only two members of TLDC (The Tre2/Bub2/Cdc16 (TBC), lysin motif (LysM), domain catalytic) proteins in yeast, provide new insights in V-ATPase regulation in yeast, and because the interaction is conserved in mammals its relevance to mammalian V-ATPases regulation as well.

TLDC proteins are best known for their role in protection from oxidative stress, in particular in yeast and in the nervous system in mammals. The discovery of the novel Rtc5-V-ATPase interaction points to the role of V-ATPase not only in protection from oxidative stress but also in restriction of telomere capping in yeast and most likely higher species. The studies of other species also highlight the possible conserved role of V-ATPase in lifespan determination and Torc1 signaling, mediated through these interactions. Thus, the discovery of this new functionally important interaction between the second TLDC family member in yeast, Rtc5, and V-ATPase will shed light on the molecular mechanisms of all these essential biological processes and pathways.

In addition, because the authors performed a comprehensive proteomics protein-protein interaction study of the purified yeast vacuole it provides a valuable resource for all researchers who study vacuoles and/or related to them lysosomes.

The follow-up functional studies using the *rav1Δ* strain clearly demonstrated that Rtc5 and Oxr1 disassemble V-ATPase and counteract the function of V-ATPase assembly RAVE complex in vivo in yeast. Thus, they are essentially the first discovered endogenous eukaryotic protein inhibitors of V-ATPase. Moreover, because the authors obtained the evidence that Oxr1 is the regulator of the specific subunit isoform of V-ATPase Stv1p in vivo in yeast, it suggests that different TLDC proteins may regulate different specific V-ATPase subunit isoforms in cell- and tissue-specific manner in higher eukaryotes. The mechanism of this isoform-specific regulation in yeast and other species needs further investigation in the future.

Because of the conservation of the TLDC-V-ATPase interactions, all this information can be extrapolated to higher species, all the way to humans, in whom genetic mutations in various TLDC proteins are known to cause devastating diseases and syndromes.

We are thankful to the reviewer for their positive comments about the significance of our work.

Repetitions of all growth tests included in the manuscript

Figure 5 - Panel B

Figure 5 - Supplement 1 Panel B

Repetition #2

Repetition #3

Figure 6 - Panel A

Repetition #2

Repetition #3

Figure 6 - Panel C

Figure 7- Panel A and B

Panel A - Repetition #2

Panel A - Repetition #3

Panel B - Repetition #2

Panel B - Repetition #3

Figure 7 - Panel E and Panel F

Panel E - Repetition #2

Panel E - Repetition #3

Panel F - Repetition #2

Panel F - Repetition #3

Figure 7 - Panel H and Panel I

Repetition #2

Repetition #3

Figure 7 - S2 Panel A and Panel B

Panel A - Repetition #2

Panel A - Repetition #3

Panel B - Repetition #2

Panel B - Repetition #3

Figure 7 - S2 Panel C and Panel D

Panel C - Repetition #2

Panel C - Repetition #3

Panel D - Repetition #2

Panel D - Repetition #3

Figure 8 - Panel H

Repetition #2

Repetition #3

Figure 8 - Supplement 1 Panel A

Repetition #2

Repetition #3

Figure 8- Supplement 2 Panel A

Figure 8- Supplement 2 Panel B and Panel C

Panel B - Repetition #2

Panel C - Repetition #2

Panel B and Panel C - Repetition #3

Panel B and Panel C - Repetition #4

Dear Dr. González Montoro,

Congratulations on a great revision! Overall, the referees have been positive. However, referee 1 has some further concerns that we ask you to address in a revised version. In this case we will leave it to your discretion whether you want to address the remaining concerns with further experiments or discussion. Regarding the title and abstract, we will work together to edit the final versions of those so you can wait or update now if you prefer. As the referee suggests, please do ensure that figure legends do not include data interpretation.

When you submit your revised version, please also take care of the following editorial items and add this also to your point-by-point response:

1. Please provide an author checklist.
2. Please upload the main figures as individual, high resolution figure files. The legends should be after the references. The supplemental figures should be compiled with their legends into a PDF labeled "Appendix". The appendix will need a table of contents with page numbers, and the figures should be renamed "Appendix Figure S1" etc. and these names also updated in the main manuscript.
3. Please reduce the number of keywords to 5.
4. Please include a Data Availability section as described in our author guide.
5. For funding information: LI 3260/5-1 is in eJP but not in the manuscript; ERC-STG No. 949184 is in the manuscript but not in eJP, please clarify/correct.
6. Please remove the author contribution section from the main manuscript.
7. Please rename the conflict of interest statements to "Disclosure and competing interests statement"
8. Suppl. Tables 1, 4, 5 and 6 should be renamed "Dataset EV1" - "Dataset EV4", with legends added on a separate sheet for each dataset. Suppl. Tables 2 and 3 should be renamed Table EV1 and Table EV2.
9. We include a synopsis of the paper (see <http://emboj.embojpress.org/>). Please provide me with a general summary statement and 3-5 bullet points that capture the key findings of the paper.
10. We also need a summary figure for the synopsis. The size should be 550 wide by 200-440 high (pixels). You can also use something from the figures if that is easier.
11. We require that all figures be referred to in the main manuscript. Please include a figure call out for Fig 4B, C, D, and for Suppl. Table S5. We also require these be called out in order. Please ensure that Table S3 is referred to before Table S4.
12. Please remove the list of abbreviations and the list of supplemental tables at the end of the manuscript.
13. As described in more detail via email, please update the figure legends to describe any reuse of images in F5 and 6D, 7A and 7E, and 7H and 8 and S2B.
14. In the figure legend, Figure 6b should be provided before 6c.
15. Please indicate the statistical test used for data analysis in the legends of figures 2c-d; 5d-e; 7d, supplementary figures 5 supplement 1: 5d-e; 5 supplement 2: 5a-b; 6b-c; 7 supplement 1: a-c; 8 supplement 1: b-c.
16. Please define the yellow lines in the legend of figure 3c-d, f, and 4d.

Thank you for the opportunity to consider your work for publication. I look forward to your revision.

Warm regards,
Kelly

Kelly M Anderson, PhD
Editor, The EMBO Journal
k.anderson@embojournal.org

Revision to The EMBO Journal should be submitted online within 90 days, unless an extension has been requested and approved by the editor; please click on the link below to submit the revision online before 24th May 2024:

Link Not Available

Referee #1:

In the revised manuscript, the authors have added substantial data to support their characterization of TLDC domain proteins in regulating yeast V-ATPase. These results strengthen the functional characterization of Rtc5 and Oxr1 and provide interesting insights into the relationship with the V1 domain subunit C, which leaves the vacuole upon V-ATPase disassembly. Overall, this is an interesting study that provides a comprehensive resource of protein interactions at the yeast vacuole and characterizes a new gene relevant for V-ATPase regulation. A few remaining issues should be addressed prior to publication in EMBO J.

Major points:

1. The authors add new data concerning Vma5 and its relationship to TLDC domain proteins in V-ATPase disassembly. These findings are interesting, but they could be further strengthened. Does Vma5 deletion alter Rtc5 localization? According to the author's model, this might be more relevant than the V-ATPase subunits tested in Fig. 3A. Is the interaction between Rtc5 and Vma5 altered under conditions that promote V-ATPase disassembly?
2. Because genetic manipulations are slow, they are of limited use to dissect the dynamic process of V-ATPase assembly/disassembly. By contrast, the newly added experiments in which yeast are switched to galactose could be used to monitor the kinetics of V-ATPase disassembly in the context of TLDC protein deletion/overexpression. The authors show one time point (20 min), where I have the impression that much of the disassembly has already happened. An optional but insightful experiment would be to include several earlier time points.
3. The authors might want to reconsider the title. They show that the V-ATPase is largely functional in Rtc5 and Oxr1 KO cells. "Yeast TLDC domain-containing proteins control assembly ..." seems like an overstatement in this regard. It is also misleading, considering the suggested role of TLDC proteins in disassembly.

Minor point:

1. On a stylistic note, the figure legends are rather lengthy and tend to include the author's interpretation of the data. They might want to concisely describe the displayed data and allow the readers to draw their own conclusions.
2. Abstract: "equivalent to lysosomes in other eukaryotes". I guess plants also have a vacuole. Maybe "equivalent to lysosomes in animal cells"?

Referee #2:

The authors revised the manuscript very well, and improved it considerably by performing additional experiments, adding more figures or clarifying the existing figures and text. They responded thoroughly and exhaustively to all points raised by me. I believe that the revised manuscript meets high standards of the EMBO Journal and strongly recommend it for publication.

Referee #3:

The authors have addressed my suggestions.

Rev_Com_number: RC-2023-02140

New_manu_number: EMBOJ-2024-116712

Corr_author: González Montoro

Title: Yeast TLDC domain-containing proteins control assembly and subcellular localization of the V-ATPase

Commentss from reviewers

Referee #1:

In the revised manuscript, the authors have added substantial data to support their characterization of TLDC domain proteins in regulating yeast V-ATPase. These results strengthen the functional characterization of Rtc5 and Oxr1 and provide interesting insights into the relationship with the V1 domain subunit C, which leaves the vacuole upon V-ATPase disassembly. Overall, this is an interesting study that provides a comprehensive resource of protein interactions at the yeast vacuole and characterizes a new gene relevant for V-ATPase regulation. A few remaining issues should be addressed prior to publication in EMBO J.

Major points:

1. The authors add new data concerning Vma5 and its relationship to TLDC domain proteins in V-ATPase disassembly. These findings are interesting, but they could be further strengthened. Does Vma5 deletion alter Rtc5 localization? According to the author's model, this might be more relevant than the V-ATPase subunits tested in Fig. 3A. Is the interaction between Rtc5 and Vma5 altered under conditions that promote V-ATPase disassembly?

The experiments regarding Vma5 localization were performed to address V-ATPase disassembly in intact cells, which was suggested by one of the Reviewers in the first revision. We addressed localization specifically of Vma5, because this is the only subunit that shifts subcellular localization *in vivo* upon shift to galactose medium (Tabke et al, 2014 – Also reproduced by us). With the results that we report, we do not claim any special relationship between TLDC domain proteins and Vma5, but rather we strengthen the role of Rtc5 and Oxr1 in promoting disassembly, which we had originally observed with the growth phenotypes and vacuole proteomics experiments. Furthermore, we describe that Rtc5 has a similar behavior to Vma5 regarding localization change. Nonetheless, we have now assessed the localization of Rtc5 in a VMA5 deletion (see below). We observed the same phenotype as for the other deletions of V-ATPase subunits: lack of Vma5 results in mislocalization of Rtc5 from the vacuole. These results are in agreement with the reports that all subunits other than subunit H are required for the complete assembly of the V₀V₁ complex (Jefferies et al, 2008). We have included this additional strain in Figure 3A to make it available in the literature.

Exp. 1

Exp. 2

Exp. 3

2. Because genetic manipulations are slow, they are of limited use to dissect the dynamic process of V-ATPase assembly/disassembly. By contrast, the newly added experiments in which yeast are switched to galactose could be used to monitor the kinetics of V-ATPase disassembly in the context of TLDC protein deletion/overexpression. The authors show one time point (20 min), where I have the impression that much of the disassembly has already happened. An optional but insightful experiment would be to include several earlier time points.

When we set up these experiments, we initially assessed different time points. We found that the responses observed for early time points (2 minutes, 5 minutes) showed a very high cell-to-cell variability, probably based on cell-to-cell differences in the speed of adaptation to the new metabolic conditions, coupled with small experimental handling differences among experiments. For this reason, we performed all experiments including the conditions of 10 and 20 minutes after shift to Galactose. As can be observed in the figure below, these conditions showed very similar results, and we thus considered that there was no benefit in including both. From the two conditions, we chose the 20 minutes one because it showed slightly less variability and because this would make our dataset comparable to the one from Tabke et al 2014.

3. The authors might want to reconsider the title. They show that the V-ATPase is largely functional in *Rtc5* and *Oxr1* KO cells. "Yeast TLDC domain-containing proteins control assembly ..." seems like an overstatement in this regard. It is also misleading, considering the suggested role of TLDC proteins in disassembly.

We thank the reviewer for this suggestion and we have discussed by e-mail with the editors that we will work together on an improved title before final publication. For now our suggestion is:

Yeast TLDC domain proteins regulate the assembly state and subcellular localization of the V-ATPase

Minor point:

1. On a stylistic note, the figure legends are rather lengthy and tend to include the author's interpretation of the data. They might want to concisely describe the displayed data and allow the readers to draw their own conclusions.

We are thankful for this stylistic suggestion and we have now modified the Figure legends to remove all data interpretation.

2. Abstract: "equivalent to lysosomes in other eukaryotes". I guess plants also have a vacuole. Maybe "equivalent to lysosomes in animal cells"?

We are thankful for this suggestion. We have now removed the mention of lysosomes in the abstract and just described vacuoles and their function:

"Yeast vacuoles perform crucial cellular functions as acidic degradative organelles, storage compartments, and signaling hubs."

Referee #2:

The authors revised the manuscript very well, and improved it considerably by performing additional experiments, adding more figures or clarifying the existing figures and text. They responded thoroughly and exhaustively to all points raised by me. I believe that the revised manuscript meets high standards of the EMBO Journal and strongly recommend it for publication.

We are thankful to the reviewer for their time and effort, as well as for their positive evaluation of our work. We additionally thank them for the original suggestions, which have helped us strengthen our manuscript.

Referee # 3:

The authors have addressed my suggestions.

We are thankful to the reviewer for the time invested in re-assessing our manuscript as well as for their original suggestions, which have helped us strengthen it.

Dear Ayelén,

Congratulations on an excellent manuscript, I am pleased to inform you that your manuscript has been accepted for publication in The EMBO Journal. Thank you for your comprehensive response to the referee concerns and for providing detailed source data. It has been a pleasure to work with you to get this to the acceptance stage.

I will begin the final checks on your manuscript before submitting to the publisher next week. Once at the publisher, it will take about three weeks for your manuscript to be published online. As a reminder, the entire review process, including referee concerns and your point-by-point response, will be available to readers. Thank you also for depositing the mass spec data into the PRIDE repository. I notice that this data is not yet accessible. Please ensure that this is made accessible to readers so we can move forward with publication of your manuscript.

I will be in touch throughout the final editorial process until publication. In the meantime, I hope you find time to celebrate!

Warm regards,
Kelly

Kelly M Anderson, PhD
Editor, The EMBO Journal
k.anderson@embojournal.org
